

# Topological holography: The example of the D2-D4 brane system

Nafiz Ishtiaque[1,2*], Seyed Faroogh Moosavian[1,2] and Yehao Zhou[1,2]

**1** Perimeter Institute for Theoretical Physics, Waterloo, ON, Canada
**2** University of Waterloo, Waterloo, ON, Canada

* nishtiaque@perimeterinstitute.ca

## Abstract

We propose a toy model for holographic duality. The model is constructed by embedding a stack of $N$ D2-branes and $K$ D4-branes (with one dimensional intersection) in a 6d topological string theory. The world-volume theory on the D2-branes (resp. D4-branes) is 2d BF theory (resp. 4D Chern-Simons theory) with $GL_N$ (resp. $GL_K$) gauge group. We propose that in the large $N$ limit the BF theory on $\mathbb{R}^2$ is dual to the closed string theory on $\mathbb{R}^2 \times \mathbb{R}_+ \times S^3$ with the Chern-Simons defect on $\mathbb{R} \times \mathbb{R}_+ \times S^2$. As a check for the duality we compute the operator algebra in the BF theory, along the D2-D4 intersection – the algebra is the Yangian of $\mathfrak{gl}_K$. We then compute the same algebra, in the guise of a scattering algebra, using Witten diagrams in the Chern-Simons theory. Our computations of the algebras are exact (valid at all loops). Finally, we propose a physical string theory construction of this duality using a D3-D5 brane configuration in type IIB – using supersymmetric twist and $\Omega$-deformation.



# 1 Introduction and Summary

Holography is a duality between two theories, referred to as a bulk theory and a boundary theory, in two different space-time dimensions that differ by one [1–3]. A familiar manifestation of the duality is an equality of the partition function of the two theories - the boundary partition function as a function of sources, and the bulk partition function as a function of boundary values of fields. This in turns implies that correlation functions of operators in the boundary theory can also be computed in the bulk theory by varying boundary values of its fields [2,3]. This dictionary has been extended to include expectation values of non-local operators as well [4–7]. This is a strong-weak duality, relating a strongly coupled boundary theory to a weakly coupled bulk theory. As is usual in strong-weak dualities, exact computations on both sides of the duality are hard. Topological theories have provided interesting examples of holographic dualities where exact computations are possible [8–13].

Recently, it has been shown that some instances of holography can be described as an algebraic relation, known as Koszul duality, between the operator algebras of the two dual theories [14,15]. It was previously known that the algebra of operators restricted to a line in the holomorphic twist of 4d $\mathcal{N} = 1$ gauge theory with the gauge group $\mathrm{GL}_K$ is the Koszul dual of the Yangian of $\mathfrak{gl}_K$ [16]. In light of the connection between Koszul duality and holography, this result suggests that if there is a theory whose local operator algebra is the Yangian of $\mathfrak{gl}_K$ then that theory could be a holographic dual to the twisted 4d theory. Since the inception of holography, brane constructions played a crucial role in finding dual theories. It turns out that

the particular twisted 4d theory is the world-volume theory of $K$ D4-branes[1] embedded in a particular 6d topological string theory [18]. Since the operators whose algebra is the Koszul dual of the Yangian lives on a line, it is a reasonable guess that we need to include some other branes that intersect this stack of D4-branes along a line. Beginning from such motivations we eventually find (and demonstrate in this paper) that the correct choice is to embed a stack of $N$ D2-branes in the 6d topological string theory so that they intersect the D4-branes along a line. The world-volume theory of the D2-branes is 2d BF theory with $GL_N$ gauge group coupled to a fermionic quantum mechanics along the D2-D4 intersection. The algebra of gauge invariant local operators along this D2-D4 intersection is precisely the Yangian of $\mathfrak{gl}_K$.

This connected the D2 world-volume theory and the D4 world-volume theory via holography in the sense of Koszul duality. The connection between these two theories via holography in the sense of [2, 3] was still unclear. In this paper we begin to establish this connection. We take the D2-brane world-volume theory to be our boundary theory. This implies that the closed string theory in some background, including the D4-brane theory should give us the dual bulk theory. In the boundary theory, we consider the OPE (operator product expansion) algebra of gauge invariant local operators, we argue that this algebra can be computed in the bulk theory by computing a certain algebra of scatterings from the asymptotic boundary in the limit $N \to \infty$. Our computation of the boundary local operator algebra using the bulk theory follows closely the computation of boundary correlation functions using Witten diagrams [3].

The Feynman diagrams and Witten diagrams we compute in this paper have at most two loops, however, we would like to emphasize that the identification we make between the operator algebras and the Yangian is true at *all* loop orders. In the boundary theory (D2-brane theory) this will follow from the simple fact that, for the operator product that we shall compute, there will be no non-vanishing diagrams beyond two loops. In the bulk theory this follows from a certain classification of anomalies in the D4-brane theory [19] and independently from the very rigid nature of the deformation theory of the Yangian. We explain some of these mathematical aspects underlying our results in appendix B – we begin the appendix with motivations for and a light summary of the purely mathematical results to follow.

We note that there is a long history of studying links between quantum integrable systems soluble by Yangians and quantum affine algebras on one hand and supersymmetric gauge theory dynamics in the vacuum sector on the other hand – including early pioneering work [20] and subsequent developments [21–25]. In this paper we study particular examples of (quasi) topological gauge theories with similar connection to Yangins, the novelty is that we propose a certain holographic duality linking the theories.

A particular motivation for studying these topological/holomorphic theories and their duality is that these theories can be constructed from certain brane setup in a physical 10d string theory. In particular, we can identify these theories as certain supersymmetric subsectors of some theories on D-branes in type IIB string theory by applying supersymmetric twists and $\Omega$-deformation.

The organization of the paper is as follows. In §2 we describe, in general terms, how holographic duality in the sense of [2, 3] leads to the construction of two isomorphic algebras from the two dual theories. In §3 we start from a brane setup involving $N$ D2-branes and $K$ D4-branes in a 6d topological string theory and describe the two theories that we claim to be holographic dual to each other. In §4 we compute the local operator algebra in the D2-brane theory, this algebra will be the Yangian $Y(\mathfrak{gl}_K)$ in the limit $N \to \infty$. In §5 we show that the same algebra can be computed using Witten diagrams in the D4-brane theory. In the last section, §6, we propose a physical string theory realization of the duality.

---

[1]We are following the convention of [17], according to which, by a D$p$-brane in topological string theory we mean a brane with a $p$-dimensional world-volume. In §6 when we discuss branes in type IIB string theory, we of course use the standard convention that a D$p$-brane refers to a brane with $(p + 1)$-dimensional world-volume.

**Relevant New Literature.** Since the preprint of this paper appeared online, there has been a series of interesting developments exploring the idea of topological holography – also referred to as *twisted holography* – we mention the ones we found conceptually related to this paper. In [26] the authors studied the holographic duality between a gauged $\beta\gamma$ system and a Kodaira-Spencer theory on the SL(2, $\mathbb{C}$) manifold (the deformed conifold) – emphasizing the role of global symmetry matching. Authors of [27] studied a twisted holography closely related to AdS$_3$/CFT$_2$ duality, they highlight in particular the link to Koszul duality and contains a rare (in physics literature) introduction to Koszul duality. [28] computes certain operator algebra of a topological quantum mechanics living at the intersection of M2 and M5 branes in an $\Omega$-deformed M-theory using Feynman diagram techniques similar to the ones we shall use in our computations. In this setup the M2 and M5 branes play roles analogous to certain D3 and D5 branes that we shall study in §6. The M5 brane world-volume theory in the $\Omega$-background is the 5d Chern-Simons theory, a close cousin of the 4d Chern-Simons theory we are going to study. This M2-M5 brane setup in $\Omega$-deformed M-theory is also studied in [29, 30] in the context of twisted holography.

## 2 Isomorphic algebras from holography

In [2,3], two theories, $\mathcal{T}_{\text{bd}}$ and $\mathcal{T}_{\text{bk}}$ were considered on two manifolds $M_1$ and $M_2$ respectively, with the property that $M_1$ was conformally equivalent to the boundary of $M_2$. The theory $\mathcal{T}_{\text{bd}}$ was considered with background sources, schematically represented by $\phi$. The theory $\mathcal{T}_{\text{bk}}$ was such that the values of its fields at the boundary $\partial M_2$ can be coupled to the fields of $\mathcal{T}_{\text{bd}}$, then $\mathcal{T}_{\text{bk}}$ was quantized with the fields $\phi$ as the fixed profile of its fields at the boundary $\partial M_2$. These two theories were considered to be holographic dual when their partition functions were equal:

$$Z_{\text{bd}}(\phi) = Z_{\text{bk}}(\phi). \tag{1}$$

This equality leads to an isomorphism of two algebras constructed from the two theories, as follows. Consider local operators $O_i$ in $\mathcal{T}_{\text{bd}}$ with corresponding sources $\phi^i$. The partition function $Z_{\text{bd}}(\phi)$ with these sources has the form:

$$Z_{\text{bd}}(\phi) = \int \mathcal{D}X \exp\left(-\frac{1}{\hbar}S_{\text{bd}} + \sum_i O_i \phi^i\right), \tag{2}$$

where $X$ schematically represents all the dynamical fields in $\mathcal{T}_{\text{bd}}$. Correlation functions of the operators $O_i$ can be computed from the partition function by taking derivatives with respect to the sources:

$$\langle O_1(p_1)\cdots O_n(p_n)\rangle = \frac{1}{Z_{\text{bd}}(\phi)}\frac{\delta}{\delta\phi^1(p_1)}\cdots\frac{\delta}{\delta\phi^n(p_n)}Z_{\text{bd}}(\phi)\bigg|_{\phi=\phi_0}. \tag{3}$$

We can consider the algebra generated by the operators $O_i$ using operator product expansion (OPE). However, this algebra is generally of singular nature, due to its dependence on the location of the operators and the possibility of bringing two operators too close to each other. In specific cases, often involving supersymmetry, we can consider sub-sectors of the operator spectrum that can generate algebras free from such contact singularity, so that a position independent algebra can be defined.[2] Suppose the set $\{O_i\}$ represents such a restricted set with an algebra:

$$O_i O_j = C_{ij}^k O_k. \tag{4}$$

---

[2]Various *chiral rings*, for example.

Let us call this algebra $\mathcal{A}^{\mathrm{Op}}(\mathcal{T}_{\mathrm{bd}})$. In terms of the partition function and the sources the relation (4) becomes:

$$\frac{\delta}{\delta\phi^i}\frac{\delta}{\delta\phi^j}Z_{\mathrm{bd}}(\phi)\bigg|_{\phi=0} = C_{ij}^k\frac{\delta}{\delta\phi^k}Z_{\mathrm{bd}}(\phi)\bigg|_{\phi=\phi_0}. \tag{5}$$

The statement of duality (1) then tells us that the above equation must hold if we replace $Z_{\mathrm{bd}}$ by $Z_{\mathrm{bk}}$:

$$\frac{\delta}{\delta\phi^i}\frac{\delta}{\delta\phi^j}Z_{\mathrm{bk}}(\phi)\bigg|_{\phi=0} = C_{ij}^k\frac{\delta}{\delta\phi^k}Z_{\mathrm{bk}}(\phi)\bigg|_{\phi=\phi_0}. \tag{6}$$

This gives us a realization of the operator algebra $\mathcal{A}^{\mathrm{Op}}(\mathcal{T}_{\mathrm{bd}})$ in the dual theory $\mathcal{T}_{\mathrm{bk}}$.

This suggests a check for holographic duality. The input must be two theories, say $\mathcal{T}_{\mathrm{bd}}$ and $\mathcal{T}_{\mathrm{bk}}$, with some compatibility:

- $\mathcal{T}_{\mathrm{bd}}$ can be put on a manifold $M_1$ and $\mathcal{T}_{\mathrm{bk}}$ can be put on a manifold $M_2$ such that $\partial M_2 \cong M_1$, where equivalence between $\partial M_2$ and $M_1$ must be equivalence of whatever geometric/topological structure is required to define $\mathcal{T}_{\mathrm{bd}}$.[3]

- Quantum numbers of fields of the two theories are such that the boundary values of the fields in $\mathcal{T}_{\mathrm{bk}}$ can be coupled to the fields in $\mathcal{T}_{\mathrm{bd}}$.[4]

Suppose $\mathcal{T}_{\mathrm{bd}}$ has a sub-sector of its operator spectrum that generates a suitable algebra[5] $\mathcal{A}^{\mathrm{Op}}(\mathcal{T}_{\mathrm{bd}})$. We denote the operators in this algebra by $\{O_i\}$ with corresponding sources $\phi^i$. According to the first compatibility condition these sources can be thought of as boundary values for the fields in $\mathcal{T}_{\mathrm{bk}}$, so that we can quantize $\mathcal{T}_{\mathrm{bk}}$ by fixing the values of the fields at the boundary to be $\phi$. Then, we can define another algebra by taking functional derivatives of the partition function of $\mathcal{T}_{\mathrm{bk}}$ with respect to $\phi$, as in (6). Let's call this algebra the *scattering algebra*, $\mathcal{A}^{\mathrm{Sc}}(\mathcal{T}_{\mathrm{bk}})$. Now a check of holographic duality is the following isomorphism:

$$\mathcal{A}^{\mathrm{Op}}(\mathcal{T}_{\mathrm{bd}}) \cong \mathcal{A}^{\mathrm{Sc}}(\mathcal{T}_{\mathrm{bk}}). \tag{7}$$

This is the general idea that we employ in this paper to check holographic duality. The operator algebra $\mathcal{A}^{\mathrm{Op}}(\mathcal{T}_{\mathrm{bd}})$ can be computed in perturbation theory using Feynman diagrams and we can use Witten diagrams, introduced in [3], to compute the scattering algebra $\mathcal{A}^{\mathrm{Sc}}(\mathcal{T}_{\mathrm{bk}})$. We will do this concretely in the rest of this paper.

## 3 The dual theories

### 3.1 Brane construction

The quickest way to introduce the theories we claim to be holographic dual to each other is to use branes to construct them. Our starting point is a 6d topological string theory, in particular, the product of the A-twisted string theory on $\mathbb{R}^4$ and the B-twisted string theory on $\mathbb{C}$ [18]. The brane setup is the following:

|  | $\mathbb{R}_v$ | $\mathbb{R}_w$ | $\mathbb{R}_x$ | $\mathbb{R}_y$ | $\mathbb{C}_z$ | No. of branes |
|---|---|---|---|---|---|---|
| D2 | 0 | × | × | 0 | 0 | $N$ |
| D4 | 0 | 0 | × | × | × | $K$ |

$\tag{8}$

---

[3]In case of AdS/CFT, it is conformal equivalence, perfect for defining the CFT. In this paper we shall only be concerned with topology.

[4]To clarify, this is merely a compatibility condition for the duality, the two dual theories are not supposed to be coupled, they are supposed to be alternative descriptions of the same dynamics.

[5]Ideally we should consider the OPE algebra of all the operators, but if that is too hard, we can restrict to smaller sub-sectors which may still provide a non-trivial check.

The subscripts denote the coordinates we use to parameterize the corresponding directions, and it is implied that the complex direction is parameterized by the complex variable $z$, along with its conjugate variable $\bar{z}$.

Our first theory, denoted by $\mathcal{T}_{\mathrm{bd}}$, is the theory of open strings on the stack of D2-branes. This is a 2d topological gauge theory with the complexified gauge group $\mathrm{GL}_N$ [18]. The intersection of the D2-branes with the D4-branes introduces a line operator in this theory. We describe this theory in §3.3.

Next, we consider the product of two theories, open string theory on the stack of D4-branes, and closed string theory on the 6d background sourced by the stack of D2-branes. The theory on the stack of D4-branes is a 4d analogue of Chern-Simons (CS) gauge theory with the complexified gauge group $\mathrm{GL}_K$ [18]. As it does in the theory on the D2-branes, the intersection between the D2 and the D4-branes introduces a line operator in this theory as well. This line sources a flux supported on the 3-sphere linking the line. Our bulk theory is the Kaluza-Klein compactification of the total 6d theory – 6d closed string theory coupled to 4d CS theory – on the 3-sphere. We describe the 4d CS theory in §3.4. Let us describe the closed sting theory in the next section.

## 3.2 The closed string theory

The closed string theory, denoted by $\mathcal{T}_{\mathrm{cl}}$, is a product of Kodaira-Spencer (also known as BCOV) theory [31, 32] on $\mathbb{C}$ and Kähler gravity [33] on $\mathbb{R}^4$, along with a 3-form flux sourced by the stack of D2-branes.[6] Fields in this theory, including ghosts and anti-fields, are given by:

$$\text{Set of fields,} \quad \mathcal{F} := \Omega^{\bullet}(\mathbb{R}^4) \otimes \Omega^{\bullet,\bullet}(\mathbb{C}), \tag{9}$$

i.e., the fields are differential forms on $\mathbb{R}^4$ and $(p,q)$-forms on $\mathbb{C}$.[7] The linearized BRST differential acting on these fields is a sum of the de Rham differential on $\mathbb{R}^4$ and the Dolbeault differential on $\mathbb{C}$, leading to the following equation of motion:

$$\left(\mathrm{d}_{\mathbb{R}^4} + \overline{\partial}_{\mathbb{C}}\right)\alpha = 0, \qquad \alpha \in \mathcal{F}. \tag{10}$$

The background field sourced by the D2-branes, let it be denoted by $F_3 \in \mathcal{F}$, measures the flux through a topological $S^3$ surrounding the D2-branes, it can be normalized as:

$$\int_{S^3} F_3 = N. \tag{11}$$

Note that the $S^3$ is only topological, i.e., continuous deformation of the $S^3$ should not affect the above equation. This is equivalent to saying that, the 3-form must be closed on the complement of the support of the D2-branes:

$$\mathrm{d}_{\mathbb{R}^4 \times \mathbb{C}} F_3(p) = 0, \quad p \notin \mathrm{D2}. \tag{12}$$

Here the differential is the de Rham differential for the entire space, i.e., $\mathrm{d}_{\mathbb{R}^4 \times \mathbb{C}} = \mathrm{d}_{\mathbb{R}^4} + \overline{\partial}_{\mathbb{C}} + \partial_{\mathbb{C}}$. Moreover, as a dynamically determined background it is also constrained by the equation of motion (10). In addition to satisfying these equations, $F_3$ must also be translation invariant along the directions parallel to the D2-branes. The solution is:

$$F_3 = \frac{iN}{2\pi(v^2 + y^2 + z\bar{z})^2}(v\,\mathrm{d}y \wedge \mathrm{d}z \wedge \mathrm{d}\bar{z} - y\,\mathrm{d}v \wedge \mathrm{d}z \wedge \mathrm{d}\bar{z} - 2\bar{z}\,\mathrm{d}v \wedge \mathrm{d}y \wedge \mathrm{d}z). \tag{13}$$

---

[6]This flux is analogous to the 5-form flux sourced by the stack of D3-branes in Maldacena's setup of AdS/CFT duality between $\mathcal{N} = 4$ super Yang-Mills and supergravity on $\mathrm{AdS}_5 \times S^5$ [1].

[7]We are not being careful about the degree (ghost number) of the fields since this will not be used in this paper.

In general, a closed string background like this might deform the theory on a brane, however, the pullback of the form (13) to the D4-branes vanishes:

$$\iota^* F_3 = 0 \,, \tag{14}$$

where $\iota : \mathbb{R}^2_{x,y} \times \mathbb{C}_z \hookrightarrow \mathbb{R}^4_{v,w,x,y} \times \mathbb{C}_z$ is the embedding of the D4-branes into the entire space. So the closed string background leaves the D4-brane world-volume theory unaffected.[8] Such a lack of backreaction is a rather drastic simplification of the holographic setup which can occur in a topological setting such as ours (see also [15]) but this is not a general feature. For examples of topological holography with nontrivial backreaction see [26, 27].

The flux (13) signals a change in the topology of the closed string background:

$$\mathbb{R}^4_{v,w,x,y} \times \mathbb{C}_z \to \mathbb{R}^2_{w,x} \times \mathbb{R}_+ \times S^3 \,, \tag{15}$$

where the $\mathbb{R}_+$ is parameterized by $r := \sqrt{v^2 + y^2 + z\bar{z}}$. This change follows from requiring translation symmetry in the directions parallel to the D2-branes and the existence of an $S^3$ supporting the flux $F_3$. This $S^3$ is analogous to the $S^5$ in the D3-brane geometry supporting the 5-form flux sourced by the said D3-branes in Maldacena's AdS/CFT [1]. The coordinate $r$ measures distance from the location of the D2-branes. In the absence of a metric we can only distinguish between the two extreme limits $r \to 0$ and $r \to \infty$. The $r \to 0$ region would be analogous to Maldacena's near horizon geometry. In our topological setting there is no distinction between near and distant, and we treat the entire $\mathbb{R}^2_{w,x} \times \mathbb{R}_+ \times S^3$ as analogous to the near horizon geometry. This makes $\mathbb{R}^2_{w,x} \times \mathbb{R}_+$ analogous to the AdS geometry. We recall that, in the AdS/CFT correspondence the location of the black branes and the boundary of AdS correspond to two opposite limits of the non-compact coordinate transverse to the branes. In our case $r = 0$ corresponds to the location of the D2-branes, and we treat the plane at $r = \infty$, namely:

$$\mathbb{R}^2_{w,x} \times \{\infty\} \,, \tag{16}$$

as analogous to the asymptotic boundary of AdS.

The D4-branes in (8) appear as a defect in the closed string theory, they are analogous to the D5-branes that were considered in [34] in Maldacena's setup of AdS/CFT, where they were presented as holographic duals of Wilson loops in 4d $\mathcal{N} = 4$ super Yang-Mills. For the world-volume of these branes, the transformation (15) corresponds to:

$$\mathbb{R}^2_{x,y} \times \mathbb{C}_z \to \mathbb{R}_x \times \mathbb{R}_+ \times S^2 \,, \tag{17}$$

where the $\mathbb{R}_+$ direction is parameterized by $r' := \sqrt{y^2 + z\bar{z}}$. The intersection of the boundary plane (16) and this world-volume is then the line:

$$\mathbb{R}_x \times \{\infty\}, \tag{18}$$

at infinity of $r'$. We draw a cartoon representing some aspects of the brane setup in figure 1.

We can now talk about two theories:

1. The 2d world-volume theory of the D2-branes. This is our analogue of the CFT (with a line operator) in AdS/CFT.

---

[8]The flux (13) is the only background turned on in the closed string theory. This can be argued as follows: The D2-branes introduce a 4-form source (the Poincaré dual to the support of the branes) in the closed string theory. This form can appear on the right hand side of the equation of motion (10) only for a 3-form field $\alpha$, which can then have a non-trivial solution, as in (13). Furthermore, since the equation of motion (10) is free, the non-trivial solution for the 3-form field does not affect any other field.

2. The effective[9] 3d theory on world-volume $\mathbb{R}^2_{w,x} \times \mathbb{R}_+$ with a defect supported on $\mathbb{R}_x \times \mathbb{R}_+$. This is our analogue of the gravitational theory in AdS background (with defect) in AdS/CFT.

To draw parallels once more with the traditional dictionary of AdS/CFT [1–3], we should establish a duality between the operators in the D2-brane world-volume theory and variations of boundary values of fields in the "gravitational" theory on $\mathbb{R}^2_{w,x} \times \mathbb{R}_+$ (the boundary is $\mathbb{R}_{w,x} \times \{\infty\}$). Both of these surfaces have a line operator/defect and this leads to two types of operators, ones that are restricted to the line, and others that can be placed anywhere. Local operators in a 2d surface are commuting, unless they are restricted to a line. Therefore, in both of our theories, we have non-commutative associative algebras whose centers consist of operators that can be placed anywhere in the 2d surface. For this paper we are mostly concerned with the non-commuting operators:

1. Operators in the world-volume theory of the D2-branes that are restricted to the D2-D4 intersection.

2. Variations of boundary values of fields in the effective theory along the intersection (18) of the boundary $\mathbb{R}^2_{w,x} \times \{\infty\}$ and the defect on $\mathbb{R}_x \times \mathbb{R}_+$.

In physical string theory, the analogue of the D4-branes would be coupled to the closed string modes. In an appropriate large $N$ low energy limit such gravitational couplings can be ignored, leading to the notion of *rigid holography* [35]. Since we are working with topological theory at large $N$, we are assuming such a decoupling.

The computations in the "gravitational" side will be governed by the effective dynamics on the defect on $\mathbb{R}_x \times \mathbb{R}_+$. This is the Kaluza-Klein compactification of the world-volume theory of the D4-branes (with a line operator due to D2-D4 intersection). This 4d theory (which we describe in §3.4) is familiar from previous works such as [19]. Therefore we use the 4d dynamics, instead of the effective 2d one for our computations. In terms of Witten diagrams (which we compute in §5) this means that while we have a 1d boundary, the propagators are from the 4d theory and the bulk points are integrated over the 4d world-volume $\mathbb{R}^2 \times \mathbb{C}$. We take the boundary line to be at $y = \infty$ with some fixed coordinate $z$ in the complex direction. In future we shall refer to this line as $\ell_\infty(z)$:

$$\ell_\infty(z) := \mathbb{R}_x \times \{y = \infty\} \times \{z\}. \tag{19}$$

**A cartoon of our setup**

Let us make a diagrammatic summary of our brane setup in Fig 1. In the figure we draw the non-compact part, namely $\mathbb{R}^2_{w,x} \times \mathbb{R}_+$, of the closed string background (the right hand side of (15)). We identify the location of the 2d black brane and the defect D4-branes, the asymptotic boundary $\mathbb{R}^2_{w,x} \times \{\infty\}$, and the intersection between the boundary and the defect. At the top of the picture, parallel to the asymptotic boundary, we also draw the D2-branes. We draw the D2-branes independently of the rest of the diagram because the D2-branes do not exist in the backreacted bulk, they become the black brane. However, traditionally, parallels are drawn between the asymptotic boundary and the brane sourcing the bulk (the D2-brane in this case). The dots on the asymptotic boundary represent local variations of boundary values of fields in the bulk theory $\mathcal{T}_{\text{bk}}$. The corresponding dots on the D2-brane represent the local operators in the boundary theory $\mathcal{T}_{\text{bd}}$ that are dual to the aforementioned variations. By the duality map in the figure we schematically represent boundary excitations in the bulk theory corresponding to some local operators in the dual description of the same dynamics in terms of the boundary theory.

---

[9]Effective, in the sense that this is the Kaluza-Klein reduction of a 6d theory with three compact directions, though we don't want to lose any dynamics, i.e., we don't throw away massive modes.

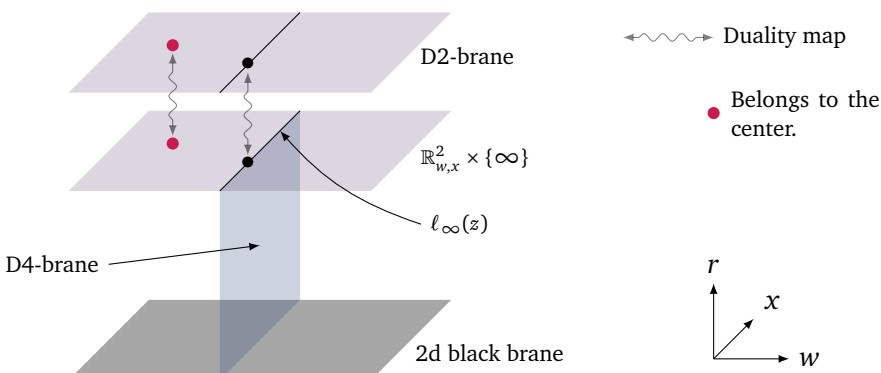

Figure 1: D2-brane, and the non-compact part of the backreacted bulk.

### 3.3 BF: The theory on D2-branes

This is a 2d topological gauge theory on the stack of $N$ D2-branes (see (8)), supported on $\mathbb{R}^2_{w,x}$, with complexified gauge group $GL_N$. The field content of this theory is:

| Field | Valued in |
|---|---|
| B | $\Omega^0(\mathbb{R}^2) \times \mathfrak{gl}_N$ |
| A | $\Omega^1(\mathbb{R}^2) \times \mathfrak{gl}_N$ |

$$\quad (20)$$

A is a Lie algebra valued connection and B is a Lie algebra valued scalar, both complex. The curvature of the connection is denoted as $F = dA + A \wedge A$. The action is given by:

$$S_{\text{BF}} := \int_{\mathbb{R}^2_{w,x}} \text{tr}_{\mathbf{N}}(BF), \qquad (21)$$

where the trace is taken in the fundamental representation of $\mathfrak{gl}_N$.

We consider this theory in the presence of a line operator supported on $\mathbb{R}_x \times \{0\}$, caused by the intersection of the D2 and D4-branes. The line operator is defined by a fermionic quantum mechanical system living on it.[10] The fields in the quantum mechanics (QM) are $K$ fundamental (of $\mathfrak{gl}_N$) fermions and their complex conjugates:

| Field | Valued in | |
|---|---|---|
| $\psi^i$ | $\Omega^0(\mathbb{R}_x) \times \mathbf{N}$ , | $i \in \{1, \cdots, K\},$ |
| $\overline{\psi}_i$ | $\Omega^0(\mathbb{R}_x) \times \overline{\mathbf{N}}$ | |

$$\quad (22)$$

where $\mathbf{N}$ refers to the fundamental representation of $\mathfrak{gl}_N$ and $\overline{\mathbf{N}}$ to the anti-fundamental. The fermionic system has a global symmetry $GL_N \times GL_K$. These fermions couple naturally to the $\mathfrak{gl}_N$ connection A of the BF theory. The action for the QM is given by:

$$S_{\text{QM}} := \int_{\mathbb{R}_x} \left( \overline{\psi}_i d\psi^i + \overline{\psi}_i A \psi^i + \overline{\psi}_j A^j_i \psi^i \right), \qquad (23)$$

where we have introduced a *background* $\mathfrak{gl}_K$-valued gauge field $A \in \Omega^1(\mathbb{R}_x) \times \mathfrak{gl}_K$. Note that the terms in the above action are made $\mathfrak{gl}_N$ invariant by pairing up elements of $\mathbf{N}$ with elements of the dual space $\overline{\mathbf{N}}$.

---

[10]This closely resembles the D3-D5 system in type IIB string theory considered in [34], there too a fermionic quantum mechanics lived on the intersection, giving rise to Wilson lines upon integrating out the fermions. Note that we could have considered bosons, instead of fermions, living on the line, without any significant change to our following computations. This would be similar to the D3-D3 system considered in [34, 36].

Our first theory is this BF theory with the line operator, schematically:

$$\mathcal{T}_{\mathrm{bd}} := \mathrm{BF}_N \otimes_N \mathrm{QM}_{N \times K},$$ (24)

where the subscripts on BF and QM refer to the symmetries ($\mathrm{GL}_N$ and $\mathrm{GL}_N \times \mathrm{GL}_K$ respectively) of the respective theories and the subscript on $\otimes$ implies that the $\mathrm{GL}_N$ is gauged. There are two types of gauge ($\mathfrak{gl}_N$) invariant operators in the theory:[11]

$$\text{for } n \in \mathbb{N}_{\geq 0}, \qquad \begin{array}{ll} \text{operators restricted to } \mathbb{R}_x: & O^i_j[n] := \frac{1}{\hbar}\overline{\psi}_j B^n \psi^i, \\ \text{operators } \textit{not} \text{ restricted to } \mathbb{R}_x: & O[n] := \frac{1}{\hbar}\mathrm{tr}_N B^n. \end{array}$$ (25)

Unrestricted local operators in two topological dimensions can be moved around freely, implying that for any $n \geq 0$, the operator $O[n]$ commutes with all of the operators defined above.[12] The operator algebra of the 2d BF theory consists of all theses operators but in this paper we focus on the non-commuting ones, in other words we, focus on the quotient of the full operator algebra of the boundary theory by its center.[13] We shall compute their Lie bracket in §4, which will establish an isomorphism with the Yangian. Had we included the commuting operators as well we would have found a *central extension* of the Yangian. In sum, the operator algebra we construct from the theory $\mathcal{T}_{\mathrm{bd}}$ is:

$$\mathcal{A}^{\mathrm{Op}}(\mathcal{T}_{\mathrm{bd}}) := \left(O^i_j[n], O[n]\right)/(O[n]).$$ (26)

By the notation $(x, y, \cdots)$ we mean the algebra generated by the set of operators $\{x, y, \cdots\}$ over $\mathbb{C}$.

*Remark* 1 (A speculative link). Note that it is possible to lift our D2 and D4 branes to type IIB string theory while maintaining a one dimensional intersection. This results in a D3-D5 setup (studied in particular in [34]) where on the D3 brane we find the $\mathcal{N} = 4$ Yang-Mills theory with a Wilson line.[14] In [38–40], the authors considered local operators in the $\mathcal{N} = 4$ Yang-Mills that are restricted to certain Wilson lines. With the proper choice of Wilson lines, Localization reduces this setup to 2d Yang-Mills theory with Wilson lines – local operator insertions along the Wilson lines in 4d reduce to local operator insertions along the Wilson lines in 2d [41]. 2d BF theory is the zero coupling limit of 2d Yang-Mills theory. We therefore expect the algebra constructed in this section to be related to the algebra constructed in the aforementioned references, at least in some limit.[15] The algebra in [40] would correspond to the $K = 1$ instance of our algebra, it may be an interesting check to compute the analogue of the algebra in [40] for higher $K$. $\triangle$

### 3.4 4d Chern-Simons: The theory on D4-branes

This is a 4d gauge theory on the stack of $K$ D4-branes, supported on $\mathbb{R}^2_{x,y} \times \mathbb{C}_z$ with the line $L := \mathbb{R}_x \times (0,0,0)$ removed and with the (complexified) gauge group $\mathrm{GL}_K$. The notation of distinguishing directions by $\mathbb{R}$ and $\mathbb{C}$ is meant to highlight the fact that observables in this theory depend only on the topology of the real directions and depend holomorphically on the complex direction. In particular, they are independent of the coordinates $x$ and $y$ that parameterize the $\mathbb{R}^2$, and depend holomorphically on $z$ which parameterizes the $\mathbb{C}$. Due to the removed line, we can represent the topology of the support of this theory as (c.f. (17)):

$$M := \mathbb{R} \times \mathbb{R}_+ \times S^2.$$ (27)

---

[11]The $\hbar^{-1}$ appears in these definitions because the action (23) will appear in path integrals as $\exp\left(-\hbar^{-1}S_{\mathrm{QM}}\right)$, which means functional derivatives with respect to $A^i_j$ inserts operators that carry $\hbar^{-1}$.

[12]These operators are represented by the red dot on the D2-brane in figure 1.

[13]We shall similarly quotient out the center in the bulk theory as well.

[14]It is also interesting to note that the D5 brane in an Omega background reproduces the 4d CS theory [37].

[15]We thank Shota Komatsu for pointing out this interesting possibility.

The field of this theory is just a connection:

$$
\begin{array}{c|c}
\text{Field} & \text{Valued in} \\
\hline
A & \frac{\Omega^1(\mathbb{R}^2 \times \mathbb{C} \setminus L)}{(\mathrm{d}z)} \otimes \mathfrak{gl}_K
\end{array} \,.
\tag{28}
$$

The above notation simply means that $A$ is a $\mathfrak{gl}_K$-valued 1-form without a $\mathrm{d}z$ component. The theory is defined by the action:[16]

$$
S_{\mathrm{CS}} := \frac{i}{2\pi} \int_M \mathrm{d}z \wedge \mathrm{CS}(A) \,,
\tag{29}
$$

where $\mathrm{CS}(A)$ refers to the standard Chern-Simons Lagrangian:

$$
\mathrm{CS}(A) = \mathrm{tr}_{\mathbf{K}} \left( A \wedge \mathrm{d}A + \frac{2}{3} A \wedge A \wedge A \right) \,,
\tag{30}
$$

where the trace is taken over the fundamental representation of $\mathfrak{gl}_K$. This theory is a 4d analogue of the, perhaps more familiar, 3d Chern-Simons theory. We shall therefore refer to it as the 4d Chern-Simons theory and sometimes denote it by $\mathrm{CS}_K^4$ or just CS.

The removal of the line $L$ from $\mathbb{R}^2 \times \mathbb{C}$ is caused by the D2-D4 brane intersection. Note that from the perspective of the CS theory, the D2-D4 intersection looks like a Wilson line. This means that we should be quantizing the CS theory on $M$ with a background electric flux supported on the $S^2$ inside $M$. Alternatively, we can quantize the CS theory on $\mathbb{R}^2 \times \mathbb{C}$ with a Wilson line inserted along $L$.[17] The choice of representation for the Wilson line is determined by the number, $N$, of D2-branes – let us denote this representation as $\varrho : \mathfrak{gl}_K \to \mathrm{End}(V)$. With this choice, the Wilson line is defined as the following operator:

$$
W_\varrho(L) := P \exp \left( \int_L \varrho(A) \right) \,,
\tag{31}
$$

where $P \exp$ implies path ordered exponentiation, made necessary by the fact that the exponent is matrix valued. The above operator is valued in $\mathrm{End}(V)$. This in general means that the following expectation value:

$$
\left\langle W_\varrho(L) \right\rangle = \frac{\int \mathcal{D}A \, W_\varrho(L) \exp\left(-\frac{1}{\hbar} S_{\mathrm{CS}}\right)}{\int \mathcal{D}A \exp\left(-\frac{1}{\hbar} S_{\mathrm{CS}}\right)} \,,
\tag{32}
$$

is valued in $\mathrm{Hom}(\mathcal{H}_{-\infty} \otimes V, \mathcal{H}_{+\infty} \otimes V)$, where $\mathcal{H}_{\pm\infty}$ are the Hilbert spaces of the $\mathrm{CS}_K^4$ theory on the Cauchy surfaces perpendicular to $L$ at $x = \pm\infty$, in the absence of the Wilson line. However, for the particular CS theory, these Hilbert spaces are trivial and we end up with a map that transports vectors in $V$ from $x = -\infty$ to $x = +\infty$:

$$
\left\langle W_\varrho(L) \right\rangle : V_{-\infty} \to V_{+\infty} \,.
\tag{33}
$$

In picture this operator may be represented as:

$$
\left\langle W_\varrho(L) \right\rangle : \qquad V \;\begin{array}{c} W_\varrho(L) \\ \longrightarrow \end{array}\; V \qquad .
\tag{34}
$$

$$
x = -\infty \qquad\qquad x = +\infty
$$

---

[16]This theory was proposed in [42] to explain the representation theory of quantum affine algebras and more recently studied in [16, 19, 43, 44] as a way of producing integrable lattice models using Wilson lines.

[17]Recall that in case of the BF theory the line operator at the D2-D4 intersection was described by a fermionic QM. We could do the same in this case. However, in this case it proves more convenient to integrate out the fermion, leaving a Wilson line in its place. The mechanism is the same that appeared for intersection of D3 and D5-branes in physical string theory [34].

The CS theory is quantized with some fixed boundary profile of the connection along the boundary $\mathbb{R}_x \times \{\infty\} \times S^2$.[18] To express the dependence of expectation values on this boundary value we put a subscript, such as $\langle W_\varrho(L) \rangle_A$. Since we are essentially interested in the Kaluza-Klein reduced theory on $\mathbb{R}_x \times \mathbb{R}_+$ we mostly care about the value of the connection along the boundary line (defined in (19)) $\ell_\infty(z) \subset \mathbb{R}_x \times \{\infty\} \times S^2$.

To define our second theory, we start with the product of the closed string theory and the CS theory, $\mathcal{T}_{\rm cl} \otimes {\rm CS}_K^4$, supported on $\mathbb{R}_{w,x}^2 \times \mathbb{R}_+ \times S^3$ and compactify on $S^3$, our notation for this theory is the following:

$$\mathcal{T}_{\rm bk} := \pi_*^{S^3} \left( \mathcal{T}_{\rm cl} \otimes {\rm CS}_K^4 \right). \tag{35}$$

We can put the theory $\mathcal{T}_{\rm bd}$ (24) on the plane $\mathbb{R}_{w,x}^2$ at infinity of $\mathbb{R}_{w,x}^2 \times \mathbb{R}_+$. This plane has a distinguished line $\mathbb{R}_x \times \{\infty\}$ (18) where the D4-brane world volume intersects.[19] Along this line we have the $\mathfrak{gl}_K$ gauge field which couples to the fermions of the QM in $\mathcal{T}_{\rm bd}$ (this coupling corresponds to the last term in (23)). Boundary excitations from arbitrary points on $\mathbb{R}_{w,x} \times \{\infty\}$ will correspond to operators in the BF theory that are commuting, since these local excitations on a plane are not ordered. The non-commutative algebra we are interested in the BF theory is the algebra of gauge invariant operators restricted to a particular line. Similarly, in the "gravitational" side of the setup, we are interested in boundary excitations restricted to the line $\ell_\infty(z)$. Let us look a bit more closely at the coupling between the connection $A$ and the fermions:

$$I_z := \frac{1}{\hbar} \int_{\ell_\infty(z)} \overline{\psi}^i A_i^j \psi_j, \qquad \ell_\infty(z) = \mathbb{R}_x \times \{y = \infty\} \times \{z\}. \tag{36}$$

A small variation of $z$ leads to coupling between the fermions and $z$-derivatives of the connection:

$$I_{z+\delta z} = \sum_{n=0}^{\infty} \frac{1}{\hbar} \int_{\ell_\infty(z)} \frac{(\delta z)^n}{n!} \overline{\psi}^i \partial_z^n A_i^j \psi_j. \tag{37}$$

In the BF theory, the field B corresponds to the fluctuation of the D2-branes in the transverse $\mathbb{C}$ direction [18]. Therefore, we can interpret the above varied coupling term as saying that the operator in the boundary theory $\mathcal{T}_{\rm bd}$ that couples to the derivative $\partial_z^n A_i^j$ is precisely the operator $O_j^i[n] = \hbar^{-1} \overline{\psi}^i {\rm B}^n \psi_j$ (c.f. (25), (26)). This motivates us to look at functional derivatives of $\langle W_\varrho(L) \rangle_A$ with respect to $\partial_z^n A_i^j$ at fixed points along $\ell_\infty(z)$, such as:

$$\frac{\delta}{\delta \partial_z^{n_1} A_{i_1}^{j_1}(p_1)} \cdots \frac{\delta}{\delta \partial_z^{n_m} A_{i_m}^{j_m}(p_m)} \langle W_\varrho(L) \rangle_A, \qquad p_1, \cdots, p_m \in \ell_\infty(z). \tag{38}$$

Just as the expectation value $\langle W_\varrho(L) \rangle_A$ is ${\rm End}(V)$-valued, these functional derivatives are ${\rm End}(V)$-valued as well. The action is given by applying the functional derivative on $\langle W_\varrho(L) \rangle_A(\psi)$ for any $\psi \in V$. Let us denote this operator as

$$T_j^i[n] : \ell_\infty(z) \times V \to V,$$

$$p \in \ell_\infty(z), \qquad T_j^i[n](p) : \psi \mapsto \frac{\delta}{\delta \partial_z^n A_i^j(p)} \langle W_\varrho(L) \rangle_A(\psi), \tag{39}$$

---

[18]The boundary was chosen to respect the symmetry of the Wilson line along $L$.

[19]After aligning the $v$-coordinates of the plane and the D4-branes.

which can be pictorially represented by slight modifications of (34):

$$
\begin{array}{c}
y = 0,\ \psi \quad \overset{W_\varrho(L)}{\longrightarrow} \quad T^i_j[n](p)(\psi) \\[2ex]
\frac{\delta}{\delta\partial_z^n A^j_i} \\[1ex]
y = \infty \quad\quad\quad\quad\quad \times \\
x = -\infty \quad\quad x = p \quad\quad x = +\infty
\end{array}
\tag{40}
$$

Composition of these operators, such as $T^{i_1}_{j_1}(p_1)\cdots T^{i_m}_{j_m}(p_m)$, is defined by the expression (38). A more precise and computable characterization of these operators and their composition in terms of Witten diagrams [3] will be given in §5 (see (122)). Due to topological invariance along the $x$-direction, the operator $T^i_j[n](p)$ must be independent of the position $p$. However, since these operators are positioned along a line, their product should be expected to depend on the ordering, leading to a non-commutative associative algebra. We can now define the second algebra to appear in our example of holography:

$$
\mathcal{A}^{\mathrm{Sc}}(\mathcal{T}_{\mathrm{bk}}) := \left( T^i_j[n] \right),
\tag{41}
$$

i.e., the complex algebra generated by the set $\{T^i_j[n]\}$.

*Remark* 2 (Center of the algebra). In the BF theory we mentioned gauge invariant operators that belong to the center of the algebra. Clearly, the holographic dual of those operators do not come from the CS theory, rather they come from the closed string theory. A 2-form field $\phi = \phi_{wx}\mathrm{d}w \wedge \mathrm{d}x + \cdots$ from the closed string theory deforms the BF theory as:

$$
S_{\mathrm{BF}} \to S_{\mathrm{BF}} + \int_{\mathbb{R}^2_{w,x}} \mathrm{d}w \wedge \mathrm{d}x \left( \partial_z^n \phi_{wx} \right) \mathrm{tr}_{\mathbf{N}} \left( \mathrm{B}^n \right).
\tag{42}
$$

Functional derivatives with respect to the fields $\partial_z^n \phi_{w,x}$ placed at arbitrary locations on the asymptotic boundary $\mathbb{R}^2_{w,x} \times \{\infty\}$ correspond to inserting the operators $\mathrm{tr}_{\mathbf{N}}\mathrm{B}^n$ in the BF theory.[20] As we did in the BF theory, we are going to ignore these operators now as well. △

After all this setup, we can present the main result of this paper:

**Theorem 1.** *In the limit $N \to \infty$, both the algebra of local operators (26) along the line operator in the theory $\mathcal{T}_{\mathrm{bd}} = \mathrm{BF}_N \otimes_N \mathrm{QM}_{N\times K}$, and the algebra of scatterings from a line in the boundary (41) of the theory $\mathcal{T}_{\mathrm{bk}} = \pi^{S^3}_* \left( \mathcal{T}_{\mathrm{cl}} \otimes \mathrm{CS}^4_K \right)$ are isomorphic to the Yangian of $\mathfrak{gl}_K$, i.e.:*

$$
\mathcal{A}^{\mathrm{Op}}(\mathcal{T}_{\mathrm{bd}}) \overset{N\to\infty}{\cong} Y_\hbar(\mathfrak{gl}_K) \overset{N\to\infty}{\cong} \mathcal{A}^{\mathrm{Sc}}(\mathcal{T}_{\mathrm{bk}}).
\tag{43}
$$

The rest of the paper is devoted to the explicit computations of these algebras.

# 4 $\mathcal{A}^{\mathrm{Op}}(\mathcal{T}_{\mathrm{bd}})$ from BF $\otimes$ QM theory

In this section we prove the first half of our main result (Theorem 1):

**Proposition 1.** *The algebra $\mathcal{A}^{\mathrm{Op}}(\mathcal{T}_{\mathrm{bd}})$, defined in the context of 2d BF theory with the gauge group $\mathrm{GL}_N$ coupled to a 1d fermionic quantum mechanics with global symmetry $\mathrm{GL}_N \times \mathrm{GL}_K$, is isomorphic to the Yangian of $\mathfrak{gl}_K$ in the limit $N \to \infty$:*

$$
\mathcal{A}^{\mathrm{Op}}(\mathcal{T}_{\mathrm{bd}}) \overset{N\to\infty}{\cong} Y_\hbar(\mathfrak{gl}_K).
\tag{44}
$$

---

[20]These functional derivatives are represented by the red dot on the asymptotic boundary in figure 1.

The BF theory coupled to a fermionic quantum mechanics was defined in §3.3, let us repeat the actions here:

$$S_{\mathcal{T}_{bd}} = S_{BF} + S_{QM}, \tag{45}$$

where:

$$S_{BF} = \int_{\mathbb{R}^2_{w,x}} \mathrm{tr}_\mathbf{N}(BdA + B[A,A]) \tag{46}$$

$$\text{and} \quad S_{QM} = \int_{\mathbb{R}_x} \left( \overline{\psi}_i d\psi^i + \overline{\psi}_i A \psi^i \right). \tag{47}$$

We no longer need the source term, i.e., the coupling to the background $\mathfrak{gl}_K$ connection (c.f. (23)). Let us determine the propagators now.

The BF propagator is defined as the 2-point correlation function:

$$\mathsf{P}^{\alpha\beta}(p,q) := \left\langle B^\alpha(p) A^\beta(q) \right\rangle. \tag{48}$$

We choose a basis $\{\tau_\alpha\}$ of $\mathfrak{gl}_N$ which is orthonormal with respect to the trace $\mathrm{tr}_\mathbf{N}$:

$$\mathrm{tr}_\mathbf{N}(\tau_\alpha \tau_\beta) = \delta_{\alpha\beta}. \tag{49}$$

Then the two point correlation function becomes diagonal in the color indices:

$$\mathsf{P}^{\alpha\beta}(p,q) \equiv \delta^{\alpha\beta} \mathsf{P}(p,q). \tag{50}$$

We shall often refer to just $\mathsf{P}$ as the propagator, it is determined by the following equation:[21]

$$\frac{1}{\hbar} d\mathsf{P}(0,p) = \delta^2(p) dw \wedge dx. \tag{51}$$

Once we impose the following gauge fixing condition, analogous to the Lorentz gauge:

$$d \star \mathsf{P}(0,p) = 0, \tag{52}$$

the solution is (using translation invariance to replace the 0 with an arbitrary point):

$$\mathsf{P}(p,q) = \frac{\hbar}{2\pi} d\phi(p,q), \tag{53}$$

where $\phi(p,q)$ is the angle (measured counter-clockwise) between the line joining $p$-$q$ and any other reference line passing through $p$. In Feynman diagrams we shall represent this propagator as:

$$\mathsf{P}(p,q) = \ p \longrightarrow q \ . \tag{54}$$

Similarly, the propagator in the QM is defined by:

$$\frac{1}{\hbar} \partial_{x_2} \left\langle \overline{\psi}^a_i(x_1) \psi^j_b(x_2) \right\rangle = \delta^a_b \delta^j_i \delta^1(x_1 - x_2), \tag{55}$$

with the solution:

$$\left\langle \overline{\psi}^a_i(x_1) \psi^j_b(x_2) \right\rangle = \delta^a_b \delta^j_i \hbar \vartheta(x_2 - x_1), \tag{56}$$

where $\vartheta(x_2 - x_1)$ is a unit step function. Anti-symmetry of the fermion fields dictates:

$$\left\langle \psi^j_b(x_1) \overline{\psi}^a_i(x_2) \right\rangle = - \left\langle \overline{\psi}^a_i(x_2) \psi^j_b(x_1) \right\rangle = -\delta^a_b \delta^j_i \hbar \vartheta(x_1 - x_2). \tag{57}$$

---

[21]A minor technicality: $\mathsf{P}(p,q)$ is a 1-form on $\mathbb{R}^2_p \times \mathbb{R}^2_q$ and in (51), by $\mathsf{P}(0,p)$ we mean the pull-back of $\mathsf{P} \in \Omega^2(\mathbb{R}^4)$ by the diagonal embedding $\mathbb{R}^2 \hookrightarrow \mathbb{R}^2 \times \mathbb{R}^2$.

We take the step function to be:

$$\vartheta(x) = \frac{1}{2}\text{sgn}(x) = \begin{cases} 1/2 & \text{for } x > 0 \\ 0 & \text{for } x = 0 \\ -1/2 & \text{for } x < 0 \end{cases}. \tag{58}$$

Then we can write:

$$\left\langle \overline{\psi}_i^a(x_1)\psi_b^j(x_2) \right\rangle = \left\langle \psi_b^j(x_1)\overline{\psi}_i^a(x_2) \right\rangle = \delta_b^a \delta_i^j \frac{\hbar}{2}\text{sgn}(x_2 - x_1). \tag{59}$$

This propagator does not distinguish between $\psi$ and $\overline{\psi}$ and it depends only on the order of the fields, not their specific positions. In Feynman diagrams we shall represent this propagator as:

$$\frac{\hbar}{2}\text{sgn}(x_2 - x_1) = \underset{x_1 \quad x_2}{\overset{\frown}{\bullet \quad \bullet}}, \tag{60}$$

where the curved line refers to the propagator itself and the horizontal line refers to the support of the QM, i.e., the line $w = 0$. We now move on to computing operator products that will give us the algebra $\mathcal{A}^{\text{Op}}(\mathcal{T}_{\text{bd}})$.

*Remark* 3 (Fermion vs. Boson - Propagator). We might as well have considered a bosonic QM instead of a fermionic QM. At present, this is an arbitrary choice, however, if one starts from some brane setup in physical string theory and reduce it to the topological setup we are considering by twists and $\Omega$-deformations,[22] then depending on the starting setup one might end up with either statistics. Let us make a few comments about the bosonic case. In the first order formulation of bosonic QM the action looks exactly as in the fermionic action 47 except the fields would be commuting – let us denote the bosonic counterpart of $\overline{\psi}$ and $\psi$ by $\overline{\phi}$ and $\phi$ respectively. Then, instead of the propagator (59), we would have the following propagator:[23]

$$-\left\langle \overline{\phi}_i^a(x_1)\phi_b^j(x_2) \right\rangle = \left\langle \phi_b^j(x_1)\overline{\phi}_i^a(x_2) \right\rangle = \delta_b^a \delta_i^j \frac{\hbar}{2}\text{sgn}(x_2 - x_1). \tag{61}$$

Note that the extra sign in the first term (compared to (59)) is consistent with the commutativity of the bosonic fields:

$$\left\langle \overline{\phi}_i^a(x_1)\phi_b^j(x_2) \right\rangle = \left\langle \phi_b^j(x_2)\overline{\phi}_i^a(x_1) \right\rangle. \tag{62}$$

The bosonic propagator (61) distinguishes between $\phi$ and $\overline{\phi}$, in that, the propagator is positive if $\phi(x_1)$ is placed before $\overline{\phi}(x_2)$, i.e., $x_1 < x_2$, and negative otherwise. $\triangle$

## 4.1 Free theory limit, $\mathcal{O}(\hbar^0)$

Interaction in the quantum mechanics is generated via coupling to the $\mathfrak{gl}_N$ gauge field (see (47)). Without this coupling, the quantum mechanics is free. In this section we compute the operator product between $O_j^i[m]$ and $O_l^k[n]$ in this free theory, which will give us the classical algebra.

Let us denote the operator product by $\star$, as in:

$$O_j^i[m] \star O_l^k[n]. \tag{63}$$

---

[22] We describe one such specific procedure in §6.

[23] We have chosen the overall sign of the propagator to make comparision between Feynman diagrams involving bosonic operators and fermionic operators as simple as possible. However, the overall sign is not important for the determination of the algebra. The parameter $\hbar$ enters the algebra as the formal variable deforming the universal enveloping algebra $U(\mathfrak{gl}_K[z])$ to its Yangian, and the sign of $\hbar$ is irrelevant for this purpose.

The classical limit of this product has an expansion in Feynman diagrams where we ignore all diagrams with BF propagators. Before evaluating this product let us illustrate the computations of the relevant diagrams by computing one exemplary diagram in detail.

Consider the following diagram:[24]

$$G_{jl}^{ik}[\triangle \cdot \blacktriangle](x_1, x_2) := \quad\underbrace{\bullet\bullet\bullet}_{\substack{x_1 \\ O_j^i[m]}}\overset{\frown}{\phantom{xx}}\underbrace{\bullet\bullet\bullet}_{\substack{x_2 \\ O_l^k[n]}} \tag{64}$$

We are representing the operator $O_j^i[m] = \frac{1}{\hbar}\overline{\psi}_j^a(B^m)_a^b\psi_b^i$ by the symbol $\overset{\frown}{\bullet\bullet\bullet}$ where the three dots represent the three fields $\overline{\psi}_j^a$, $(B^m)_a^b$, and $\psi_b^i$ respectively. The coordinate below an operator in (64) represents the position of that operator and the lines connecting different dots are propagators. Depending on which dots are being connected a propagator is either the BF propagator (53) or the QM propagator (59). The value of the diagram is then given by:

$$\begin{aligned} G_{jl}^{ik}[\triangle \cdot \blacktriangle](x_1, x_2) &= \frac{1}{\hbar}\overline{\psi}_j^a(x_1)(B(x_1)^m)_a^b\frac{1}{2}\hbar\delta_b^c\delta_l^i\frac{1}{\hbar}(B(x_2)^n)_c^d\psi_d^k(x_2), \\ &= \frac{1}{2\hbar}\delta_l^i\overline{\psi}_j(x_1)B(x_1)^mB(x_2)^n\psi^k(x_2). \end{aligned} \tag{65}$$

In the second line we have hidden away the contracted $\mathfrak{gl}_N$ indices. In computing the operator product (63) only the following limit of the diagram is relevant:

$$\lim_{x_2 \to x_1} G_{jl}^{ik}[\triangle \cdot \blacktriangle](x_1, x_2) = \frac{1}{2\hbar}\delta_l^i\overline{\psi}_j B^{m+n}\psi^k = \frac{1}{2}\delta_l^i O_j^k[m+n]. \tag{66}$$

We have ignored the positions of the operators, because the algebra we are computing must be translation invariant. Reference to position only matters when we have different operators located at different positions.

We can now give a diagramatic expansion of the operator product (63) in the free theory:

$$O_j^i[m] \star O_l^k[n] \overset{x_2 \to x_1}{=} \quad\underbrace{\bullet\bullet\bullet}_{x_1}\overset{\phantom{x}}{\phantom{xx}}\underbrace{\bullet\bullet\bullet}_{x_2} \; + \; \underbrace{\bullet\bullet\bullet}_{x_1}\overset{\frown}{\phantom{xx}}\underbrace{\bullet\bullet\bullet}_{x_2}$$

$$+ \; \underbrace{\bullet\bullet\bullet}_{x_1}\overset{\frown}{\phantom{xx}}\underbrace{\bullet\bullet\bullet}_{x_2} \; + \; \underbrace{\bullet\bullet\bullet}_{x_1}\overset{\frown\frown}{\phantom{xx}}\underbrace{\bullet\bullet\bullet}_{x_2}. \tag{67}$$

We have omitted the labels for the operators in the diagrams. It is understood that the first operator is $O_j^i[m]$ and the second one is $O_l^k[n]$. Summing these four diagrams we find:

$$O_j^i[m] \star O_l^k[n] = O_j^i[m]O_l^k[n] + \frac{1}{2}\delta_l^i O_j^k[m+n] - \frac{1}{2}\delta_j^k O_l^i[m+n] + \frac{1}{4}\delta_l^i\delta_j^k \mathrm{tr_N}B^{m+n}. \tag{68}$$

The product in the first term on the right hand side of the above equation is a c-number product, hence commuting. The sign of the third term comes from the first diagram in the second line in (67). In short, this comes about by commuting two fermions, as follows:

$$\lim_{x_2 \to x_1} G_{jl}^{ik}[\blacktriangle \cdot \triangle](x_1, x_2) = \frac{1}{2\hbar}\delta_j^k\psi^i B^{m+n}\overline{\psi}_l = -\frac{1}{2\hbar}\delta_j^k\overline{\psi}_l B^{m+n}\psi^i = -\frac{1}{2}\delta_j^k O_l^i[m+n]. \tag{69}$$

---

[24]The reader can ignore the elaborate symbols (triangles and as such) that we use to refer to a diagram. They are meant to systematically identify a diagram, but for practical purposes the entire expression can be thought of as an unfortunately long unique symbol assigned to a diagram, just to refer to it later on.

Using (68) we can compute the Lie bracket of the algebra $\mathcal{A}^{\mathrm{Op}}(\mathcal{T}_{\mathrm{bd}})$ in the classical limit:

$$\left[O_j^i[m], O_l^k[n]\right]_\star = \delta_l^i O_j^k[m+n] - \delta_j^k O_l^i[m+n]. \tag{70}$$

This is the Lie bracket in the loop algebra $\mathfrak{gl}_K[z]$.[25]

*Remark* 4 (Fermion vs. Boson - Classical Algebra). How would the bracket (70) be affected if we had a bosonic QM? It would not. The first and the fourth diagrams from (67) would still cancel with their counterparts when we take the commutator. The value of the second diagram, (66), remains unchanged. In computing the value of the third diagram (see (69)) we get an extra sign compared to the fermionic case because we don't pick up any sign by commuting bosonos, however, we pick up yet another sign from the propagator relative to the fermionic propagator (see Remark 3 – compare the bosonic (61) and fermionic (59) propagators).

## 4.2   Loop corrections from BF theory

Interaction in the BF theory comes from the following term in the BF action (46):

$$f_{\alpha\beta\gamma} \int_{\mathbb{R}^2} \mathsf{B}^\alpha \mathsf{A}^\beta \wedge \mathsf{A}^\gamma, \tag{72}$$

where the structure constant $f_{\alpha\beta\gamma}$ comes from the trace in our orthonormal basis (49):

$$f_{\alpha\beta\gamma} = \mathrm{tr}_{\mathbf{N}}(\tau_\alpha[\tau_\beta, \tau_\gamma]). \tag{73}$$

In Feynman diagrams this interaction will be represented by a trivalent vertex with exactly 1 outgoing and 2 incoming edges. Including the propagators for the edges, such a vertex will look like:

$$
\begin{aligned}
&= \frac{\hbar^2}{(2\pi)^3} f^{\alpha\beta\gamma} \int_{p \in \mathbb{R}^2} \mathrm{d}_{q_1}\phi(p, q_1) \wedge \mathrm{d}_{q_2}\phi(p, q_2) \wedge \mathrm{d}_{q_3}\phi(p, q_3), \\
&=: V^{\alpha\beta\gamma}(q_1, q_2, q_3).
\end{aligned}
\tag{74}
$$

We have given the name $V^{\alpha\beta\gamma}$ to this vertex function.

Possibilities of Feynman diagrams are rather limited in the BF theory. In particular, there are no cycles.[26] This means that there is only one possible BF diagram that will appear in our computations, which is the following:

$$. \tag{75}$$

---

[25]The isomorphism is given by: $O_j^i[m] \mapsto z^m \mathrm{e}_i^j$, where $\mathrm{e}_i^j$ are the elementary matrices of dimension $K \times K$ satisfying the relation:

$$[\mathrm{e}_i^j, \mathrm{e}_k^l] = \delta_i^l \mathrm{e}_k^j - \delta_k^j \mathrm{e}_i^l. \tag{71}$$

[26]By cycle we mean loop in the sense of graph theory. In this paper when we write loop without any explanation, we mean the exponent of $\hbar$, as is customary in physics. This exponent is related but not always equal to the number of loops (graph theory). Therefore, we reserve the word loop for the exponent of $\hbar$, and the word cycle for what would be loop in graph theory.

Let us illustrate why there are no cycles in BF Feynman diagrams. Consider the cycle ⬡. The three propagators in the cycle contribute the 3-form $\mathrm{d}\phi_1 \wedge \mathrm{d}\phi_2 \wedge \mathrm{d}\phi_3$ to a diagram containing the cycle, where the $\phi$'s are the angles between two successive vertices. However, due to the constraint $\phi_1 + \phi_2 + \phi_3 = 2\pi$, only two out of the three propagators are linearly independent. Therefore, their product vanishes.

The middle operator looks slightly different because this operator involves the connection A and an integration, as opposed to just the B field, to be specific,

$$\bullet\!\!\!\circ\!\!\!\bullet = \frac{1}{\hbar}\int_{\mathbb{R}}\overline{\psi}_i A\psi^i. \tag{76}$$

This term is the result of the insertion of the term coupling the fermions to the $\mathfrak{gl}_N$ connection in the QM action (47). In doing the above integrationover $\mathbb{R}$ we shall take $\overline{\psi}$ and $\psi$ to be constant. In other words, we are taking derivatives of the fermions to be zero. The reason is that, the equations of motion for the fermions (derived from the action (47)), namely $d\psi^i = -A\psi^i$ and $d\overline{\psi}_i = A\overline{\psi}_i$, tell us that derivatives of the fermions are not gauge-invariant quantities – and we want to expand the operator product of gauge invariant operators in terms of other gauge invariant operators only.[27]

In the following we shall consider the diagram (75) with all possible fermionic propagators added to it.

### 4.2.1   0 **fermionic propagators**

We are mostly going to compute products of level 1 operators, i.e., $O_j^i[1]$, this is because together with the level 0 operators, they generate the entire algebra. Without any fermionic propagators, we just have the diagram (75):

$$G_{jl}^{ik}[\cdot\cdot](x_1, x_2) := \quad\text{<image diagram>}\quad. \tag{77}$$

$$\begin{array}{ccc} x_1 & x & x_2 \\ O_j^i[1] & \frac{1}{\hbar}\int\overline{\psi}A\psi & O_l^k[1] \end{array}$$

In future, we shall omit the labels below the operators to reduce clutter. In terms of the BF vertex function (74), the above diagram can be expressed as:

$$G_{jl}^{ik}[\cdot\cdot](x_1, x_2) = \frac{1}{\hbar^3}\overline{\psi}_j\tau_\alpha\psi^i\overline{\psi}\tau_\beta\psi\overline{\psi}_l\tau_\gamma\psi^k\int_{\mathbb{R}_x}V^{\alpha\beta\gamma}(x_1, x, x_2). \tag{78}$$

We have used the expansions of $B = B^\alpha\tau_\alpha$ and $A = A^\beta\tau_\beta$ in the orthonormal $\mathfrak{gl}_N$ basis $\{\tau_\alpha\}$. As defined in (74), the vertex function $V^{\alpha\beta\gamma}$ is a 2d integral of a 3-form, therefore, the integration of the vertex function on a line gives us a number. It will be convenient to divide up the integral of the vertex function into three integrals depending on the location of the point $x$ relative to $x_1$ and $x_2$:

$$\int_{\mathbb{R}_x}V^{\alpha\beta\gamma}(x_1, x, x_2) = \mathcal{V}_{\cdot||}^{\alpha\beta\gamma}(x_1, x_2) + \mathcal{V}_{|\cdot|}^{\alpha\beta\gamma}(x_1, x_2) + \mathcal{V}_{||\cdot}^{\alpha\beta\gamma}(x_1, x_2), \tag{79}$$

where,

$$\mathcal{V}_{\cdot||}^{\alpha\beta\gamma}(x_1, x_2) := \int_{x<x_1}V^{\alpha\beta\gamma}(x_1, x, x_2) = \frac{\hbar^2}{24}f^{\alpha\beta\gamma}, \tag{80a}$$

$$\mathcal{V}_{|\cdot|}^{\alpha\beta\gamma}(x_1, x_2) := \int_{x_1<x<x_2}V^{\alpha\beta\gamma}(x_1, x, x_2) = \frac{\hbar^2}{24}f^{\alpha\beta\gamma}, \tag{80b}$$

$$\mathcal{V}_{||\cdot}^{\alpha\beta\gamma}(x_1, x_2) := \int_{x_2<x}V^{\alpha\beta\gamma}(x_1, x, x_2) = \frac{\hbar^2}{24}f^{\alpha\beta\gamma}. \tag{80c}$$

---

[27]An alternative, and perhaps more streamlined, way to say this would be to formulate all the theories in the BV/BRST formalism, where operators are defined, a priori, to be in the cohomology of the BRST operator, which would exclude derivatives of the fermions to begin with.

We evaluate these integrals in Appendix §A. Adding them up and substituting in (78) we get from the diagram (77):

$$G_{jl}^{ik}[\cdot\cdot](x_1,x_2) \overset{x_1 \to x_2}{=} \frac{1}{8\hbar}\overline{\psi}_j\tau_\alpha\psi^i\overline{\psi}\tau_\beta\psi\overline{\psi}_l\tau_\gamma\psi^k f^{\alpha\beta\gamma}. \tag{81}$$

Since the $\mathfrak{gl}_N$ indices are all contracted, we can choose a particular basis to get an expression independent of any reference to $\mathfrak{gl}_N$. Choosing the elementary matrices as the basis we get the following expression:

$$G_{jl}^{ik}[\cdot\cdot] = \frac{\pi^2}{2\hbar}\overline{\psi}_j e_b^a\psi^i\overline{\psi}e_d^c\psi\overline{\psi}_l e_f^e\psi^k f_{ace}^{bdf}. \tag{82}$$

Using the definition of the elementary matrices $\left(e_b^a\right)_d^c = \delta_d^a\delta_b^c$ we get $\overline{\psi}_j e_b^a\psi^i = \overline{\psi}_j^d\left(e_b^a\right)_d^c\psi_c^i = \overline{\psi}_j^a\psi_b^i$ and in this basis the structure constant is:

$$f_{ace}^{bdf} = \delta_a^d\delta_c^f\delta_e^b - \delta_c^b\delta_e^d\delta_a^f. \tag{83}$$

Using these expressions in (82) we get:

$$\begin{aligned}
G_{jl}^{ik}[\cdot\cdot] &= \frac{1}{8\hbar}\left(\overline{\psi}_j\psi^m\overline{\psi}_m\psi^k\overline{\psi}_l\psi^i - \overline{\psi}_l\psi^m\overline{\psi}_m\psi^i\overline{\psi}_j\psi^k\right), \\
&= \frac{1}{8}\hbar^2\left(O_j^m[0]O_m^k[0]O_l^i[0] - O_l^m[0]O_m^i[0]O_j^k[0]\right).
\end{aligned} \tag{84}$$

The above expression is anti-symmetric under the exchange $(i,j) \leftrightarrow (k,l)$, therefore, the contribution of this diagram to the Lie bracket (70) is twice the value of the diagram.

### 4.2.2  1 **fermionic propagator**

We have the following six diagrams:

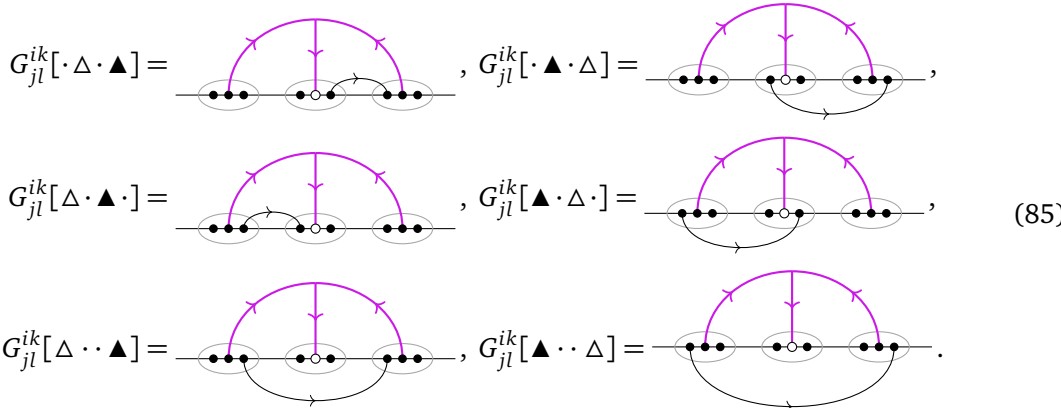

$$\tag{85}$$

In all the above diagrams, the left and the right most operators are $O_j^i[1]$ and $O_l^k[1]$ respectively, and all the graphs are functions of $x_1$ and $x_2$, where these two operators are located. Let us explain the evaluation of the top left diagram in detail. Written explicitly, this diagram is:

$$\begin{aligned}
G_{jl}^{ik}[\cdot\triangle\cdot\blacktriangle](x_1,x_2) = \frac{1}{\hbar^3}\int_{\mathbb{R}_x}\overline{\psi}_j(x_1)\tau_\alpha\psi^i(x_1)\overline{\psi}_m^a(x)\left(\tau_\beta\right)_a^b\left\langle\psi_b^m(x)\overline{\psi}_l^c(x_2)\right\rangle \\
\times\left(\tau_\gamma\right)_c^d\psi_d^k(x_2)V^{\alpha\beta\gamma}(x_1,x,x_2),
\end{aligned} \tag{86}$$

where the two point correlation function is the QM propagator (59). The integrand above depends on the position only to the extend that they depend on the ordering of the positions, since we are only quantizing the constant modes of the fermions.[28] The propagator between the two fermions gives a propagator which depends on the sign of $x_2 - x$ (see (59), (60)), since we are integrating over $x$, this propagator will change sign depending on whether $x$ is to the left or to the right of $x_2$.[29] Therefore, we can write this graph as:

$$G_{jl}^{ik}[\cdot \triangle \cdot \blacktriangle] = \frac{1}{\hbar^2}\overline{\psi}_j \tau_\alpha \psi^i \overline{\psi}_l \tau_\beta \tau_\gamma \psi^k \left( \mathcal{V}_{\cdot||}^{\alpha\beta\gamma} + \mathcal{V}_{|\cdot|}^{\alpha\beta\gamma} - \mathcal{V}_{||\cdot}^{\alpha\beta\gamma} \right),$$

$$= \frac{1}{24}\overline{\psi}_j \tau_\alpha \psi^i \overline{\psi}_l \tau_\beta \tau_\gamma \psi^k f^{\alpha\beta\gamma} = \frac{1}{24}\overline{\psi}_j \tau_\alpha \psi^i \overline{\psi}_l \tau_\delta \psi^k f_{\beta\gamma}{}^\delta f^{\alpha\beta\gamma}. \tag{87}$$

Due to the symmetry $f_{\beta\gamma}{}^\delta f^{\alpha\beta\gamma} = f_{\beta\gamma}{}^\alpha f^{\delta\beta\gamma}$, the above expression is symmetric under the exchange $(i,j) \leftrightarrow (k,l)$, therefore this diagram does not contribute to the Lie bracket (70). The diagrams $G_{jl}^{ik}[\cdot \blacktriangle \cdot \triangle]$, $G_{jl}^{ik}[\triangle \cdot \blacktriangle \cdot]$, and $G_{jl}^{ik}[\blacktriangle \cdot \triangle \cdot]$ do not contribute to the Lie bracket for exactly the same reason. The remaining two diagrams evaluate to the following expressions:

$$G_{jl}^{ik}[\triangle \cdot\cdot \blacktriangle] = \frac{1}{8\hbar} f^{\alpha\beta\gamma} \delta_l^i \overline{\psi}_j \tau_\alpha \tau_\gamma \psi^k \overline{\psi} \tau_\beta \psi, \tag{88a}$$

$$G_{jl}^{ik}[\blacktriangle \cdot\cdot \triangle] = -\frac{1}{8\hbar} f^{\alpha\beta\gamma} \delta_j^k \overline{\psi}_l \tau_\gamma \tau_\alpha \psi^i \overline{\psi} \tau_\beta \psi. \tag{88b}$$

Their sum is symmetric under the exchange $(i,j) \leftrightarrow (k,l)$,[30] and therefore these diagrams do not contribute to the Lie bracket either.

None of the diagrams with one fermionic propagator contributes to the Lie bracket.

### 4.2.3 2 **fermionic propagators**

There are nine ways to join two pairs of fermions with propagators:

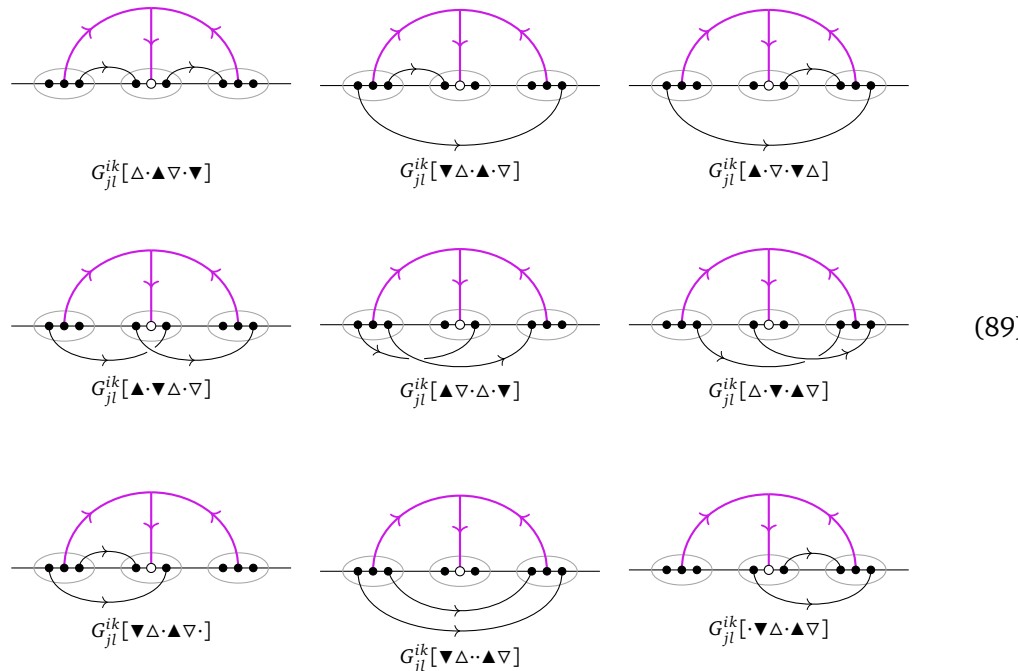

$$(89)$$

[29]This is the reason why we computed the integrals (80) separately depending on the position of $x$.

[30]The opposite ordering of $\tau_\alpha$ and $\tau_\gamma$ cancels the sign, using the anti-symmetry of the indices on the structure constant.

The left and the right most operators in all of the above diagrams are $O_j^i[1]$ and $O_l^k[1]$ respectively.

All three of the diagrams in the bottom line vanish. This is because joining all the fermions in two operators with propagators introduces a trace $\text{tr}_{\mathbf{N}}(\tau_\alpha \tau_\beta)$ of $\mathfrak{gl}_N$ generators when the same color indices, $\alpha$ and $\beta$ in this case, are contracted with the structure constant coming from the BF interaction vertex, as in $\text{tr}_{\mathbf{N}}(\tau_\alpha \tau_\beta) f^{\alpha\beta\gamma}$. Since the trace is symmetric and the structure constant is anti-symmetric, these three diagrams vanish.

Computation also reveals the following relations among the four diagrams at the top right $2 \times 2$ corner of (89):

$$G_{jl}^{ik}[\blacktriangledown \vartriangle \cdot \blacktriangle \triangledown] = G_{jl}^{ik}[\blacktriangle \cdot \triangledown \cdot \blacktriangledown \vartriangle], \quad G_{jl}^{ik}[\blacktriangle \triangledown \cdot \vartriangle \cdot \blacktriangledown] = G_{jl}^{ik}[\vartriangle \cdot \blacktriangledown \cdot \blacktriangle \triangledown], \tag{90}$$

together with the fact that $G_{jl}^{ik}[\blacktriangledown \vartriangle \cdot \blacktriangle \cdot \triangledown] + G_{jl}^{ik}[\blacktriangle \triangledown \cdot \vartriangle \cdot \blacktriangledown]$ is symmetric under the exchange $(i, j) \leftrightarrow (k, l)$. The above relations and symmetry implies that when anti-symmetrized with respect to $(i, j) \leftrightarrow (k, l)$, the sum of the four diagrams appearing in the above relations vanish. In a similar vein, the sum $G_{jl}^{ik}[\vartriangle \cdot \blacktriangle \triangledown \cdot \blacktriangledown] + G_{jl}^{ik}[\blacktriangle \cdot \blacktriangledown \vartriangle \cdot \triangledown]$ also turns out to be symmetric under $(i, j) \leftrightarrow (k, l)$ and therefore these two diagrams do not contribute to the Lie bracket either.

None of the diagrams with two fermionic propagators contributes to the Lie bracket.

### 4.2.4  3 **fermionic propagators**

There are two ways to join all the fermions with propagators:

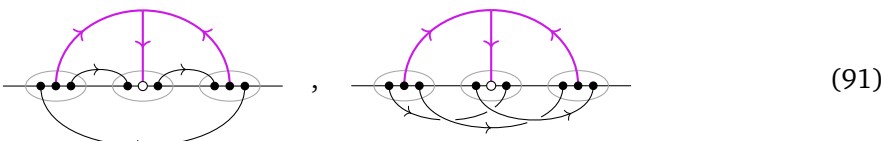

$$\tag{91}$$

As before, the left and the right most operators are $O_j^i[1]$ and $O_l^k[1]$ respectively. Both of these diagrams are proportional to $\delta_l^i \delta_j^k$, in particular, they are symmetric under the exchange $(i, j) \leftrightarrow (k, l)$, and therefore do not contribute to the Lie bracket.

### 4.2.5  **Lie bracket**

Since only the diagram with zero fermionic propagator (84) survives the anti-symmetrization, the Lie bracket (70) up to $\mathcal{O}(\hbar^2)$ corrections becomes:

$$\begin{aligned}
\left[ O_j^i[1], O_l^k[1] \right]_\star &= \delta_l^i O_j^k[2] - \delta_j^k O_l^i[2] + G_{jl}^{ik}[\cdot\cdot] - G_{lj}^{ki}[\cdot\cdot], \\
&= \delta_l^i O_j^k[2] - \delta_j^k O_l^i[2] + \frac{\hbar^2}{4} \left( O_j^m[0] O_m^k[0] O_l^i[0] - O_l^m[0] O_m^i[0] O_j^k[0] \right).
\end{aligned} \tag{92}$$

Though we have only computed up to 2-loops diagrams, this result is exact, because there are no more non-vanishing Feynman diagrams that can be drawn.

Since (92) is not among the standard relations of the Yangian that are readily available in the literature, we shall now make a change of basis to get to a standard relation. First note that, the product of operators in the right hand side of the above equation is not the operator product, this product is commutative (anti-commutative for fermions) and therefore we can write it in an explicitly symmetric form, such as:

$$O_j^m[0] O_m^k[0] O_l^i[0] = \left\{ O_j^m[0], O_m^k[0], O_l^i[0] \right\}, \tag{93}$$

where the bracket means complete symmetriazation, i.e., for any three symbols $O_1, O_2$ and $O_3$ with a product we have:

$$\{O_1, O_2, O_3\} = \frac{1}{3!} \sum_{s \in S_3} O_{s(1)} O_{s(2)} O_{s(3)}, \tag{94}$$

where $S_3$ is the symmetric group of order 3!. With this symmetric bracket, let us now define:

$$Q_{jl}^{ik} := f_{jvm}^{iun} f_{uor}^{vpq} f_{qsl}^{rtk} \left\{ O_n^m[0], O_p^o[0], O_t^s[0] \right\}, \tag{95}$$

where $f_{lmn}^{ijk}$ are the $\mathfrak{gl}_K$ structure constants in the basis of elementary matrices. Using the form of the $\mathfrak{gl}$ structure constant in the basis of elementary matrices (c.f. (83)) we can write:

$$Q_{jl}^{ik} = 3 \left\{ O_l^i, O_j^m, O_m^k \right\} - 3 \left\{ O_j^k, O_l^m, O_m^i \right\} + \delta_j^k \left\{ O_l^m, O_m^n, O_n^i \right\} - \delta_l^i \left\{ O_j^m, O_m^n, O_n^k \right\}. \tag{96}$$

We have ignored to write the [0] for each of the operators. Using the above expression we can re-write (92) as:

$$\left[ O_j^i[1], O_l^k[1] \right]_\star = \delta_l^i \widetilde{O}_j^k[2] - \delta_j^k \widetilde{O}_l^i[2] + \frac{\hbar^2}{12} Q_{jl}^{ik}, \tag{97}$$

with the redefinition:

$$\widetilde{O}_j^k[2] := O_j^k[2] - \frac{\hbar^2}{12} \left\{ O_j^m, O_m^n, O_n^k \right\}. \tag{98}$$

Note that, $\left\{ O_j^m, O_m^n, O_n^k \right\}$ does indeed transform as an element of $\mathfrak{gl}_K$, since it only has a pair of fundamental-anti-fundamental $\mathfrak{gl}_K$ indices free. This makes the redefinition of $O_j^k[2]$ possible. The Lie bracket (97) is how the Yangian was presented in [19].

*Remark* 5 (Fermion vs. Boson - Quantum Algebra). In Remark 4 we pointed out that the classical part of the algebra (97) remains unchanged if we replace the fermionic QM on the defect with a bosonic QM. This remains true at the quantum level – though a bit tedious, it can be readily verified by using the bosonic propagator (61) and keeping track of signs through the computations of this section without any other modifications. $\triangle$

## 4.3 Large $N$ limit: The Yangian

In (97) we have already found a defining relation for the Yangian, and this relation holds at finite $N$. However, for finite $N$ there can be extra relations among the operators $O_j^i[m]$. For example, for finite $N$, B is a finite dimensional (namely $N \times N$) matrix and therefore $B^N$ can be written as a linear combination of $B^m$ with $m < N$. This can lead to relations among the operators $O_j^i[m]$. To find all the relations precisely one must quantize the BF theory taking the direction of the 1d defect to be time and establish the relations among the operators $O_j^i[m]$ on the resulting Hilbert space. Any such relation reduces the operator algebra to a quotient of the Yangian. This will be done in a future work, for now we note that we can avoid these incidental relations by taking the large $N$ limit where all matrices are infinite dimensional. In this limit we therefore have the standard Yangian, as opposed to some quotient of it. This concludes our proof for Proposition 1.

## 5 $\mathcal{A}^{\mathrm{Sc}}(\mathcal{T}_{\mathrm{bk}})$ from 4d Chern-Simons Theory

In this section we prove the second half of our main result (Theorem 1):

**Proposition 2.** *The algebra $\mathcal{A}^{\mathrm{Sc}}(\mathcal{T}_{\mathrm{bk}})$, defined in (41) in the context of 4d Chern-Simons theory, is isomorphic to the Yangian $Y_{\hbar}(\mathfrak{gl}_K)$:*

$$\mathcal{A}^{\mathrm{Sc}}(\mathcal{T}_{\mathrm{bk}}) \overset{N \to \infty}{\cong} Y_{\hbar}(\mathfrak{gl}_K). \tag{99}$$

The 4d Chern-Simons theory with gauge group $\mathrm{GL}_K$, also denoted by $\mathrm{CS}_K^4$, is defined by the action (29), which we repeat here for convenience:

$$S_{\mathrm{CS}} := \frac{i}{2\pi} \int_{\mathbb{R}^2_{x,y} \times \mathbb{C}_z} \mathrm{d}z \wedge \mathrm{tr}_{\mathbf{K}}\left(A \wedge \mathrm{d}A + \frac{2}{3} A \wedge A \wedge A\right). \tag{100}$$

The trace in the fundamental representation defines a positive-definite metric on $\mathfrak{gl}_K$, moreover, we choose a basis of $\mathfrak{gl}_K$, denoted by $\{t_\mu\}$, in which the metric becomes diagonal:

$$\mathrm{tr}_{\mathbf{K}}(t_\mu t_\nu) \propto \delta_{\mu\nu}. \tag{101}$$

We consider this theory in the presence of a Wilson line in some representation $\varrho : \mathfrak{gl}_K \to \mathrm{End}(V)$, supported along the line $L$ defined by $y = z = 0$:

$$W_\varrho(L) = P \exp\left(\int_L \varrho(A)\right). \tag{102}$$

Consideration of fusion of Wilson lines to give rise to Wilson lines in tensor product representation shows that it is not only the connection $A$ that couples to a Wilson line but also its derivatives $\partial_z^n A$ [19]. Furthermore, gauge invariance at the classical level requires that $\partial_z^n A$ couples to the Wilson line via a representation of the loop algebra $\mathfrak{gl}_K[z]$. So the line operator that we consider is the following:

$$P \exp\left(\sum_{n \geq 0} \varrho_{\mu,n} \int_L \partial_z^n A^\mu\right), \tag{103}$$

where the matrices $\varrho_{\mu,n} \in \mathrm{End}(V)$ satisfy:

$$\left[\varrho_{\mu,m}, \varrho_{\nu,n}\right] = f_{\mu\nu}{}^\xi \varrho_{\xi,m+n}. \tag{104}$$

The structure constant $f_{\mu\nu}{}^\xi$ is that of $\mathfrak{gl}_K$. In particular, we have $\varrho_{\mu,0} = \varrho(t_\mu)$.

In (28), $A$ was defined to not have a $\mathrm{d}z$ component. The reason is that, due to the appearance of $\mathrm{d}z$ in the above action (100), the $\mathrm{d}z$ component of the connection $A$ never appears in the action anyway.[31]

Though the theory is topological, in order to do concrete computations, such as imposing gauge fixing conditions, computing propagator, and evaluating Witten diagrams etc. we need to make a choice of metric on $\mathbb{R}^2_{x,y} \times \mathbb{C}_z$, we choose:[32]

$$\mathrm{d}s^2 = \mathrm{d}x^2 + \mathrm{d}y^2 + \mathrm{d}z\mathrm{d}\bar{z}. \tag{107}$$

---

[31]Had we defined the space of connections to be $\Omega^1(\mathbb{R}^2_{x,y} \times \mathbb{C}_z) \otimes \mathfrak{gl}_K$, then, in addition to the usual $\mathrm{GL}_K$ gauge symmetry, we would have to consider the following additional gauge transformation:

$$A \to A + f \, \mathrm{d}z, \tag{105}$$

for arbitrary function $f \in \Omega^0(\mathbb{R}^2 \times \mathbb{C})$. We could fix this gauge by imposing:

$$A_z = 0. \tag{106}$$

This would get us back to the space $\left(\Omega^1(\mathbb{R}^2_{x,y} \times \mathbb{C}_z)/(\mathrm{d}z)\right) \otimes \mathfrak{gl}_K$.

[32]For this theory we follow the choices of [19] whenever possible.

For the $\mathrm{GL}_K$ gauge symmetry we use the following gauge fixing condition:

$$\partial_x A_x + \partial_y A_y + 4\partial_z A_{\bar{z}} = 0. \tag{108}$$

The propagator is defined as the two-point correlation function:

$$P^{\mu\nu}(v_1, v_2) := \langle A^\mu(v_1) A^\nu(v_2) \rangle. \tag{109}$$

Since in the basis of our choice the Lie algebra metric is diagonal (101), this propagator is proportional to a Kronecker delta in the Lie algebra indices:

$$P^{\mu\nu}(v_1, v_2) = \delta^{\mu\nu} P(v_1, v_2), \tag{110}$$

where $P$ is a 2-form on $\mathbb{R}^4_{v_1} \times \mathbb{R}^4_{v_2}$. We can fix one of the coordinates to be the origin, this amounts to taking the projection:

$$\varpi : \mathbb{R}^4_{v_1} \times \mathbb{R}^4_{v_1} \to \mathbb{R}^4_v, \qquad \varpi : (v_1, v_2) \mapsto v_1 - v_2 =: v. \tag{111}$$

Due to translation invariance, $P$ can be written as a pullback of some 2-form on $\mathbb{R}^4$ by $\varpi$, i.e., $P = \varpi^* \overline{P}$ for some $\overline{P} \in \Omega^2(\mathbb{R}^4)$. The propagator $P$ can be characterized as the Green's function for the differential operator $\frac{i}{2\pi\hbar} \mathrm{d}z \wedge \mathrm{d}$ that appears in the kinetic term of the action $S_{\mathrm{CS}}$. For $\overline{P}$ this results in the following equation:

$$\frac{i}{2\pi\hbar} \mathrm{d}z \wedge \mathrm{d}\overline{P}(v) = \delta^4(v) \mathrm{d}x \wedge \mathrm{d}y \wedge \mathrm{d}z \wedge \mathrm{d}\bar{z}, \tag{112}$$

The propagator $P$, and in turns $\overline{P}$, must also satisfy the gauge fixing condition (108):

$$\partial_x \overline{P}_x + \partial_y \overline{P}_y + 4\partial_z \overline{P}_{\bar{z}} = 0. \tag{113}$$

The solution to (112) and (113) is given by:

$$\overline{P}(x, y, z, \bar{z}) = \frac{\hbar}{2\pi} \frac{x \, \mathrm{d}y \wedge \mathrm{d}\bar{z} + y \, \mathrm{d}\bar{z} \wedge \mathrm{d}x + 2\bar{z} \, \mathrm{d}x \wedge \mathrm{d}y}{(x^2 + y^2 + z\bar{z})^2}. \tag{114}$$

The propagator $P(v_1, v_2)$ will be referred to as the *bulk-to-bulk* propagator, since the points $v_1$ and $v_2$ can be anywhere in the world-volume $\mathbb{R}^2_{x,y} \times \mathbb{C}_z$ of CS theory. To compute Witten diagrams we also need a *boundary-to-bulk* propagator. We will denote it as $\mathsf{K}_\mu(v, x) \equiv \mathsf{K}(v, x) t_\mu$, where $v \in \mathbb{R}^2_{x,y} \times \mathbb{C}_z$ and $x \in \ell_\infty(z)$ is restricted to the boundary line. The boundary-to-bulk propagator is a 1-form defined as a solution to the classical equation of motion:

$$\mathrm{d}z_v \wedge \mathrm{d}_v \mathsf{K}(v, x) = 0, \tag{115}$$

and by the condition that when pulled back to the boundary, in this case $\ell_\infty(z)$, it must become a delta function supported at $x$:

$$\varepsilon^* \mathsf{K}(x', x) = \delta^1(x' - x) \mathrm{d}x', \qquad x' \in \ell_\infty(z), \tag{116}$$

where $\varepsilon : \ell_\infty(z) \hookrightarrow \mathbb{R}^2 \times \mathbb{C}$ is the embedding of the line in the larger 4d world-volume. As our boundary-to-bulk propagator we choose the following:

$$\mathsf{K}(v, x) = \mathrm{d}_v \theta(x_v - x) = \delta^1(x_v - x) \mathrm{d}x_v, \tag{117}$$

where $x_v$ refers to the $x$-coordinate of the bulk point $v$. The function $\theta$ is the following step function:

$$\theta(x) = \begin{cases} 1 & \text{for } x > 0 \\ 1/2 & \text{for } x = 0 \\ 0 & \text{for } x < 0 \end{cases}. \tag{118}$$

Note that we have functional derivatives with respect to $\partial_z^n A$ for $n \in \mathbb{N}_{\geq 0}$. The propagator (117) corresponds to the functional derivative with $n = 0$. Let us denote the propagator corresponding to $\frac{\delta}{\delta \partial_z^n A}$, more generally, as $\mathsf{K}_n$, and for $n \geq 0$, we modify the condition (116) by imposing:

$$\lim_{v \to x'} \varepsilon^* \partial_z^n \mathsf{K}(v, x) = \delta^1(x' - x)\mathrm{d}x', \qquad x' \in \ell_\infty(z). \tag{119}$$

This leads us to the following generalization of (117):

$$\mathsf{K}_n(v, x) = z_v^n \delta^1(x_v - x)\mathrm{d}x_v. \tag{120}$$

Apart from the two propagators, we shall need the coupling constant of the theory to compute Witten diagrams. The coupling constant of this theory can be read off from the interaction term in the action $S_{\text{CS}}$, it is:

$$\frac{i}{2\pi\hbar} f_{\mu\nu}{}^\xi \mathrm{d}z. \tag{121}$$

Now we can give a diagrammatic definition of the operators in the algebra $\mathcal{A}^{\text{Sc}}(\mathcal{T}_{\text{bk}})$, namely the ones defined in (39), and their products:

$$T_{\mu_1}[n_1](p_1) \cdots T_{\mu_m}[n_m](p_m) = \sum_{l=1}^{\infty} \sum_{j_i \geq 0} \quad . \tag{122}$$

Let us clarify some points about the picture. We have replaced the pair of fundamental-anti-fundamental indices on $T$ with a single adjoint index. The bottom horizontal line represents the boundary line $\ell_\infty(z)$, and the top horizontal line represents the Wilson line in representation $\varrho : \mathfrak{gl}_K \to V$ at $y = 0$. The sum is over the number of propagators attached to the Wilson line and all possible derivative couplings. The orders of the derivatives are mentioned in the boxes. The points $q_1 \leq \cdots \leq q_l$ on the Wilson line are all integrated along the line without changing their order. The gray blob represents a sum over all possible graphs consistent with the external lines. We use different types of lines to represent different entities:

$$\begin{aligned}
\text{Bulk-to-bulk propagator, } P(v_1, v_2) = \ & v_1 \;\text{———}\; v_2 \;, \\
\text{Boundary-to-bulk propagator, } \mathsf{K}(v, x) = \ & v \;\text{———}\; x \;, \\
\text{The boundary line } \ell_\infty(z) : \ & \text{-----} \;, \\
\text{Wilson line} : \ & \text{··········}.
\end{aligned} \tag{123}$$

The labels $\mu_i, n_i$ below the points along the boundary line implies that the corresponding boundary-to-bulk propagator is $\mathsf{K}_{n_i} = z^{n_i} \mathsf{K}$ and that it carries a $\mathfrak{gl}_K$-index $\mu_i$. Finally, the $j$th derivative of $A^\nu$ couples to the Wilson line via the matrix $\varrho_{\nu,j}$. Such a diagram with $m$ boundary-to-bulk propagators and $l$ bulk-to-bulk propagators attached to the Wilson lines will be evaluated to an element of $\text{End}(V)$ which will schematically look like:

$$(\Gamma_{m \to l})^{\mu_1 \cdots \mu_l}_{\nu_1 \cdots \nu_m} \varrho_{\mu_1, j_1} \cdots \varrho_{\mu_l, j_l}, \tag{124}$$

where $(\Gamma_{m \to l})^{\mu_1 \cdots \mu_m}_{\nu_1 \cdots \nu_l}$ is a number that will be found by evaluating the Witten diagram. Since the bulk-to-bulk propagator (114) is proportional to $\hbar$ and the interaction vertex (121) is

proportional to $\hbar^{-1}$, each diagram will come with a factor of $\hbar$ that will be related to the Euler character of the graph.[33] In the following we start computing diagrams starting from $\mathcal{O}(\hbar^0)$ and up to $\mathcal{O}(\hbar^2)$, by the end of which we shall have proven the main result (Proposition 2) of this section.

*Remark* 6 (Diagrams as $m \to l$ maps, and deformation). Each $m \to l$ Witten diagram that appears in sums such as (122) can be interpreted as a map whose image is the value of the diagram:

$$\Gamma_{m \to l} : \bigotimes_{i=1}^{m} z^{n_i} \mathfrak{gl}_K \to \bigotimes_{i=1}^{l} z^{j_i} \mathfrak{gl}_K \to \mathrm{End}(V),$$

$$\Gamma_{m \to l} : \bigotimes_{i=1}^{m} z^{n_i} t_{\mu_i} \mapsto (\Gamma_{m \to l})^{\mu_1 \cdots \mu_l}_{\nu_1 \cdots \nu_m} \varrho_{\mu_1, j_1} \cdots \varrho_{\mu_l, j_l}. \tag{125}$$

As we shall see explicitly in our computations, diagrams in (122) without loops (diagrams of $\mathcal{O}(\hbar^0)$) define an associative product that leads to classical algebras such as $\mathrm{U}(\mathfrak{gl}_K[z])$. However, there are generally more diagrams in (122) involving loops (diagrams of $\mathcal{O}(\hbar)$ and higher order) that change the classical product to something else. Since loops in Witten or Feynman diagrams are the essence of the quantum interactions, classical algebras deformed by such loop diagrams are aptly called quantum groups (of course, why they are called groups is a different story entirely [45].) △

## 5.1 Relation to anomaly of Wilson line

As we shall compute relevant Witten diagrams of the 4d Chern-Simons theory in detail in later sections, we shall find that the computations are essentially similar to the computations of gauge anomaly of the Wilson line [19] in this theory. This of course is not a coincidence. To see this, let us consider the variation of the expectation value of the Wilson line, $\langle W_\varrho(L) \rangle_A$, as we vary the connection $A$ along the boundary line $\ell_\infty(z)$:

$$\delta \langle W_\varrho(L) \rangle_A = \sum_{n=0}^{\infty} \int_{p \in \ell_\infty(z)} \frac{\delta}{\delta \partial_z^n A^\mu(p)} \langle W_\varrho(L) \rangle_A \, \delta \partial_z^n A^\mu(p). \tag{126}$$

Let us make the following variation:

$$\delta \partial_z A^\mu(x) = \delta^1(x-p)\eta^\mu = \mathrm{d}_x \theta(x-p)\eta^\mu, \tag{127}$$

for some fixed Lie algebra element $\eta^\mu t_\mu \in \mathfrak{gl}_K$. Then we find:

$$\delta \langle W_\varrho(L) \rangle_A = \frac{\delta}{\delta \partial_z A^\mu(p)} \langle W_\varrho(L) \rangle_A \, \eta^\mu. \tag{128}$$

An exact variation of the boundary value of the connection is like a gauge transformation that does not vanish at the boundary. In [19] it was proved that such a variation of the connection leads to a variation of the Wilson line which is a local functional supported on the line:

$$\delta \langle W_\varrho(L) \rangle_A = \left( \left[ \varrho_{\mu,1}, \varrho_{\nu,1} \right] + \Theta_{\mu,1,\nu,1} \right) \int_L \partial_z A^\mu \partial_z \mathsf{c}^\nu, \tag{129}$$

---

[33]In a Feynman diagram all propagators are proportional to $\hbar$ and the power of $\hbar$ of a diagram relates simply to the number of faces of the diagram, which is why $\hbar$ is called the loop counting parameter. In a Witten diagram the boundary-to-bulk propagators do not carry any $\hbar$ and therefore the power of $\hbar$ depends also on the number of boundary-to-bulk propagators. However, we are going to ignore this point and simply refer to the power of $\hbar$ in a diagram as the loop order of that diagram.

where c was the generator of the gauge transformation:

$$\partial_z \mathrm{d}c^\mu = \delta \partial_z A^\mu, \tag{130}$$

$\varrho_{\mu,1} \in \mathrm{End}(V)$ is part of the representation of $\mathfrak{gl}_K[z]$ that couples $\partial_z A^\mu$ to the Wilson line (see (103)), and $\Theta_{\mu,1,\nu,1}$, which is anti-symmetric in $\mu$ and $\nu$, is a matrix that acts on $V$. Variations such as the above measure gauge anomaly associated to the line, though in our case it is not an anomaly since we are varying the connection at the boundary, and such "large gauge" transformations are not actually part of the gauge symmetry of the theory. The matrix $\Theta_{\mu,1,\nu,1}$ which signals the presence of anomaly is not an arbitrary matrix and in [19], all constraints on this matrix were worked out, we shall not need them at the moment. Comparing with (127) we see that for us $\partial_z c^\mu(x) = \theta(x - p)\eta^\mu$, which leads to:

$$\delta \left\langle W_\varrho(L) \right\rangle_A = \left( f_{\mu\nu}{}^\xi \varrho_{\xi,2} + \Theta_{\mu,1,\nu,1} \right) \int_{x>p} \partial_z A^\mu \eta^\nu, \tag{131}$$

where we have used the fact that the matrices $\varrho_{\mu,1}$ satisfy the loop algebra (104). The integral above is along $L$. The connection $A$ above is a background connection satisfying the equation of motion, i.e., it is flat. Since the D4 world-volume, even in the presence of a Wilson line, has no non-contractible loop, all flat connections are exact. Symmetry of world-volume dictates in particular that the connection must also be translation invariant along the direction of the Wilson line $L$. By considering the integral of $A$ along the following rectangle:

$$\tag{132}$$

and using translation invariance in the $x$-direction along with Stoke's theorem, we can change the support of the integral in (131) from $L$ to $\ell_\infty(z)$, to get:

$$\delta \left\langle W_\varrho(L) \right\rangle_A = \left( f_{\mu\nu}{}^\xi \varrho_{\xi,2} + \Theta_{\mu,1,\nu,1} \right) \int_{\ell_\infty(z) \ni x>p} \partial_z A^\mu \eta^\nu. \tag{133}$$

Comparing with (128) we find:

$$\frac{\delta}{\delta \partial_z A^\nu(p)} \left\langle W_\varrho(L) \right\rangle_A = \left( f_{\mu\nu}{}^\xi \varrho_{\xi,2} + \Theta_{\mu,1,\nu,1} \right) \int_{x>p} \partial_z A^\mu, \tag{134}$$

where the integral is now along the boundary line $\ell_\infty(z)$. This leads to the following relation between our algebra and anomaly:

$$\begin{aligned}
&\left[ T_\mu[1], T_\nu[1] \right] \\
&= \lim_{\iota \to 0} \left[ \frac{\delta}{\delta \partial_z A^\mu(p+\iota)} \frac{\delta}{\delta \partial_z A^\nu(p)} - \frac{\delta}{\delta \partial_z A^\nu(p)} \frac{\delta}{\delta \partial_z A^\mu(p+\iota)} \right] \left\langle W_\varrho(L) \right\rangle_A \\
&= f_{\mu\nu}{}^\xi \varrho_{\xi,2} + \Theta_{\mu,1,\nu,1}.
\end{aligned} \tag{135}$$

The first term with the structure constant gives us the loop algebra $\mathfrak{gl}_K[z]$, which is the classical result. The anomaly term is the result of 2-loop dynamics [19], i.e., it is proportional to $\hbar^2$. This term gives the quantum deformation of the classical loop algebra. This also explains

why our two loop computation of the algebra is similar to the two loop computation of anomaly from [19].

At this point, we note that we can actually just use the result of [19] to find out what $\Theta_{\mu,1,\nu,1}$ is and we would find that the deformed algebra of the operators $T^\mu[n]$ is indeed the Yangian $Y_\hbar(\mathfrak{gl}_K)$. However, we think it is illustrative to derive this result from a direct computation of Witten diagrams.

## 5.2 Classical algebra, $\mathcal{O}(\hbar^0)$

### 5.2.1 Lie bracket

We denote a diagram by $\Gamma^d_{n\to m}$ when there are $n$ boundary-to-bulk propagators, $m$ propagators attached to the Wilson line, and the diagram is of order $\hbar^d$. If there are more than one such diagrams we denote them as $\Gamma^d_{n\to m,i}$ with $i = 1, \cdots$.

Our aim in this section is to compute the product $T_\mu[m](p_1)T_\nu[n](p_2)$ and eventually the commutator

$$\left[T_\mu[m], T_\nu[n]\right] := \lim_{p_2 \to p_1}\left(T_\mu[m](p_1)T_\nu[n](p_2) - T_\nu[n](p_1)T_\mu[m](p_2)\right), \tag{136}$$

at 0-loop.[34]

We have the following two $2 \to 2$ diagrams:

$$\Gamma^0_{2\to2,1}\left(\begin{smallmatrix}p_1\\\mu,m\end{smallmatrix};\begin{smallmatrix}p_2\\\nu,n\end{smallmatrix}\right) = \quad , \qquad \Gamma^0_{2\to2,2}\left(\begin{smallmatrix}p_1\\\mu,m\end{smallmatrix};\begin{smallmatrix}p_2\\\nu,n\end{smallmatrix}\right) = \quad , \tag{137}$$

where a label $m$ in a box on the Wilson line refers to the coupling between the Wilson line and the $m$th derivative of the connection. The first diagram evaluates to (note that $p_1 < p_2$ and $q_1 < q_2$):

$$\begin{aligned}\Gamma^0_{2\to2,1}\left(\begin{smallmatrix}p_1\\\mu,m\end{smallmatrix};\begin{smallmatrix}p_2\\\nu,n\end{smallmatrix}\right) &= \int_{q_1<q_2} dq_1 dq_2\, \delta^1(q_1-p_1)\delta^1(q_2-p_2)\varrho^\mu_m\varrho^\nu_n \\ &= \varrho_{\mu,m}\varrho_{\nu,n},\end{aligned} \tag{138}$$

and the second one (with $p_1 < p_2$ and $q_1 > q_2$):

$$\begin{aligned}\Gamma^0_{2\to2,2}\left(\begin{smallmatrix}p_1\\\mu,m\end{smallmatrix};\begin{smallmatrix}p_2\\\nu,n\end{smallmatrix}\right) &= \int_{q_1>q_2} dq_2 dq_1\, \delta^1(q_1-p_1)\delta^1(q_2-p_2)\varrho_{\nu,n}\varrho_{\mu,m} \\ &= 0.\end{aligned} \tag{139}$$

Therefore their contribution to the commutator is:

$$\begin{aligned}\left[T_\mu[m], T_\nu[n]\right] &= \lim_{p_2\to p_1}\left(\Gamma^0_{2\to2,1}\left(\begin{smallmatrix}p_1\\\mu,m\end{smallmatrix};\begin{smallmatrix}p_2\\\nu,n\end{smallmatrix}\right) - \Gamma^0_{2\to2,1}\left(\begin{smallmatrix}p_1\\\nu,n\end{smallmatrix};\begin{smallmatrix}p_2\\\mu,m\end{smallmatrix}\right)\right) \\ &= \left[\varrho_{\mu,m}, \varrho_{\nu,n}\right] = f_{\mu\nu}{}^\xi\varrho_{\xi,m+n} = f_{\mu\nu}{}^\xi T_\xi[m+n],\end{aligned} \tag{140}$$

---

[34] $\left[T_\mu[m](p_1), T_\nu[n](p_1)\right]$ may be a more accurate notation but this algebra must be position invariant and therefore we shall ignore the position. Reference to the position only matters when different operators are positioned at different locations.

where the last equality is established by evaluating the diagram:

$$
\tag{141}
$$

The bracket (140) is precisely the Lie bracket in the loop algebra $\mathfrak{gl}_K[z]$. Note in passing that had we considered the same diagrams as the ones in (137) except with different derivative couplings at the Wilson line then the diagrams would have vanished, either because there would be more $z$-derivatives than $z$, or there would be less, in which case there would be $z$'s floating around which vanish along the Wilson line located at $y = z = 0$.

There is one $2 \to 1$ diagram as well:

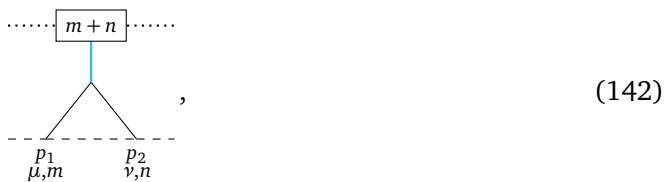

$$
\tag{142}
$$

however, since the two boundary-to-bulk propagators are two parallel delta functions, i.e., their support are restricted to $x = p_1$ and $x = p_2$ respectively with $p_1 \neq p_2$, they never meet in the bulk and therefore the diagram vanishes. There are no more classical diagrams, so the Lie bracket in the classical algebra is just the bracket in (140).

### 5.2.2 Coproduct

Apart from the Lie algebra structure, the algebra $\mathcal{A}^{\mathrm{Sc}}(\mathcal{T}_{\mathrm{bk}})$ also has a coproduct structure. This can be seen by considering the Wilson line in a tensor product representation, say $U \otimes V$. Such a Wilson line can be produced by considering two Wilson lines in representations $U$ and $V$ respectively and bringing them together, and asking how $T_\mu[n]$ acts on $U \otimes V$. Since there are going to be multiple vector spaces in this section, let us distinguish the actions of $T_\mu[n]$ on them by a superscript, such as, $T_\mu^U[n]$, $T_\mu^V[n]$, etc. At the classical level the answer to the question we are asking is simply given by computing the following diagrams:

$$
\tag{143}
$$

Evaluation of these diagrams is very similar to that of the diagrams in (137) and the result is:

$$
T_\mu^{U \otimes V}[m] = T_\mu^U[m] \otimes \mathrm{id}_V + \mathrm{id}_U \otimes T_\mu^V[m].
\tag{144}
$$

This is the same coproduct structure as that of the universal enveloping algebra $\mathrm{U}(\mathfrak{gl}_K[z])$.

Combining the results of this section and the previous one we find that, at the classical level we have an associative algebra with generators $T_\mu[n]$ with a Lie bracket and coproduct given by the Lie bracket of the loop algebra $\mathfrak{gl}_K[z]$ and the coproduct of its universal enveloping algebra. This identifies $\mathcal{A}^{\mathrm{Sc}}(\mathcal{T}_{\mathrm{bk}})$, clasically, as the universal enveloping algebra itself:

**Lemma 1.** *The large $N$ limit of the algebra $\mathcal{A}^{\mathrm{Sc}}(\mathcal{T}_{\mathrm{bk}})$ at the classical level is the universal enveloping algebra $U(\mathfrak{gl}_K[z])$:*

$$
\mathcal{A}^{\mathrm{Sc}}(\mathcal{T}_{\mathrm{bk}})/\hbar \overset{N \to \infty}{\cong} U(\mathfrak{gl}_K[z]) \cong Y_\hbar(\mathfrak{gl}_K)/\hbar.
\tag{145}
$$

The reason why we need to take the large $N$ limit is that, the operators $T_\mu[m]$ acts on a vector space which is finite dimensional for finite $N$. This leads to some extra relations in the algebra, which we can get rid of in the large $N$ limit. A similar argument was presented for the operator algebra coming from the BF theory in §4.3 and the argument in the context of the CS theory will be explained in more detail in §5.4.

## 5.3 Loop corrections

### 5.3.1 1-loop, $\mathcal{O}(\hbar)$

Now we want to compute 1-loop deformation to both the Lie algebra structure and the co-product structure of $\mathcal{A}^{\mathrm{Sc}}(\mathcal{T}_{\mathrm{bk}})$.

**Lie bracket.** The $2 \to 1$ diagrams at this loop order are:[35]

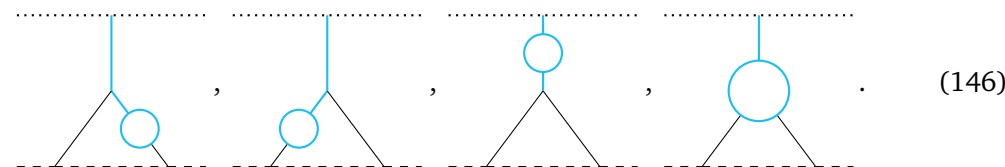

$$\tag{146}$$

All of these vanish due to Lemma 6 of §C.1.

The $2 \to 2$ diagrams at this loop order are:

$$\tag{147}$$

Note that, since the bulk points are being integrated over, crossing the boundary-to-bulk propagators does not produce any new diagram, it just exchanges the two diagrams that we have drawn:

$$\tag{148}$$

For this reason, in future we shall only draw diagrams up to crossing of the boundary-to-bulk propagators that are connected to bulk interaction vertices.

Now let us comment on the evaluation of the diagrams in (147). We start by doing integration by parts with respect the differential corresponding to either one of the two boundary-to-bulk propagators. As mentioned in §C.2, this gives two kinds of contributions, one coming from collapsing a bulk-to-bulk propagator, the other coming from boundary terms. Collapsing any of the bulk-to-bulk propagators leads to a configuration which will vanish due to Lemma 7 (§C.1). Therefore, doing integration by parts will only result in a boundary term. Recall from the general discussion in §C.2 that only the boundary component of the integrals along the Wilson line can possibly contribute. Since there are two points on the Wilson line, let us call them $p_1$ and $p_2$, the domain of integration is:

$$\Delta_2 = \{(p_1, p_2) \in \mathbb{R}^2 \mid p_1 < p_2\}. \tag{149}$$

---

[35]Sometimes we ignore to specify the derivative couplings at the Wilson line, when the diagrams we draw are vanishing regardless.

The boundary of this domain is:

$$\partial \Delta_2 = \{(p_1, p_2) \in \mathbb{R}^2 \,|\, p_1 = p_2\}. \tag{150}$$

Once restricted to this boundary, both of the diagrams in (147) will involve a configuration such as the following:

(151)

which vanishes due to Lemma 6.[36] The diagrams (147) thus vanish.

There are four other $2 \to 2$ diagrams at 1-loop, they can be generated by starting with:

(152)

and then

1. Permuting the two points on the Wilson line.

2. Permuting the two points on the boundary.

3. Simultaneously permuting the two points on the Wilson line and the two points on the boundary.

All of these diagrams vanish due to Lemma 6.

There are also six $2 \to 3$ diagrams. All of these can be generated from the following:

(153)

by permuting the points along the Wilson line and the boundary. However, due to Lemma 7, all of these diagrams vanish.

There are no more $2 \to m$ diagrams at 1-loop. Thus, we conclude that there is no 1-loop contribution to the Lie bracket in $\mathcal{A}^{\mathrm{Sc}}(\mathcal{T}_{\mathrm{bk}})$.

**Coproduct.** We use the same superscript notation we used in §5.2.2 to distinguish between the actions of $T_\mu[m]$ on different vector spaces. The 1-loop diagram that deforms the classical coproduct is the following:

$$\Gamma^1_{1 \to 2}\left(\begin{smallmatrix} p \\ \mu, 1 \end{smallmatrix}\right) = \tag{154}$$

Happily for us, precisely this diagram was computed in eq. 5.6 of [19] to answer the question "how does a background connection couple to the product Wilson line?". The result of

---

[36]These diagrams actually require a UV regularization due to logarithmic divergence coming from the two points on the Wilson line being coincident. To regularize, the domain of integration needs to be restricted from $\Delta_2$ to $\widetilde{\Delta}_2 := \{(p_1, p_2) \in \mathbb{R}^2 \,|\, p_1 \le p_2 - \epsilon\}$ for some small positive number $\epsilon$, which leads to the modified boundary equation $p_1 = p_2 - \epsilon$, however, this does not affect the arguments presented in the proof of Lemma 6 (essentially because $\epsilon$ is a constant and $\mathrm{d}\epsilon = 0$, resulting in no new forms other than the ones considered in the proof), and therefore we are not going to describe the regularization of these diagrams in detail.

that paper involved an arbitrary background connection where we have our boundary-to-bulk propagator, so we just need to replace that with $\mathsf{K}_1(v,p) = z_v \delta^1(x_v - p)$ and we find:

$$\Gamma^1_{1\to2}\binom{p}{\mu,1} = -\frac{\hbar}{2} f_\mu{}^{\nu\xi} T^U_\nu[0] \otimes T^V_\xi[0]. \tag{155}$$

This deforms the classical coproduct (144) as follows:

$$T^{U\otimes V}_\mu[1] = T^U_\mu[1] \otimes \mathrm{id}_V + \mathrm{id}_U \otimes T^V_\mu[1] - \frac{\hbar}{2} f_\mu{}^{\nu\xi} T^U_\nu[0] \otimes T^V_\xi[0]. \tag{156}$$

The exact same computation with $\mathsf{K}_0$ instead of $\mathsf{K}_1$ shows that $\Gamma^1_{1\to2}\binom{p}{\mu,0} = 0$, i.e., the classical algebra of the 0th level operators remain entirely undeformed at this loop order.[37]

Thus we see that at 1-loop, the Lie algebra structure in $\mathcal{A}^{\mathrm{Sc}}(\mathcal{T}_{\mathrm{bk}})$ remains undeformed, but there is a non-trivial deformation of the coalgebra structure. At this point, there is a mathematical shortcut to proving that the algebra $\mathcal{A}^{\mathrm{Sc}}(\mathcal{T}_{\mathrm{bk}})$, including all loop corrections, is the Yangian. The proof relies on a uniqueness theorem (Theorem 12.1.1 of [45]) concerning the deformation of $U(\mathfrak{gl}_K[z])$. Being able to use the theorem requires satisfying some technical conditions, we discuss this proof in Appendix B. This proof is independent of the rest of the paper, where we compute two loop corrections to the commutator (140) which will directly show that the algebra is the Yangian.

### 5.3.2  2-loops, $\mathcal{O}(\hbar^2)$

The number of 2-loop diagrams is too large to list them all, and most of them are zero. Instead of drawing all these diagrams let us mention how we can quickly identify a large portion of the diagrams that end up being zero.

Consider the following transformations that can be performed on a propagator or a vertex in any diagram:



$$\tag{157}$$

All these transformations increase the order of $\hbar$ by one, however, all the diagrams constructed using these modifications are zero due to Lemma 6. We will therefore ignore such diagrams. Let us now identify potentially non-zero $2 \to m$ diagrams at 2-loops.

All 2-loop $2 \to 1$ diagrams are created from lower order diagrams using modifications such as (157). All of them vanish.

For $2 \to 2$ diagrams, ignoring those that are results of modifications such as (157) or that are product of lower order vanishing diagrams, we are left with the sum of the following

---

[37]Note that the 0th level operators form a closed algebra which is nothing but the Lie algebra $\mathfrak{gl}_K$. Reductive Lie algebras belong to discrete isomorphism classes and therefore they are robust against continuous deformations. So the algebra of $T_\mu[0]$ will in fact remain undeformed at all loop orders. We will not make more than a few remarks about them in the future.

diagrams:

$$\Gamma^2_{2\to2,1} = \quad , \qquad \Gamma^2_{2\to2,2} = \quad ,$$

$$\Gamma^2_{2\to2,3} = \quad , \qquad \Gamma^2_{2\to2,4} = \quad . \tag{158}$$

Let us first consider the first two diagrams $\Gamma^2_{2\to2,1}$ and $\Gamma^2_{2\to2,2}$. Collapsing any of the bulk-to-bulk propagators will result in a configuration where either Lemma 6 or 7 is applicable. Therefore, when we do integration by parts with respect to the differential in one of the two boundary-to-bulk propagators we only get a boundary term. The boundary corresponds to the boundary of $\Delta_2$ (defined in (149)), and when restricted to this boundary, the integrand vanishes due to Lemma 7, in the same way as for the diagrams in (147).[38]

The diagrams $\Gamma^2_{2\to2,3}$ and $\Gamma^2_{2\to2,4}$ are symmetric under the exchange of the color labels associated to the boundary-to-bulk propagators, for a proof see the discussion following (239). So these diagrams don't contribute to the anti-symmetric commutator we are computing.

Now we come to the most involved part of our computations, $2 \to 3$ diagrams at 2-loops. We have the sum of the following diagrams:

$$\Gamma^2_{2\to3,1} = \quad , \qquad \Gamma^2_{2\to3,2} = \quad , \qquad \Gamma^2_{2\to3,3} = \quad ,$$

$$\Gamma^2_{2\to3,4} = \quad , \qquad \Gamma^2_{2\to3,5} = \quad , \qquad \Gamma^2_{2\to3,6} = \quad . \tag{159}$$

All of these diagrams are in fact non-zero. We proceed with the evaluation of the diagram $\Gamma^2_{2\to3,1}$:

$$\Gamma^2_{2\to3,1}\left(\begin{smallmatrix} p_1 \\ \mu,1 \end{smallmatrix}; \begin{smallmatrix} p_2 \\ \nu,1 \end{smallmatrix}\right) = \quad \tag{160}$$

The $\mathfrak{gl}_K$ factor of the diagram is easily evaluated to be:

$$f_\mu{}^{\xi o} f_\xi{}^{\pi\rho} f_{\nu\pi}{}^\sigma \varrho(t_o)\varrho(t_\rho)\varrho(t_\sigma). \tag{161}$$

The numerical factor takes a bit more care. Each of the bulk points $v_i = (x_i, y_i, z_i, \bar{z}_i)$ is integrated over $M = \mathbb{R}^2 \times \mathbb{C}$ and the points $q_i$ on the Wilson line take value in the simplex

---

[38]These diagrams are linearly divergent when the two points on the Wilson line are coincident and they require similar UV regularization as their 1-loop counterparts.

$\Delta_3 = \{(q_1, q_2, q_3) \in \mathbb{R}^3 \mid q_1 < q_2 < q_3\}$. For the sake of integration we can partially compactify the bulk to $M = \mathbb{R} \times S^3$. So the domain of integration for this diagram is:

$$M^3 \times \Delta_3. \tag{162}$$

However, this domain needs regularization due to UV divergences coming from the points $q_i$ all coming together. As in [19], we use a point splitting regulator, by restricting integration to the domain:

$$\widetilde{\Delta}_3 := \{(q_1, q_2, q_3) \in \Delta_3 \mid q_1 < q_3 - \epsilon\}, \tag{163}$$

for some small positive number $\epsilon$. We are not going to discuss the regulator here, as it would be identical to the discussion in [19]. We shall now do integration by parts with respect to the differential in the propagator connecting $p_1$ and $v_1$. Note that collapsing any of the bulk-to-bulk propagators leads to a configuration where the vanishing Lemma 7 applies. Therefore, contribution to the integral only comes from the boundary $M^3 \times \partial\widetilde{\Delta}_3$. The boundary of the simplex has three components, respectively defined by the constraints $q_1 = q_2$, $q_2 = q_3$, and $q_1 = q_3 - \epsilon$. However, when $q_1 = q_2$ or $q_2 = q_3$, we can use the vanishing Lemma 6 and the integral vanishes. Therefore the contribution to the diagram comes only from integration over:

$$M^3 \times \{(q_1, q_2, q_3) \in \widetilde{\Delta}_3 \mid q_1 = q_3 - \epsilon\}. \tag{164}$$

Further simplification can be made using the fact that the propagator connecting $p_2$ and $v_3$ is $z\delta^1(x_3 - p_2)$. This restricts the integration over $v_3$ to $\{p_2\} \times S^3$. However, using translation symmetry in the $x$-direction we can fix the position of $q_1$ at $(0,0,0,0)$ and allow the integration of $v_3$ over all of $M$. However, due to the presence of the delta function $\delta^1(x_3 - p_2)$ in the boundary-to-bulk propagator, $x_3$ and $p_1 = p_2 - \delta$ are rigidly tied to each other. This way, we end up with the following integration for the numerical factor:[39]

$$\frac{1}{2}\left(\frac{i}{2\pi\hbar}\right)^3 \int_{\substack{0 < q_2 < \epsilon \\ v_1, v_2, v_3}} dq_2 d^4v_1 d^4v_2 d^4v_3 \, \theta(x_1 - x_3^-) z_1 z_3 P(v_2, v_1)$$
$$\times P(v_3, v_2) P(q_1, v_1) P(q_2, v_2) P(q_3, v_3), \tag{165}$$

where $q_1 = (0,0,0,0)$, $q_2 = (p_2, 0, 0, 0)$, $q_3 = (\epsilon, 0, 0, 0)$, and $x_3^- := x_3 - \delta$, and since all the forms that appear are even we have ignored the wedge product symbols to be economic.

Before evaluating the above integral, we note that the diagram $\Gamma^2_{2\to3,4}$ evaluates to the same color factor and almost same numerical factor, except for a different step function:

$$\frac{1}{2}\left(\frac{i}{2\pi\hbar}\right)^3 \int_{\substack{0 < q_2 < \epsilon \\ v_1, v_2, v_3}} dq_2 d^4v_1 d^4v_2 d^4v_3 \, \theta(x_3 - x_1^-) z_1 z_3 P(v_2, v_1)$$
$$\times P(v_3, v_2) P(q_1, v_1) P(q_2, v_2) P(q_3, v_3). \tag{166}$$

Since we have to sum over all the diagrams, we use the fact that:

$$\lim_{\delta \to 0} \left(\theta(x_1 - x_3^-) + \theta(x_3 - x_1^-)\right) = 1, \tag{167}$$

to write:

$$\lim_{p_2 \to p_1} \left(\Gamma^2_{2\to3,1}\left(\begin{smallmatrix} p_1 \\ \mu,1 \end{smallmatrix}; \begin{smallmatrix} p_2 \\ \nu,1 \end{smallmatrix}\right) + \Gamma^2_{2\to3,4}\left(\begin{smallmatrix} p_1 \\ \mu,1 \end{smallmatrix}; \begin{smallmatrix} p_2 \\ \nu,1 \end{smallmatrix}\right)\right)$$
$$= f_\mu{}^{\xi o} f_\xi{}^{\pi \rho} f_{\nu\pi}{}^\sigma \varrho(t_o)\varrho(t_\rho)\varrho(t_\sigma) \left(\frac{i}{2\pi\hbar}\right)^3 \frac{1}{2} \int_{\substack{0 < q_2 < \epsilon \\ v_1, v_2, v_3}} dq_2 d^4v_1 d^4v_2 d^4v_3 \tag{168}$$
$$\times z_1 z_3 P(v_2, v_1) P(v_3, v_2) P(q_1, v_1) P(q_2, v_2) P(q_3, v_3).$$

---

[39]The factor of $1/2$ comes from diagram automorphisms.

Let us refer to the above integral by $\hbar^2 I_1$, so that we can write the right hand side of the above equation as:

$$\hbar^2 f_\mu{}^{\xi o} f_\xi{}^{\pi\rho} f_{\nu\pi}{}^\sigma \varrho(t_o)\varrho(t_\rho)\varrho(t_\sigma) I_1. \tag{169}$$

Similar considerations for the rest of the diagrams in (159) lead to similar expressions:

$$\lim_{p_2 \to p_1} \left( \Gamma^2_{2\to3,2} \left( \begin{smallmatrix} p_1 ; p_2 \\ \mu,1 ; \nu,1 \end{smallmatrix} \right) + \Gamma^2_{2\to3,5} \left( \begin{smallmatrix} p_1 ; p_2 \\ \mu,1 ; \nu,1 \end{smallmatrix} \right) \right) = \hbar^2 f_\mu{}^{\xi o} f_\xi{}^{\pi\rho} f_{\nu\pi}{}^\sigma \varrho(t_\rho)\varrho(t_o)\varrho(t_\sigma) I_2, \tag{170a}$$

$$\lim_{p_2 \to p_1} \left( \Gamma^2_{2\to3,2} \left( \begin{smallmatrix} p_1 ; p_2 \\ \mu,1 ; \nu,1 \end{smallmatrix} \right) + \Gamma^2_{2\to3,5} \left( \begin{smallmatrix} p_1 ; p_2 \\ \mu,1 ; \nu,1 \end{smallmatrix} \right) \right) = \hbar^2 f_\mu{}^{\xi o} f_\xi{}^{\pi\rho} f_{\nu\pi}{}^\sigma \varrho(t_o)\varrho(t_\sigma)\varrho(t_\rho) I_3, \tag{170b}$$

for two integrals $I_2$ and $I_3$ that are only slightly different from $I_1$.[40] To get the 2-loop contributions to the commutator $\left[ T_\mu[1], T_\nu,[1] \right]$ we need only to anti-symmetrize the expressions (169), (170). Putting them together with the classical result (140) we get the Lie bracket up to 2-loops:

$$\begin{aligned} \left[ T_\mu[1], T_\nu[1] \right] = f_{\mu\nu}{}^\xi T_\xi[2] + 2\hbar^2 f_{[\mu}{}^{\xi o} f_\xi{}^{\pi\rho} f_{\nu]\pi}{}^\sigma \big( & T_o[0]T_\rho[0]T_\sigma[0] I_1 \\ + & T_\rho[0]T_o[0]T_\sigma[0] I_2 + T_o[0]T_\sigma[0]T_\rho[0] I_3 \big), \end{aligned} \tag{171}$$

where we have replaced matrix products such as $\varrho(t_\rho)\varrho(t_o)\varrho(t_\sigma)$ with $T_\rho[0]T_o[0]T_\sigma[0]$ which is accurate up to the loop order shown. Thus we see that quantum corrections deform the classical Lie algebra of $\mathfrak{gl}_K[z]$.

## 5.4 Large $N$ limit: The Yangian

The deformed Lie bracket (171) may not look quite like the standard relations of the Yangian found in the literature, but we can choose a different basis to get to the standard relations, which we shall do momentarily.[41] However, for finite $N$, our algebra has more relations. Recall that the generators $T_\mu[1]$ act on the space $V$ where classically $V$ is a representation space, $\varrho : \mathfrak{gl}_K[z] \to \text{End}(V)$, of the loop algebra $\mathfrak{gl}_K[z]$ and the representation $\varrho$ was determined by the number $N$. The representation $\varrho$ depends on $N$ because $\varrho$ is the representation that couples the $\mathfrak{gl}_K$ connection $A$ to the Wilson line generated by integrating out $N \times K$ fermions. The representation is found by integrating out the fermions that define the line defect (23).[42] The important point for us is that, for finite $N$, $\varrho$ is finite dimensional. This implies that the generators $T_\mu[1]$ satisfy degree $d$ polynomial equations where $d = \dim(V)$. In the limit $N \to \infty$ these relations disappear and we have our isomorphism with the Yangian $Y(\mathfrak{gl}_K)$.[43]

---

[40]These integrals can be performed and their values are $I_2 = I_3 = \frac{1}{72}\left(2 - \frac{3}{\pi^2}\right)$, $I_1 = \frac{1}{36}\left(1 + \frac{3}{\pi^2}\right)$ though we postpone computing them until we no longer need to compute them.

[41]We can also appeal to the uniqueness theorem 12.1.1 of [45], in conjunction with the result of Appendix B, to conclude that the deformed algebra must be the Yangian $Y_\hbar(\mathfrak{gl}_K)$.

[42]By integrating out the fermions from the action (23) we get the holonomy of the connection $(A, A) \in \mathfrak{gl}_N \oplus \mathfrak{gl}_K$ in the following representation [34]:

$$\bigoplus_Y Y_N^T \otimes \overline{Y}_K, \tag{172}$$

where the sum is over all possible Young tableaux. $Y^T$ is the tableau we get by transposing the tableau $Y$ (i.e., turning rows into columns). $Y_N^T$ is the representation of $GL_N$ denoted by the tableau $Y^T$, and $\overline{Y}_K$ is the representation of $GL_K$ dual to the representation (of $GL_K$) denoted by $Y$. Had we started with a bosonic quantum mechanics instead, integrating out the bosons would result in a holonomy in the following representation [34]:

$$\bigoplus_Y Y_N \otimes \overline{Y}_K. \tag{173}$$

An important difference between (172) and (173) is that while the former is finite dimensional for finite $N$ and $K$, the latter is always infinite dimensional.

[43]In the case of bosonic quantum mechanics the representation $\varrho$ is actually infinite dimensional, however, for finite $N$ our Witten diagram computations can not be trusted, as the decoupling between closed string modes and defect (4d Chern-Simons) modes that we have assumed relies on the large $N$ limit [35].

**The Yangian in a more standard basis**

To get to a standard defining bracket for the Yangian, we change basis as follows. There is an ambiguity in $T_\xi[2]$. In (140) it was equal to $\varrho_{\xi,2}$ at the classical level, but it can be shifted at 2-loops (i.e., by a term proportional to $\hbar^2$) by the image $\vartheta(t_\xi)$ for an arbitrary $\mathfrak{gl}_K$-equivariant map $\vartheta : \mathfrak{gl}_K \to \text{End}(V)$. This shift simply corresponds to a different choice for the counterterm that couples $\partial_z^2 A^\xi$ to the Wilson line. Using this freedom we want to replace products such as $\varrho(t_o)\varrho(t_\rho)\varrho(t_\sigma)$ with the totally symmetric product $\{\varrho(t_o), \varrho(t_\rho), \varrho(t_\sigma)\}$ (defined in (94)). To this end, Consider the difference:

$$\Delta_{\mu\nu} := 2\hbar^2 f_{[\mu}{}^{\xi o} f_\xi{}^{\pi\rho} f_{\nu]\pi}{}^\sigma \left( \varrho(t_o)\varrho(t_\rho)\varrho(t_\sigma) - \{\varrho(t_o), \varrho(t_\rho), \varrho(t_\sigma)\} \right). \tag{174}$$

The square brackets around $\mu$ and $\nu$ in the above equation implies anti-symmetrization with respect to $\mu$ and $\nu$. The difference $\Delta_{\mu\nu}$ can be viewed as the image of the following $\mathfrak{gl}_K$-equivariant map:

$$\Delta : \wedge^2 \mathfrak{gl}_K \to \text{End}(V), \qquad \Delta : t_\mu \wedge t_\nu \mapsto \Delta_{\mu\nu}. \tag{175}$$

We now propose the following lemma:

**Lemma 2.** *The map $\Delta$ factors through $\mathfrak{gl}_K$, i.e., $\Delta : \wedge^2 \mathfrak{gl}_K \to \mathfrak{gl}_K \to \text{End}(V)$.*

The proof of this lemma involves some algebraic technicalities which we relegate to the Appendix §D. The utility of this lemma is that, it establishes the difference (174) as the image of an element of $\mathfrak{gl}_K$ which, according to our previous argument, can be absorbed into a redefinition of $\varrho_{\xi,2}$ (equivalently $T_\xi[2]$). Therefore, with a new $T_\xi^{\text{new}}[2]$ we can rewrite (171) as:

$$\left[ T_\mu[1], T_\nu, [1] \right] = f_{\mu\nu}{}^\xi T_\xi^{\text{new}}[2] + \hbar^2 (I_1 + I_2 + I_3) Q_{\mu\nu}, \tag{176}$$

where we have also defined:

$$Q_{\mu\nu} := 2 f_{[\mu}{}^{\xi o} f_\xi{}^{\pi\rho} f_{\nu]\pi}{}^\sigma \left\{ T_o[0], T_\rho[0], T_\sigma[0] \right\}. \tag{177}$$

The reason why we have postponed presenting the evaluations of the individual integrals $I_1$, $I_2$, and $I_3$ is that we don't need their individual values, only the sum, and precisely this sum was evaluated in eq. (E.23) of [19] with the result:

$$I_1 + I_2 + I_3 = \frac{1}{12}. \tag{178}$$

We can therefore write (ignoring the "new" label on $T_\xi[2]$):

$$\left[ T_\mu[1], T_\nu[1] \right] = f_{\mu\nu}{}^\xi T_\xi[2] + \frac{\hbar^2}{12} Q_{\mu\nu}. \tag{179}$$

This is the relation for the Yangian that was presented in §8.6 of [19] and how to relate this to other standard relations of the Yangian was also discussed there. This is also the exact relation we found in the boundary theory (c.f. (97)). Note furthermore that, if we had used the relation between our algebra and anomaly (135) to derive the algebra Lie bracket, we would have arrived at precisely the same conclusion, as the second term in (179) is indeed the anomaly of a Wilson line (c.f. eq. (8.35) of [32]).

Thus we see that the algebra $\mathcal{A}^{\text{Sc}}(\mathcal{T}_{\text{bk}})$, defined in (41), at 2-loops, is the Yangian $Y_\hbar(\mathfrak{gl}_K)$:

$$\mathcal{A}^{\text{Sc}}(\mathcal{T}_{\text{bk}})/\hbar^3 \stackrel{N \to \infty}{\cong} Y_\hbar(\mathfrak{gl}_K)/\hbar^3. \tag{180}$$

The two loop result in the BF theory was exact. The above two loop result is exact as well. Though we do not prove this by computing Witten diagrams, we can argue using the form of the algebra in terms of anomaly (135). In [19] it was shown that there are no anomalies beyond two loops. This concludes our second proof of Proposition 2.[44]

---

[44] The first one, which is significantly more abstract, being in Appendix B.

# 6 Physical String Theory Construction of The Duality

The topological theories we have considered so far can be constructed from a certain brane setup in type IIB string theory and then applying a twist and an $\Omega$-deformation. This brane construction will show that the algebras we have constructed are in fact certain supersymmetric subsectors of the well studied $\mathcal{N} = 4$ SYM theory with defect and its holographic dual. We note that the idea of embedding Chern-Simons type theories inside 1 and 2 higher dimensional supersymmetric gauge theories as (quasi)-topological subsectors can be traced back to [46].

**A Caveat.** The most transparent way of probing (quasi) topological subsectors of the relevant physical (defect) AdS/CFT correspondence would be to apply twist and deformation to 4d $\mathcal{N} = 4$ SYM with a domain wall on one hand and to AdS$_5 \times S^5$ supergravity with D5-brane probes on the other hand. Localization in the AdS background is yet to be developed and this is *not* what we do in this section. We apply twist and deform the gauge theories that appear in a certain D3-D5 brane configuration in flat background and argue that we end up with the topological brane setup of section 3.1. Thus in making the claim that our topological holography embeds into physical holography we are relying on the assumption that the process of twist and deformation commutes with taking the decoupling limit.

We describe our construction below.

## 6.1 Brane Configuration

Our starting brane configuration involves a stack of $N$ D3 branes and $K$ D5 branes in type IIB string theory on a 10d target space of the form $\mathbb{R}^8 \times C$ where $C$ is a complex curve which we take to be just the complex plane $\mathbb{C}$. The D5 branes wrap $\mathbb{R}^4 \times C$ and the D3 branes wrap an $\mathbb{R}^4$ which has a 3d intersection with the D5 branes. Let us express the brane configuraiton by the following table:

| | 0 | 1 | 2 | 3 | 4 | 5 | 6 | 7 | 8 | 9 |
|----|---|---|---|---|---|---|---|---|---|---|
| | | $\mathbb{R}^4$ | | | $C$ | | | $\mathbb{R}^4$ | | |
| $D5$ | $\times$ | $\times$ | $\times$ | $\times$ | $\times$ | $\times$ | | | | |
| $D3$ | $\times$ | | $\times$ | $\times$ | | | | $\times$ | | |

(181)

The world-volume theory on the D5 branes is the 6d $\mathcal{N} = (1,1)$ SYM theory coupled to a 3d defect preserving half of the supersymmetry. Similarly, the world-volume theory on the D3 branes is the 4d SYM theory coupled to a 3d defect preserving half of the supersymmetry. To this setup we apply a particular twist, i.e., we choose a nilpotent supercharge and consider its cohomology.

## 6.2 Twisting Supercharge

### 6.2.1 From the 6d Perspective

We use $\Gamma_i$ with $i \in \{0, \cdots, 9\}$ for 10d Euclidean gamma matrices. We also use the notation:

$$\Gamma_{i_1 \cdots i_n} := \Gamma_{i_1} \cdots \Gamma_{i_n}. \tag{182}$$

Type IIB has 32 supercharges, arranged into two Weyl spinors of the same 10 dimensional chirality – let us denote them as $Q_l$ and $Q_r$. A general linear combination of them is written as $\epsilon_L Q_l + \epsilon_R Q_r$ where $\epsilon_L$ and $\epsilon_R$ are chiral spinors parameterizing the supercharge. The chirality constraints on them are:

$$i\Gamma_{0\ldots9}\epsilon_L = \epsilon_L, \qquad i\Gamma_{0\ldots9}\epsilon_R = \epsilon_R. \tag{183}$$

We shall discuss constraints on the supercharge by describing them as constraints on the parameterizing spinors.

The supercharges preserved by the D5 branes are constrained by:

$$\epsilon_R = i\Gamma_{012345}\epsilon_L \, . \tag{184}$$

This reduces the number of supercharges to 16. The D3 branes imposes the further constraint:

$$\epsilon_R = i\Gamma_{0237}\epsilon_L \, . \tag{185}$$

This reduces the number of supercharges by half once more. Therefore the defect preserves just 8 supercharges. Since $\epsilon_R$ is completely determined given $\epsilon_L$, in what follows we refer to our choice of supercharge simply by referring to $\epsilon_L$.

We want to perform a twist that makes the D5 world-volume theory topological along $\mathbb{R}^4$ and holomorphic along $C$. This twist was described in [37]. Let us give names to the two factors of $\mathbb{R}^4$ in the 10d space-time:

$$M := \mathbb{R}^4_{0123}, \qquad M' := \mathbb{R}^4_{6789} \, . \tag{186}$$

The spinors in the 6d theory transform as representations of Spin(6) under space-time rotations. $\mathcal{N} = (1,1)$ algebra has two left handed spinors and two right handed spinors transforming as $\mathbf{4}_l$ and $\mathbf{4}_r$ respectively. There are two of each chirality because the R-symmetry is $\mathrm{Sp}(1) \times \mathrm{Sp}(1) = \mathrm{Spin}(4)_{M'}$ such that the two left handed spinors transform as a doublet of one Sp(1) and the two right handed spinors transform as a doublet of the other Sp(1). The subgroup of Spin(6) preserving the product structure $\mathbb{R}^4 \times C$ is $\mathrm{Spin}(4)_M \times \mathrm{U}(1)$. Under this subgroup $\mathbf{4}_l$ and $\mathbf{4}_r$ transform as $(\mathbf{2},\mathbf{1})_{-1} \oplus (\mathbf{1},\mathbf{2})_{+1}$ and $(\mathbf{2},\mathbf{1})_{+1} \oplus (\mathbf{1},\mathbf{2})_{-1}$ respectively, where the subscripts denote the U(1) charges. Rotations along $M'$ act as R-symmetry on the spinors – the spinors transform as representations of $\mathrm{Spin}(4)_{M'}$ such that $\mathbf{4}_+$ transforms as $(\mathbf{2},\mathbf{1})$ and $\mathbf{4}_-$ transforms as $(\mathbf{1},\mathbf{2})$. In total, under the symmetry group $\mathrm{Spin}(4)_M \times \mathrm{U}(1) \times \mathrm{Spin}(4)_{M'}$ the 16 supercharges of the 6d theory transform as:

$$\left((\mathbf{2},\mathbf{1})_{-1} \oplus (\mathbf{1},\mathbf{2})_{+1}\right) \otimes (\mathbf{2},\mathbf{1}) \oplus \left((\mathbf{2},\mathbf{1})_{+1} \oplus (\mathbf{1},\mathbf{2})_{-1}\right) \otimes (\mathbf{1},\mathbf{2}) \, . \tag{187}$$

The twist we seek is performed by redefining the the space-time isometry:

$$\mathrm{Spin}(4)_M \rightsquigarrow \mathrm{Spin}(4)_M^{\mathrm{new}} \subseteq \mathrm{Spin}(4)_M \times \mathrm{Spin}(4)_{M'} \, , \tag{188}$$

where the subgroup $\mathrm{Spin}(4)_M^{\mathrm{new}}$ of $\mathrm{Spin}(4)_M \times \mathrm{Spin}(4)_{M'}$ consists of elements $(x, \theta(x))$ which is defined by the isomorphism $\theta : \mathrm{Spin}(4)_M \xrightarrow{\sim} \mathrm{Spin}(4)_{M'}$. More, explicitly, the isomorphism acts as:

$$\theta(\Gamma_{\mu\nu}) = \Gamma_{\mu+6,\nu+6} \, , \qquad \mu, \nu \in \{0,1,2,3\} \, . \tag{189}$$

The generators of the new $\mathrm{Spin}(4)_M^{\mathrm{new}}$ are then:

$$\Gamma_{\mu\nu} + \Gamma_{\mu+6,\nu+6} \, . \tag{190}$$

After this redefinition, the symmetry $\mathrm{Spin}(4)_M \times \mathrm{U}(1) \times \mathrm{Spin}(4)_{M'}$ of the 6d theory reduces to $\mathrm{Spin}(4)_M^{\mathrm{new}} \times \mathrm{U}(1)$ and under this group the representation (187) of the supercharges becomes:

$$2(\mathbf{1},\mathbf{1})_{-1} \oplus (\mathbf{3},\mathbf{1})_{-1} \oplus (\mathbf{1},\mathbf{3})_{-1} \oplus 2(\mathbf{2},\mathbf{2})_{+1} \, . \tag{191}$$

We thus have two supercharges that are scalars along $M$, both of them have charge $-1$ under the U(1) rotation along $C$. We take the generator of this rotation to be $-i\Gamma_{45}$, then if $\epsilon$ is one of the scalar (on $M$) supercharges that means:

$$i\Gamma_{45}\epsilon = \epsilon \, . \tag{192}$$

We identify the supercharge $\epsilon$ by imposing invariance under the new rotation generators on $M$, namely (190):

$$(\Gamma_{\mu\nu} + \Gamma_{\mu+6,\nu+6})\epsilon = 0 \,. \tag{193}$$

The constraints (184) and (185) put by the D-branes and the U(1)-charge on $C$ (192) together are equivalent to the following four independent constraints:

$$i\Gamma_{\mu,\mu+6}\epsilon = \epsilon \,, \qquad \mu\{0,1,2,3\} \,. \tag{194}$$

Together with the chirality constraint (183) in 10d we therefore have 5 equations, each reducing the number degrees of freedom by half. Since a Dirac spinor in 10d has 32 degrees of freedom, we are left with $32 \times 2^{-5} = 1$ degree of freedom, i.e., we have a unique supercharge,[45] which we call $Q$. It was shown in [37] that the supercharge $Q$ is nilpotent:

$$Q^2 = 0 \,, \tag{195}$$

and the 6d theory twisted by this $Q$ is topological along $M$ – which is simply a consequence of (193) – and it is holomorphic along $C$. The latter claim follows from the fact that there is another supercharge in the 2d space of scalar (on $M$) supercharges in the 6d theory, let's call it $Q'$, which has the following commutator with $Q$:

$$\{Q, Q'\} = \partial_{\bar{z}} \,, \tag{196}$$

where $z = \frac{1}{2}(x^4 - ix^5)$ is the holomorphic coordinate on $C$. This shows that $\bar{z}$-dependence is trivial ($Q$-exact) in the $Q$-cohomology.

### 6.2.2 From the 4d Perspective

What is new in our setup compared to the setup considered in [37] is the stack of D3 branes. We can figure out what happens to the world-volume theory of the D3 branes – we get the *Kapustin-Witten (KW) twist* [47], as we now show. The equations (194) can be used to to get the following six (three of which are independent) equations:

$$\begin{aligned}
(\Gamma_{02} + \Gamma_{68})\epsilon = 0 \,, \qquad (\Gamma_{03} + \Gamma_{69})\epsilon = 0 \,, \qquad (\Gamma_{23} + \Gamma_{89})\epsilon = 0 \,, \\
(\Gamma_{07} + \Gamma_{16})\epsilon = 0 \,, \qquad (\Gamma_{27} + \Gamma_{18})\epsilon = 0 \,, \qquad (\Gamma_{37} + \Gamma_{19})\epsilon = 0 \,.
\end{aligned} \tag{197}$$

These are in fact the equations that defines a scalar supercharge in the KW twist of $\mathcal{N} = 4$ theory on $\mathbb{R}^4_{0237}$ for a particular homomorphism from space-time ismoetry to the R-symmetry.[46] Space-time isometry of the theory on $\mathbb{R}^4_{0237}$ acts on the spinors as $\mathrm{Spin}(4)_{\mathrm{iso}}$, generated by the six generators:

$$\Gamma_{\mu\nu} \,, \qquad \mu, \nu \in \{0, 2, 3, 7\} \text{ and } \mu \neq \nu \,. \tag{198}$$

Rotations along the transverse directions act as R-symmetry, which is $\mathrm{Spin}(6)$, though the subgroup of the R-symmetry preserving the product structure $C \times \mathbb{R}^4_{1689}$ is $\mathrm{U}(1) \times \mathrm{Spin}(4)_R$. The KW twist is constructed by redefining space-time isometry to be a $\mathrm{Spin}(4)$ subgroup of $\mathrm{Spin}(4)_{\mathrm{iso}} \times \mathrm{Spin}(4)_R$ consisting of elements $(x, \vartheta(x))$ where $\vartheta : \mathrm{Spin}(4)_{\mathrm{iso}} \xrightarrow{\sim} \mathrm{Spin}(4)_R$ is an isomorphism. The particular isomorphism that leads to the equations (197) is:

$$\begin{aligned}
\Gamma_{02} \mapsto \Gamma_{68} \,, \qquad \Gamma_{03} \mapsto \Gamma_{69} \,, \qquad \Gamma_{23} \mapsto \Gamma_{89} \,, \\
\Gamma_{07} \mapsto \Gamma_{16} \,, \qquad \Gamma_{27} \mapsto \Gamma_{18} \,, \qquad \Gamma_{37} \mapsto \Gamma_{19} \,.
\end{aligned} \tag{199}$$

---

[45]Note that without using the constraint put by the D3 branes we would get *two* supercharges that are scalars on $M$, i.e., there are two superhcarges in the 6d theory (by itself) that are scalars on $M$.

[46]Note that we are using subscripts simply to refer to particular directions.

*Remark* 7 (A member of a $\mathbb{CP}^1$ family of twists). In [47] it was shown that there is a family of KW twists parameterized by $\mathbb{CP}^1$. The unique twist (by the supercharge $Q$) we have found is a specific member of this family. Let us identify which member that is.

The $\mathbb{CP}^1$ family comes from the fact that there is a 2d space of scalar (on $M$) supercharges (in (191)) in the twisted theory.[47] Also note from the original representation of the spinors (187) that the two scalar supercharges come from spinors transforming as $(\mathbf{1}, \mathbf{2})$ and $(\mathbf{2}, \mathbf{1})$ under the original isometry Spin(4)$^{\text{old}}$.[48] Let us choose two Spin(4)$^{\text{new}}$ scalar spinors with opposite Spin(4)$^{\text{old}}$ chiralities and call them $\epsilon_l$ and $\epsilon_r$. The Spin(4)$^{\text{old}}$ chirality operator is $\Gamma^{\text{old}} := \Gamma_{0237}$. Let us choose $\epsilon_l$ and $\epsilon_r$ in such a way that they are related by the following equation:

$$\epsilon_r = N\epsilon_l \quad \text{where} \quad N = \frac{1}{4}(\Gamma_{06} + \Gamma_{28} + \Gamma_{39} + \Gamma_{17}). \tag{200}$$

This relation is consistent with the spinors being Spin(4)$^{\text{new}}$ invariant because $N$ anti-commutes with Spin(4)$^{\text{new}}$ (thus invariant spinors are still invariant after being operated with $N$), as well as with $\Gamma^{\text{old}}$ (changing Spin(4)$^{\text{old}}$ chirality). An arbitrary scalar supercharge in the twisted theory is a complex linear combination of $\epsilon_l$ and $\epsilon_r$, such as $\alpha\epsilon_l + \beta\epsilon_r$, however, since the overall normalization of the spinor is irrelevant, the true parameter identifying a spinor is the ratio $t := \beta/\alpha \in \mathbb{CP}^1$. Furthermore, due to the equations (197), $N^2$ acts as $-1$ on any Spin(4)$^{\text{new}}$ scalar, leading to:

$$\epsilon_l = -N\epsilon_r. \tag{201}$$

To see the value of the twisting parameter $t$ for the supercharge identified by the equations (194) (in addition to the 10d chirality (183)), we first pick a linear combination $\epsilon := \epsilon_l + t\epsilon_r$ with $t \in \mathbb{CP}^1$. Then using (201) and (194) we get:

$$-i\epsilon = N\epsilon = \epsilon_r - t\epsilon_l, \tag{202}$$

where the first equality follows from (194) and the second from (201). Equating the two sides we find the twisting parameter:

$$t = i. \tag{203}$$

$\triangle$

### 6.2.3 From the 3d Perspective

Finally, at the 3 dimensional D3-D5 intersection lives a 3d $\mathcal{N} = 4$ theory consisting of bifundamental hypermultiplets coupled to background gauge fields which are restrictions of the gauge fields from the D3 and the D5 branes [48]. Considering $Q$-cohomology for the 3d theory reduces it to a topological theory as well. To identify the topological 3d theory we note that for the twisting parameter $t = i$, the 4d theory is an analogue of a 2d B-model[49] [47] and this can be coupled to a 3d analogue of the 2d B-model[50] – a B-type topological twist of 3d $\mathcal{N} = 4$ is called a *Rozansky-Witten (RW)* twist [49]. The flavor symmetry of the theory is $\text{U}(N) \times \text{U}(K)$ which acts on the hypers and is gauged by the background connections.

We can reach the same conclusion by analyzing the constraints on the twisting supercharge viewed from the 3d point of view. The bosonic symmetry of the 3d theory includes

---

[47]Though we began the discussion with a view to identifying topological-holomorphic twist of 6d $\mathcal{N} = (1, 1)$ theory, what we found in the process in particular are supercharges that are scalars on $M$. If we forget that we had a 6d theory on $M \times C$ and just consider a theory on $M$ with rotations on $C$ being part of the R-symmetry then, first of all, we find a $\mathcal{N} = 4$ SYM theory on $M$ and the twist we described is precisely the KW twist.

[48]We are writing Spin(4)$^{\text{old}}$ instead of Spin(4)$_M$ since the support of the 4d theory is not $M \equiv \mathbb{R}^4_{0123}$ but $\mathbb{R}^4_{0237}$.

[49]In particular, the 4d Theory on $\mathbb{R}^2 \times T^2$ can be compactified on the two-torus $T^2$ to get a B-model on $\mathbb{R}^2$.

[50]We want to be able to take the 3d theory on $\mathbb{R}^2 \times S^1$ and compactify it on $S^1$ to get a B-model on $\mathbb{R}^2$. If we have a 4d theory on $\mathbb{R}^2 \times T^2$ coupled to a 3d theory on $\mathbb{R}^2 \times S^1$, compactifying on $T^2$ should not make the two systems incompatible.

$\text{SU}(2)_{\text{iso}} \times \text{SU}(2)_H \times \text{SU}(2)_C$ where $\text{SU}(2)_{\text{iso}}$ is the isometry of the space-time $\mathbb{R}^3_{023}$, $\text{SU}(2)_C$ are rotations in $\mathbb{R}^3_{689}$, and $\text{SU}(2)_H$ are rotations in $\mathbb{R}^3_{145}$. The hypers in the 3d theory come from strings with one one end attached to the D5 branes and another end attached to the D3 branes. Rotations in $\mathbb{R}^3_{145}$ – the R-symmetry $\text{SU}(2)_H$ – therefore act on the hypers. This means that $\text{SU}(2)_H$ acts on the Higgs branch of the 3d theory. This leaves the other R-symmetry group $\text{SU}(2)_C$ which would act on the coulomb branch of the theory if the theory had some dynamical 3d vector multiplets. We now note that the topological twist, from the 3d perspective, involves twisting the isometry $\text{SU}(2)_{\text{iso}}$ with the R-symmetry group $\text{SU}(2)_C$, as evidenced explicitly by the three equations in the first line of (197). This particular topological twist (as opposed to the topological twist using the other R-symmetry $\text{SU}(2)_H$) of 3d $\mathcal{N} = 4$ is indeed the RW twist [50].

To summerize, taking cohomology with respect to the supercharge $Q$ leaves us with the KW twist (twisting parameter $t = i$) of $\mathcal{N} = 4$ SYM theory on $\mathbb{R}^4$ with gauge group $\text{U}(N)$ and a topological-holomorphic twist of $\mathcal{N} = (1, 1)$ theory on $\mathbb{R}^4 \times C$ with gauge group $\text{U}(K)$, and these two theories are coupled via a 3d RW theory of bifundamental hypers with flavor symmetry $\text{U}(N) \times \text{U}(K)$ gauged by background connections.[51] Note that we have not described the effect of the twist on the closed string theory. This is because we are assuming a decoupling between the closed string modes and the D5-defect modes in the large $N$ limit (referred to as rigid holography in [35]) and therefore, the operator algebra that we will concern ourselves with will be insensitive to the closed string modes.[52] We will ignore the closed string modes moving forward as well.

## 6.3 Omega Deformation

We start by noting that the dimensional reduction of the topological-holomorphic 6d theory from $\mathbb{R}^4 \times C$ to $\mathbb{R}^4$ reduces it to the KW twist of $\mathcal{N} = 4$ SYM on the $\mathbb{R}^4$.[53] This observation allows us to readily tailor the results obtained in [37] about $\Omega$-deformation of the 6d theory to the case of $\Omega$-deformation of 4d KW theory.

The fundamental bosonic field in the 10d $\mathcal{N} = 1$ SYM theory is the connection $A_I$ where $I \in \{0, \cdots, 9\}$. When dimensionally reduced to 6d, this becomes a 6d connection $A_M$ with $M \in \{0, \cdots, 5\}$ and four scalar fields $\phi_0, \phi_1, \phi_2,$ and $\phi_3$ which are just the remaining four components of the 10d connection. The $\text{Spin}(4)_M$ space-time isometry acts on the first four components of the connection, namely $A_0, A_1, A_2,$ and $A_3$ via the vector representation. The four scalars – $\phi_0, \phi_1, \phi_2,$ and $\phi_3$ – transform under the vector representation of the R-symmetry $\text{Spin}(4)_{M'}$. Once twisted according to (188), only the diagonal subgroup $\text{Spin}(4)_M^{\text{new}}$ of $\text{Spin}(4)_M \times \text{Spin}(4)_{M'}$ acts on the fields, under which the first four components of the connection and the four scalars transform in the same way – apart from the inhomogeneous transformation of the connection – and therefore we can package them together into one complex valued gauge field:

$$\mathcal{A}_\mu := A_\mu + i\phi_\mu, \qquad \mu \in \{0, 1, 2, 3\}. \tag{204}$$

We also write the remaining components of the connection in complex coordinates on $C$:

$$A_z := A_4 + iA_5 \quad \text{and} \quad A_{\bar{z}} := A_4 - iA_5. \tag{205}$$

It was shown in [37] that this topological-holomorphic 6d theory can be viewed as a 2d gauged B-model on $\mathbb{R}^2_{23}$ where the fields are valued in maps $\text{Map}(\mathbb{R}^2_{01} \times \mathbb{C}, \mathfrak{gl}_K)$. This is a

---

[51]Though it is customary to decouple the central $\text{U}(1)$ subgroup from the gauge groups as it doesn't interact with the non-abelian part, our computations look somewhat simpler if we keep the $\text{U}(1)$.

[52]This is the same argument we used in §3.2.

[53]Both the 6d $\mathcal{N} = (1, 1)$ SYM and the 4d $\mathcal{N} = 4$ SYM are dimensional reductions of the 10d $\mathcal{N} = 1$ SYM and dimensional reduction commutes with the twisting procedure.

vector space which plays the role of the Lie algebra of the 2d gauge theory. From the 2d point of view $\mathcal{A}_2$ and $\mathcal{A}_3$ are part of a connection on $\mathbb{R}^2_{23}$ and there are four chiral multiplets with the bottom components $\mathcal{A}_0, \mathcal{A}_1, A_z$, and $A_{\bar{z}}$. The 2d theory consists of a superpotential which is a holomorphic function of these chiral multiplets – the superpotential can be written conveniently in terms of a one form $\widetilde{\mathcal{A}} := \mathcal{A}_0 dx^0 + \mathcal{A}_1 dx^1 + A_z dz + A_{\bar{z}} d\bar{z}$ on $\mathbb{R}^2_{01} \times C$ consisting of these chiral fields:[54]

$$W(\mathcal{A}_0, \mathcal{A}_1, A_z, A_{\bar{z}}) = \int_{\mathbb{R}^2_{01} \times C} dz \wedge \text{tr}\left( \widetilde{\mathcal{A}} \wedge d\widetilde{\mathcal{A}} + \frac{2}{3} \widetilde{\mathcal{A}} \wedge \widetilde{\mathcal{A}} \wedge \widetilde{\mathcal{A}} \right). \tag{206}$$

The superpotential is the action functional of a 4d CS theory on $\mathbb{R}^2_{01} \times C$ for the connection $\widetilde{\mathcal{A}}$.

One of the results of [37] is the following: $\Omega$-deformation[55] applied to this topological-holomorphic 6d theory with respect to rotation on $\mathbb{R}^2_{23}$ reduces the theory to a 4d CS theory on $\mathbb{R}^2_{01} \times C$ with *complexified* gauge group $\text{GL}_K$.

The twisted 4d theory (the D3 world-volume theory) wraps the plane $\mathbb{R}^2_{23}$ as well and therefore is affected by the $\Omega$-deformation. By noting that the 4d theory is a dimensional redcution of the 6d theory from $\mathbb{R}^4 \times C$ to $\mathbb{R}^4$ and assuming that $\Omega$-deformation commutes with dimensional reduction,[56] we can deduce what the $\Omega$-deformed version of the twisted 4d theory is. This will be a 2d gauge theory with complexified gauge group $\text{GL}_N$ and the action will be the dimensional reduction of the 4d CS action (206) from $\mathbb{R}^2 \times C$ to $\mathbb{R}^2$ – this is the 2d BF theory where $A_{\bar{z}}$ plays the role of the $B$ field:

$$\int_{\mathbb{R}^2 \times C} dz \wedge \text{CS}(A_{\mathbb{R}^2 \times C}) \xrightarrow{\text{Reduce on } C} \int_{\mathbb{R}^2} \text{tr} A_{\bar{z}}\left( dA_{\mathbb{R}^2} + \frac{1}{2} A_{\mathbb{R}^2} \wedge A_{\mathbb{R}^2} \right)$$
$$= \int_{\mathbb{R}^2} \text{tr} A_{\bar{z}} F(A_{\mathbb{R}^2}), \tag{207}$$

where, as before, $\bar{z}$ is the anti-holomorphic coordinate on $C$.

Finally, it was shown in [52] that the RW twist of a 3d $\mathcal{N} = 4$ theory on $\mathbb{R}^2_\Omega \times \mathbb{R}$ with only hypers reduces, upon $\Omega$-deformation with respect to rotation in the plane $\mathbb{R}^2_\Omega$, to a free quantum mechanics on $\mathbb{R}$. A slight modification of this result, involving background connections gauging the flavor symmetry of the hypers leads to the result that the omega deformed theory is a gauged quantum mechanics, the kind of theory we have considered on the defect in the 2d BF theory.[57]

## 6.4 Takeaway from the Brane Construction

Via supersymmetric twists and $\Omega$-deformation, we have made contact with precisely the setup we have considered in this paper. We have a 4d CS theory with gauge group $\text{GL}_K$ and a 2d BF theory with gauge group $\text{GL}_N$ and they intersect along a topological line supporting a gauged quantum mechanics with $\text{GL}_K \times \text{GL}_N$ symmetry. We thus claim that the topological holographic duality that we have established in this paper is indeed a topological subsector of the standard holographic duality involving defect $\mathcal{N} = 4$ SYM.

---

[54]Up to some overall numerical factors.

[55]Introduced for the first time in [51] in the context of 4d $\mathcal{N} = 2$ gauge theories on $\mathbb{R}^2 \times \mathbb{R}^2$. The relevant space-time rotation in that case was a $U(1) \times U(1)$ action rotating the two planes – which ultimately localized the 4d theory to a 0d matrix model. Analogously, $\Omega$-deformation with respect to rotation on a plane localizes our 6d/4d/3d theory to a 4d/2d/1d theory.

[56]Alternatively, one can redo the localization computations of [37] for the 4d case, confirming that $\Omega$-deformation does indeed commute with dimensional reduction.

[57]The bosonic version, which leads to the same Yangian with minor modifications to the computations as remarked in 3, 4, and 5.

# 7 Concluding Remarks and Future Works

In the previous sections we have been able to exactly (at all loops) match a subsector of the operator algebra in the 2d BF theory with a line defect, with a subsector of the scattering algebra in a 3d closed string theory with a surface defect. The subsectors of operators we focused on are restricted to the defects on both sides of the duality. This matching provides a non-trivial check of the proposed holographic duality. Furthermore, we have shown that this holographic duality between topological/holomorphic theories is in fact a supersymmetric subsector of the more familiar $AdS_5/CFT_4$ duality. From the considerations of this paper several immediate questions and new directions arise that we have not yet addressed. Let us comment on a few such issues that we think are interesting topics to pursue for future research.

**Central extensions on two sides of the duality:** To ease computation we restricted our attention to the quotients of the full operator algebra and scattering algebra by their centers. The inclusion of the central operators will change the associative structure of the algebras. A stronger statement of duality will be to compare the centrally extended Yangians coming from the boundary and the bulk theory.

**Brane probes:** Using branes in the bulk to probe local operators in the boundary theory has been a useful tool [53, 54]. In our setup, a brane must be *Lagrangian* in the A-twisted $\mathbb{R}^4$ directions. Looking at the brane setup (8) (which we reproduce in (209) for convenience) we see that the real directions of the D2 and D4 branes are Lagrangian with respect to the following symplectic form:

$$\mathrm{d}v \wedge \mathrm{d}x + \mathrm{d}w \wedge \mathrm{d}y \,. \tag{208}$$

This leaves the possibility of two more different embeddings for D2-branes:

|        | $\mathbb{R}_v$ | $\mathbb{R}_w$ | $\mathbb{R}_x$ | $\mathbb{R}_y$ | $\mathbb{C}_z$ |
|--------|------|------|------|------|------|
| D2     | 0    | ×    | ×    | 0    | 0    |
| D4     | 0    | 0    | ×    | ×    | ×    |
| D2′    | ×    | 0    | 0    | ×    | $z$  |
| D2″    | ×    | ×    | 0    | 0    | $z$  |

$\tag{209}$

The D2′-branes are Wilson lines in the CS theory on the D4-branes perpendicular to the original Wilson line at thte D2-D4 intersection. Such crossing Wilson lines were studied in [19, 44] with the result that this corssing (of two Wilson lines carrying representations $U$ and $V$ of $\mathfrak{gl}_K$ respectively) inserts an operator $T_{VU}(z) : U \otimes V \to V \otimes U$ in the CS theory which solves the Yang-Baxter equation, which is described more easily with diagrams:

$$\tag{210}$$

where $z_1$, $z_2$, and $z_3$ are the spectral parameters (location in the complex plane) of the lines carrying representations $V$, $U$, and $W$ respectively, and $z_{21} := z_2 - z_1$ and so on. Solutions of the above equation are closely tied to Quantum Groups. The operators $T_{UV}(z)$, which are commonly referred to as $R$-matrices, can be explicitly constructed using Feynman diagrams [19]. When the complex directions of the theory are parameterized by $\mathbb{C}$ (as in our case), these $R$-matrices are rational functions of $z$. If we choose $U$ and $W$ to be the fundamental

representation of $\mathfrak{gl}_K$, then by providing an incoming and an outgoing fundamental state, we can view $\langle j|T_{\mathbf{K}V}(z)|i\rangle$ as a map $\mathsf{T}_j^i(z): V \to V$ which has an expansion is $z^{-1}$:

$$\mathsf{T}_j^i(z) = \mathrm{id}_V\,\delta_j^i - \hbar \sum_{n\geq 0}\left(-z^{-1}\right)^{n+1} T_j^i[n]\,, \tag{211}$$

where the $T_j^i[n]$ are precisely the operators that generate the scattering algebra $\mathcal{A}^{\mathrm{Sc}}(\mathcal{T}_{\mathrm{bk}})$ (see (39) and (41)). This suggests that in the dual picture we should be able to interpret the D2$'$ branes as a generating function for the operators $O_j^i[n]$.

The interpretations of the D2$''$ branes are missing on both sides of the duality.

**Finite $N$ duality:** We considered the large $N$ limit to decouple the closed string modes from the defect (CS) mode in the bulk side of the duality and to eliminate any relations among our operators that would arise from having finite dimensional matrices (see §4.3 and §5.4). It would of course be a stronger check if we could match the algebras at finite $N$, when they can be quotients of the Yangian by some extra relations.

**Duality for other quantum groups:** In [19, 44] it was shown that by replacing our complex direction $\mathbb{C}$ with the punctured plane $\mathbb{C}^\times$ or an elliptic curve, we can get, instead of the Yangian, the trigonometric or elliptic solutions to the Yang-Baxter equation (210). It will be interesting to have an analogous analysis of holographic duality for the corresponding quantum groups as well.

## Acknowledgements

We are grateful to Kevin Costello, Davide Gaiotto, Jaume Gomis, Shota Komatsu, Natalie Paquette, and Masahito Yamazaki for valuable discussions and feedbacks on the manuscript. We specially thank Kevin Costello, whose works and ideas have directly motivated and guided this project.

**Funding information**    All authors are supported by Perimeter Institute for Theoretical Physics. Research at Perimeter Institute is supported by the Government of Canada through Industry Canada and by the Province of Ontario through the Ministry of Research and Innovation.

## A  Integrating the BF interaction vertex

In this appendix we evaluate the integrals in (80).

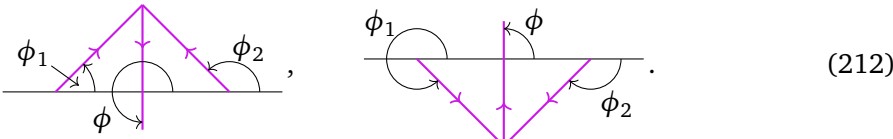

$$\tag{212}$$

We split up each integral into two, based on whether the bulk point is above or below the line operator. We use angular coordinates defined as in the above diagrams. One subtlety is that, from the definition of the propagators in the Cartesian coordinate we can see that the integrand (including the measure) is even under reflection with respect to the line. So, we just have to make sure that when we divide up the integral in the aforementioned way, even when written in angular coordinates, the integrand does not change sign under reflection. With this

in mind, the integrals we have to evaluate are:

$$\mathcal{V}^{\alpha\beta\gamma}_{\cdot||}(x_1, x_2) = \frac{\hbar^2}{(2\pi)^3} f^{\alpha\beta\gamma} \int_0^{2\pi} d\phi_1 \int_{\phi_1}^{\pi} d\phi_2 \left( \int_{\pi}^{\phi_1+\pi} d\phi + \int_{\pi}^{\phi_1-\pi} d\phi \right),$$

$$\mathcal{V}^{\alpha\beta\gamma}_{|\cdot|}(x_1, x_2) = \frac{\hbar^2}{(2\pi)^3} f^{\alpha\beta\gamma} \int_0^{2\pi} d\phi_1 \int_{\phi_1}^{\pi} d\phi_2 \left( \int_{\phi_1+\pi}^{\phi_2+\pi} d\phi + \int_{\phi_1-\pi}^{\phi_2-\pi} d\phi \right),$$

$$\mathcal{V}^{\alpha\beta\gamma}_{||\cdot}(x_1, x_2) = \frac{\hbar^2}{(2\pi)^3} f^{\alpha\beta\gamma} \int_0^{2\pi} d\phi_1 \int_{\phi_1}^{\pi} d\phi_2 \left( \int_{\phi_2+\pi}^{2\pi} d\phi + \int_{\phi_2-\pi}^{0} d\phi \right).$$

All three terms are equal to $\frac{\hbar^2}{24} f^{\alpha\beta\gamma}$.

## B Yangian from 1-loop Computations

At the end of §5.3.1, by computing 1-loop diagrams, we concluded that quantum corrections deform the coalgebra structure of the classical Hopf algebra $U(\mathfrak{gl}_K[z])$. Since $\mathcal{A}^{\mathrm{Sc}}(\mathcal{T}_{\mathrm{bk}})$ is an algebra to begin with, we conclude that at one loop, we have a deformation of the classical algebra as a Hopf algebra. We are using the term "deformation" (alternatively, "quantization") in the sense of Definition 6.1.1 of [45], which essentially means that:

- $\mathcal{A}^{\mathrm{Sc}}(\mathcal{T}_{\mathrm{bk}})$ becomes the classical algebra $U(\mathfrak{gl}_K[z])$ in the classical limit $\hbar \to 0$.

- $\mathcal{A}^{\mathrm{Sc}}(\mathcal{T}_{\mathrm{bk}})$ is isomorphic to $U(\mathfrak{gl}_K[z])[\![\hbar]\!]$ as a $\mathbb{C}[\![\hbar]\!]$-module.

- $\mathcal{A}^{\mathrm{Sc}}(\mathcal{T}_{\mathrm{bk}})$ is a *topological* Hopf algebra (with respect to $\hbar$-adic topology).

The reason that we adhere to these conditions is that, there is a well known uniqueness theorem (Theorem 12.1.1 of [45]) which says that the Yangian is the unique deformation of $U(\mathfrak{gl}_K[z])$ in the above sense. Therefore, if we can show that our algebra $\mathcal{A}^{\mathrm{Sc}}(\mathcal{T}_{\mathrm{bk}})$ satisfies all these conditions and it is a nontrivial deformation of $U(\mathfrak{gl}_K)$ then we can conclude that it is the Yangian. From 1-loop computations we already know that it is a non-trivial deformation. That the first condition in the list above is satisfied is the content of Lemma 1. The second condition is satisfied because $\hbar$ acts on the generators of our algebra by simply multiplying the external propagators by $\hbar$ in the relevant Witten diagrams, this action does not distinguish between classical diagrams and higher loop diagrams. Satisfying the last condition is less trivial. While it seems known to people working in the field, we were unable to find a reference to cite, therefore, for the sake of completion, we provide a proof in this appendix, that the algebra $\mathcal{A}^{\mathrm{Sc}}(\mathcal{T}_{\mathrm{bk}})$ is indeed an ($\hbar$-adic) *topological* Hopf algebra.

We shall prove this by reconstructing the algebra $\mathcal{A}^{\mathrm{Sc}}(\mathcal{T}_{\mathrm{bk}})$ from its representations. As mentioned in §3.4, representations of this algebra are carried by Wilson lines, which form an abelian monoidal category. A morphism between two representations $V$ and $U$ in this category is constructed by computing the expectation value of two Wilson lines in representations $U$ and $V^{\vee}$ and providing a state at one end of each of the lines. For example, if $\varrho$ and $\varrho'$ are two homomorphisms from $\mathfrak{gl}_K$ to $\mathrm{End}(U)$ and $\mathrm{End}(V^{\vee})$ respectively, then for two lines $L$ and $L'$ in the topological plane of the CS theory and any $\psi \otimes \chi^{\vee} \in U \otimes V^{\vee}$, the expectation value $\langle W_{\varrho}(L) W_{\varrho'}(L') \rangle$ is valued in $\mathrm{End}(U) \otimes \mathrm{End}(V^{\vee})$ and plugging in states we find a morphism $\langle W_{\varrho}(L) W_{\varrho'}(L') \rangle (\psi \otimes \chi^{\vee}) : V \to U$.

Classically, these same Wilson lines carry representations of the classical algebra $U(\mathfrak{gl}_K[z])$. When viewed as representations of the deformed (alternatively, quantized) algebra $\mathcal{A}^{\mathrm{Sc}}(\mathcal{T}_{\mathrm{bk}})$, we shall call the category of Wilson lines the *quantized category* and viewed as representations

of $U(\mathfrak{gl}_K[z])$ we shall refer to the category as the *classical category*. Given any two Wilson lines $U$ and $V$, any non-trivial morphism between them in the quantized category is a quantization of a non-trivial morphism in the classical category. As we mentioned, a morphism between two Wilson lines is the expectation value of the lines provided with states at one end. A classical morphism is computed with classical diagrams and its quantization amounts to adding loop diagrams. A zero morphism is constructed by providing zero states, this is independent of quantization, i.e., a quantized morphism is zero, if the provided states are zero, but then so is the original classical morphism. There is in fact a one-to-one correspondence between morphisms between two lines in the classical category and the morphisms between the same lines in the quantized category.

For the sake of proof, let us abstract the information we have. We start with a $\mathbb{C}$-linear rigid abelian monoidal category $\mathcal{C} = \mathrm{Rep}_{\mathbb{C}}(H)$ which is the representation category of a Hopf algebra $H$. We then find a $\mathbb{C}[\![\hbar]\!]$-linear abelian monoidal category $\mathcal{C}_{\hbar}$, whose objects are representations of some, yet unknown, Hopf algebra $H_{\hbar}$, with the following properties:

- $\mathrm{ob}(\mathcal{C}_{\hbar}) = \mathrm{ob}(\mathcal{C})$,

- $\mathrm{Hom}_{\mathcal{C}_{\hbar}}(U, V) \cong \mathrm{Hom}_{\mathcal{C}}(U, V)[\![\hbar]\!]$ as $\mathbb{C}[\![\hbar]\!]$-modules.

Given this information we shall now prove that $H_{\hbar}$ is unique and that it is topological with respect to $\hbar$-adic topology. Then specializing to the case $H = U(\mathfrak{gl}_K[z])$ completes the proof of $\mathcal{A}^{\mathrm{Sc}}(\mathcal{T}_{\mathrm{bk}})$ being topological.

## B.1 Tannaka formalism

The aim of this formalism is to realize certain abelian rigid monoidal categories as the representation (or corepresentation) categories of Hopf algebras (possibly with extra structures). To avoid running into some subtlety in the beginning (we shall explain the subtlety later in this section), we first consider the reconstruction from the category of corepresentations.

**Reconstruction from corepresentation.** Let $k$ be a field, $\mathcal{C}$ an abelian (resp. abelian monoidal and $\mathrm{End}(1) = k$) category such that morphisms are $k$-bilinear, and let $R$ be a commutative algebra over $k$ – if there is an exact faithful (resp. monoidal) functor $\omega$ from $\mathcal{C}$ to $\mathrm{Mod}_f(R)$[58] such that the image of $\omega$ is inside the full subcategory $\mathrm{Proj}_f(R)$[59], then we shall say that $\mathcal{C}$ has a *fiber functor* $\omega$ to $\mathrm{Mod}_f(R)$.

**Theorem 2** (Tannaka Reconstruction for Coalgebra and Bialgebra). *With the notation above, if moreover $R$ is a local ring or a PID[60], then there exists a unique flat $R$-coalgebra (resp. $R$-bialgebra) $A$, up to unique isomorphism, such that $A$ represents the endomorphism of $\omega$ in the sense that $\forall M \in IndProj_f(R)$* [61]

$$Hom_R(A, M) \cong Nat(\omega, \omega \otimes M).$$

*Moreover, there is a functor $\phi : \mathcal{C} \to Corep_R(A)$ which makes the following diagram commutative*

$$
\begin{array}{ccc}
\mathcal{C} & \xrightarrow{\phi} & Corep_R(A) \\
& \searrow{\omega} & \downarrow{forget} \\
& & Mod_f(R)
\end{array}
$$

---

[58] finitely generated modules of $R$

[59] finitely generated projective modules of $R$

[60] PID=Principal Ideal Domain

[61] IndProj$_f(R)$ means category of inductive limit of finite projective $R$-modules, which is equivalent to category of flat $R$-modules.

*and $\phi$ is an equivalence if $R = k$.*

Our strategy in proving this theorem basically follows [55]. First of all, we need the following

**Lemma 3.** $\mathcal{C}$ *is both Noetherian and Artinian.*

*Proof.* Take $X \in \mathrm{ob}(\mathcal{C})$, and an ascending chain $X_i$ of subobjects of $X$, apply the functor $\omega$ to this chain, so that $\omega(X_i)$ is an ascending chain of finitely generated projective submodules of finitely generated projective module $\omega(X)$, thus there is an index $j$ such that $\mathrm{rank}(\omega(X_j)) = \mathrm{rank}(\omega(X))$. Now the quotient of $\omega(X)$ by $\omega(X_j)$ is $\omega(X/X_j)$, which is again finitely generated projective, so it has zero rank, hence trivial. Faithfulness of $\omega$ implies that $X/X_j$ is zero, i.e. $X = X_j$, so $\mathcal{C}$ is Noetherian. It follows similarly that $\mathcal{C}$ is Artinian as well. $\square$

Next, we define a functor

$$\otimes : \mathrm{Proj}_f(R) \times \mathcal{C} \to \mathcal{C}$$

by sending $(R^n, X)$ to $X^n$, recall that every finitely generated projective module over a local ring or a PID is free, thus isomorphic to $R^n$ for some $n$. Define $\underline{\mathrm{Hom}}(M, X)$ to be $M^\vee \otimes X$. For $V \subset M$ and $Y \subset X$, we define the transporter of $V$ to Y to be

$$(Y : V) := \mathrm{Ker}(\underline{\mathrm{Hom}}(M, X) \to \underline{\mathrm{Hom}}(V, X/Y)).$$

We now have the following:

**Lemma 4.** *Take the full abelian subcategory $\mathcal{C}_X$ of $\mathcal{C}$ generated by subquotients of $X^n$, consider the largest subobject $P_X$ of $\underline{\mathrm{Hom}}(\omega(X), X)$ whose image in $\underline{\mathrm{Hom}}(\omega(X)^n, X^n)$ under diagonal embedding is contained in $(Y : \omega(Y))$ for all subobjects $Y$ of $X^n$ and all $n$. Then the Theorem (2) is true for $\mathcal{C}_X$ with coalgebra defined by $A_X := \omega(P_X)^\vee$.*

*Proof.* $P_X$ exists because $\mathcal{C}$ is Artinian. Notice that $\omega$ takes $\underline{\mathrm{Hom}}(M, X)$ to $\mathrm{Hom}_R(M, X)$ and $(Y : V)$ to $(\omega(Y) : V)$, so it takes $P_X$, which is defined by

$$\bigcap \left( \underline{\mathrm{Hom}}(\omega(X), X) \cap (Y : \omega(Y)) \right)$$

to

$$\bigcap (\mathrm{End}_R(\omega(X)) \cap (\omega(Y) : \omega(Y))) .$$

Hence $\omega(P_X)$ is the largest subring of $\mathrm{End}_R(\omega(X))$ stabilizing $\omega(Y)$ for all $Y \subset X^n$ and all $n$. It's a finitely generated projective $R$ module by construction, and so is $A_X$. Note that only finitely many intersection occurs because $\underline{\mathrm{Hom}}(\omega(X), X)$ is Artinian.

Next, take any flat $R$ module $M$,[62] since $\mathcal{C}_X$ is generated by subquotients of $X$, an element $\lambda \in \mathrm{Nat}(\omega, \omega \otimes M)$ is completely determined by it is value on $X$, so $\lambda \in \mathrm{End}_R(\omega(X)) \otimes M$. Since $- \otimes_R M$ is an exact functor, we have:

$$\bigcap (\mathrm{Hom}_R(\omega(X), \omega(X) \otimes_R M) \cap (\omega(Y) \otimes_R M : \omega(Y)))$$
$$= \left( \bigcap (\mathrm{End}_R(\omega(X)) \cap (\omega(Y) : \omega(Y))) \right) \otimes_R M .$$

This follows because there are only finitely many intersections and finite limit commutes with tensoring with flat module. Therefore,

$$\lambda \in \omega(P_X) \underset{R}{\otimes} M .$$

---

[62]Recall that a $R$ module is flat if and only if it is a filtered colimit of finitely generated projective modules.

Conversely, every element in $\omega(P_X) \otimes_R M$ gives rise to a natural transform in the way described above. Hence we establish the isomorphism

$$\text{Nat}(\omega, \omega \otimes M) \cong \omega(P_X) \otimes_R M \cong \text{Hom}_R(A_X, M).$$

$A_X$ is unique up to unique isomorphism (as a flat $R$ module) because it represents the functor $M \mapsto \text{Nat}(\omega, \omega \otimes M)$.

Next, we shall define a co-action of $A_X$ on $\omega$, a counit and a coproduct on $A_X$ which makes $A_X$ an $R$-coalgebra and $\omega$ a corepresentation:

$$\rho \in \text{Nat}(\omega, \omega \otimes A_X) \cong \text{End}_R(A_X)$$

corresponds to the identity map of $A_X$, and

$$\epsilon \in \text{Hom}_R(A_X, R) \cong \text{Nat}(\omega, \omega)$$

corresponds to $\text{Id}_\omega$. The co-action $\rho$ tensored with $\text{Id}_{A_X}$ gives a natural transform between $\omega \otimes A_X$ and $\omega \otimes A_X \otimes A_X$, whose composition with $\rho$ gives the following commutative diagram:

$$
\begin{array}{ccc}
\omega & \xrightarrow{\ \rho\ } & \omega \otimes A_X \\
& {}_{\psi}\searrow & \downarrow{}^{\rho \otimes \text{Id}_{A_X}} \\
& & \omega \otimes A_X \otimes A_X
\end{array}
\quad .
$$

Take $\Delta$ to be the image of $\psi$ in $\text{Hom}_R(A_X, A_X \otimes_R A_X)$. It follows from definition that $A_X$ is counital and $\rho : \omega \to \omega \otimes A_X$ is a corepresentation. It remains to check that $\Delta$ is coassociative.

Observe that the essential image of $\omega \otimes A_X$ is a subcategory of the essential image of $\omega$, hence every functor that shows up here can be restricted to $\omega \otimes A_X$, in particular, $\rho$, whose restriction to $\omega \otimes A_X$ is obviously $\rho \otimes \text{Id}_{A_X}$. It follows from the definition that

$$(\rho \otimes \text{Id}_{A_X}) \circ \rho = (\text{Id}_\omega \otimes \Delta) \circ \rho \in \text{Nat}(\omega, \omega \otimes A_X \otimes A_X).$$

Restrict this equation to $\omega \otimes A_X$ and we get

$$(\rho \otimes \text{Id}_{A_X} \otimes \text{Id}_{A_X}) \circ (\rho \otimes \text{Id}_{A_X}) = (\text{Id}_\omega \otimes \text{Id}_{A_X} \otimes \Delta) \circ (\rho \otimes \text{Id}_{A_X}).$$

Composing with $\rho$, the LHS corresponds to $(\Delta \otimes \text{Id}_{A_X}) \circ \Delta$ and the RHS corresponds to $(\text{Id}_{A_X} \otimes \Delta) \circ \Delta$ whose equality is exactly the coassociativity of $A_X$.

It follows that $\forall Z \in \mathcal{C}_X$,

$$\rho(Z) : \omega(Z) \to \omega(Z) \otimes_R A_X$$

gives $\omega(Z)$ a $A_X$ corepresentation structure and this is functorial in $Z$, thus $\omega$ factors through a $\phi : \mathcal{C}_X \to \text{Corep}_R(A_X)$.

Back to the uniqueness of $A_X$. It has been shown that it is unique up to unique isomorphism as a flat $R$ module. Additionally, if $\phi : A_X \to A'_X$ is an isomorphism such that it induces identity transformation on the functor $M \mapsto \text{Nat}(\omega, \omega \otimes M)$ then, $\phi$ automatically maps the triple $(\Delta, \epsilon, \rho)$ to $(\Delta', \epsilon', \rho')$, so $\phi$ is a coalgebra isomorphism.

Finally, it remains to show that when $R = k$, $\phi$ is essentially surjective[63] and full:

---

[63]In fact, $\phi$ is essentially surjective even without the assumption that $R = k$.

- Essentially Surjective: If $M \in \mathrm{Corep}_k(A_X)$, then define

$$\widetilde{M} := \mathrm{Coker}(M \otimes \omega(P_X) \otimes P_X \rightrightarrows M \otimes P_X),$$

where two arrows are $\omega(P_X)$ representation structure of $M$ and $P_X$ respectively, then

$$\omega(\widetilde{M}) = M \underset{\omega(P_X)}{\otimes} \omega(P_X) = M.$$

- Full: If $f : M \to N$ is a $A_X$-corepresentation morphism, then by the $k$-linearlity of $\mathcal{C}_X$, $f$ lifts to morphisms

$$f \otimes \mathrm{Id}_{P_X} : M \otimes P_X \to N \otimes P_X,$$

and

$$f \otimes \mathrm{Id}_{\omega(P_X)} \otimes \mathrm{Id}_{P_X} : M \otimes \omega(P_X) \otimes P_X \to N \otimes \omega(P_X) \otimes P_X.$$

Thus, passing to cokernel gives rise to $\widetilde{f} : \widetilde{M} \to \widetilde{N}$ which is mapped to $f$ by $\omega$.

$\square$

Next we move on to recover the category $\mathcal{C}$ by its subcategories $\mathcal{C}_X$. Define an index category $I$ such that its objects are isomorphism classes of objects in $\mathcal{C}$, denoted by $X_i$ for each index $i$, and a unique arrow from $i$ to $j$ if $X_i$ is a subobject of $X_j$. $I$ is directed because for any two objects $Z$ and $W$, they are subobjects of $Z \oplus W$. Observe that if $X$ is a subobject of $Y$, then $\mathcal{C}_X$ is a full subcategory of $\mathcal{C}_Y$, so a functorial restriction

$$\mathrm{Hom}_R(A_Y, M) \cong \mathrm{Nat}(\omega_Y, \omega_Y \otimes M) \to \mathrm{Nat}(\omega_X, \omega_X \otimes M) \cong \mathrm{Hom}_R(A_Y, M),$$

gives rise to a coalgebra homomorphism $A_X \to A_Y$. Futhermore, this homomorphism is injective because $\omega(P_Y) \to \omega(P_X)$ is surjective, otherwise $\mathrm{Coker}(\omega(P_Y) \to \omega(P_X))$ will be mapped to the zero object in $\mathrm{Corep}_R(A_Y)$, which contradicts with $\omega$ being faithful.

**Lemma 5.** *Define the coalgbra*

$$A := \varinjlim_{i \in I} A_{X_i},$$

*then it is the desired coalgebra in Theorem 2.*

*Proof.* $A$ is flat because it is an inductive limit of flat $R$ modules. Moreover

$$\mathrm{Hom}_R(A, M) = \varprojlim_{i \in I} \mathrm{Hom}_R(A_{X_i}, M) \cong \varprojlim_{i \in I} \mathrm{Nat}(\omega_{X_i}, \omega_{X_i} \otimes M) = \mathrm{Nat}(\omega, \omega \otimes M),$$

which gives the desired functorial property and this implies that $A$ is unique up to unique isomorphism. Finally, when $R = k$, the functor $\phi$ is defined and it is fully faithful because it is fully faithful on each subcategory $\mathcal{C}_{X_i}$. It's also essentially surjective because every corepresentation $V$ of $A$ comes from a corepresentation of a finite dimensional sub-coalgebra of $A$,[64] and $A$ is a filtered union of sub-coalgebras $A_{X_i}$, so $V$ comes from a corepresentation of some $A_{X_i}$. $\square$

*Proof of Theorem 2.* It remains to prove the theorem when $\mathcal{C}$ is monoidal. This amounts to including $m : \mathcal{C} \boxtimes \mathcal{C} \to \mathcal{C}$ and $e : \mathbf{1} \to \mathcal{C}$ with associativity and unitarity constrains, where $\mathbf{1}$ is

---

[64]Take a basis $\{e_i\}$ for $V$, the co-action $\rho$ takes $e_i$ to $\sum_j e_j \otimes a_{ji}$, then it is easy to see that span$\{a_{ji}\}$ is a finite dimensional sub-coalgebra of $A$.

the trivial tensor category with objects $\{0, 1\}$ and only nontrivial morphisms are $\text{End}(1) = k$. Using the isomorphism:

$$\text{Hom}_R(A \otimes_R A, A \otimes_R A) \cong \text{Nat}(\omega \boxtimes \omega, \omega \boxtimes \omega \otimes A \otimes_R A),$$

we get a homomorphism

$$\tau : \text{Hom}_R(A \otimes_R A, M) \to \text{Nat}(\omega \boxtimes \omega, \omega \boxtimes \omega \otimes M).$$

It is an isomorphism because for each pair of subcategories $(\mathcal{C}_X, \mathcal{C}_Y)$

$$\begin{aligned} \text{Hom}_R(A_X \otimes_R A_Y, M) &\cong \text{Hom}_R(A_X, R) \otimes_R \text{Hom}_R(A_Y, M) \\ &\cong \text{Nat}(\omega_X, \omega_X) \otimes_R \text{Nat}(\omega_Y, \omega_Y \otimes M) \\ &\cong \text{Nat}(\omega_X \boxtimes \omega_Y, \omega_X \boxtimes \omega_Y \otimes M) \end{aligned}$$

and it is compatible with the homomorphism given above, so after taking limit, $\tau$ is an isomorphism. We also have a homomorphism:

$$\text{Nat}(\omega, \omega \otimes M) \to \text{Nat}(\omega \boxtimes \omega, \omega \boxtimes \omega \otimes M),$$

by taking any $\alpha \in \text{Nat}(\omega, \omega \otimes M)$, and composing with the isomorphism $\omega \boxtimes \omega(X \boxtimes Y) \cong \omega(X \otimes Y)$. This homomorphism in turn becomes a homomorphism

$$\mu : A \otimes_R A \to A.$$

And the obvious isomorphism

$$\text{Hom}_R(R, M) = M \to \text{Nat}(\omega_{\mathbb{1}}, \omega_{\mathbb{1}} \otimes M),$$

together with the unit functor $e : \mathbb{1} \to \mathcal{C}$ give a homomorphism

$$\iota : R \to A.$$

All of the homomorphisms are functorial with respect to $M$ so $\mu$ and $\iota$ are homomorphisms between coalgebras. Now the associativity and unitarity of monoidal category $\mathcal{C}$ translates into associativity and unitarity of $\mu$ and $\iota$, which are exactly conditions for $A$ to be a bialgebra. This concludes the proof of Theorem 2. $\qquad \square$

*Remark* 8. In the statement of Theorem 2, it is assumed that $R$ is a local ring or a PID, for the following technical reason: we want to introduce the functor

$$\otimes : \text{Proj}_f(R) \times \mathcal{C} \to \mathcal{C},$$

which is defined by sending $(R^n, X)$ to $X^n$. This is feasible only if every finite projective module is free, which is not always true for an arbitary ring. Nevertheless, this is true when $R$ is local or a PID. It is tempting to eliminate this assumption when $\mathcal{C}$ is rigid, since we only use the $\underline{\text{Hom}}(\omega(X), X)$ to define the crucial object $P_X$, and there is no need to define a $\underline{\text{Hom}}$ when the category is rigid. In fact, there is no loss of information if we define $P_X$ by

$$\bigcap \left( \underline{\text{Hom}}(X, X) \cap (Y : Y) \right),$$

then the fiber functor $\omega$ takes $P_X$ to

$$\bigcap \left( \text{End}_R(\omega(X)) \cap (\omega(Y) : \omega(Y)) \right),$$

since $\omega$ is monoidal by definition and a monoidal functor between rigid monoidal categories preserves duality and thus preserves inner Hom. $\qquad \triangle$

Following the above remark, we drop the assumption on ring $R$ and state the following version of Tannaka reconstruction for Hopf algebras:

**Theorem 3** (Tannaka Reconstruction for Hopf Algebra)**.** *Let $R$ be a commutative $k$-algebra, $\mathcal{C}$ a $k$-linear abelian rigid monoidal category (resp. abelian rigid braided monoidal) with a fiber functor $\omega$ to $Mod_f(R)$, then there exists a unique flat $R$-Hopf algebra $A$ (resp. $R$-coquasitriangular Hopf algebra), up to unique isomorphism, such that $A$ represents the endomorphism of $\omega$ in the sense that $\forall M \in IndProj_f(R)$*

$$Hom_R(A, M) \cong Nat(\omega, \omega \otimes M).$$

*Moreover, there is a functor $\phi : \mathcal{C} \to Corep_R(A)$ which makes the following diagram commutative:*

$$\mathcal{C} \xrightarrow{\phi} Corep_R(A)$$
$$\omega \searrow \quad \downarrow forget$$
$$Mod_f(R)$$

*and $\phi$ is an equivalence if $R = k$.*

*Sketch of proof.* The idea of proof basically follows [56]. Accoring to Remark 8 and Theorem 2, there exists a bialgebra $A$ which satisfies all conditions in the theorem, so it remains to prove that there are compatible structures on $A$ when $\mathcal{C}$ has extra structures.

(a) $\mathcal{C}$ **is rigid.** This means that there is an equivalence between $k$-linear abelian monoidal categories

$$\sigma : \mathcal{C} \to \mathcal{C}^{op},$$

by taking the right dual of each object, so it turns into an isomophism between $R$ modules

$$\sigma : \mathrm{Nat}(\omega, \omega \otimes M) \to \mathrm{Nat}(\omega^{op}, \omega^{op} \otimes M).$$

According to the functoriality of the construction of the bialgebra $A$, there is a bialgebra isomorphism:

$$\mathcal{S} : A \to A^{op},$$

put it in another way, a bialgebra anti-automorphism of $A$. To prove that it satisfies the required compatibility:

$$\mu \circ (\mathcal{S} \otimes \mathrm{Id}) \circ \Delta = \iota \circ \epsilon = \mu \circ (\mathrm{Id} \otimes \mathcal{S}) \circ \Delta,$$

we observe that $\iota \circ \epsilon$ gives the natural transformation

$$\mathrm{Id} \otimes \rho_{\omega(1)} : \omega(X) = \omega(X) \otimes \omega(1) \mapsto \omega(X) \otimes \rho(\omega(1)),$$

but 1 is the trivial corepresentation of $A$, so $\rho(\omega(1))$ is canonically identified with $\omega(1)$, so $\iota \circ \epsilon$ is just the identity morphism on $\omega(X)$. On the other hand, $\mu \circ (\mathcal{S} \otimes \mathrm{Id}) \circ \Delta$ corresponds to the homomorphism

$$\omega(X) \to \omega(X) \otimes \omega(X)^{\vee} \otimes \omega(X) \to \omega(X) \otimes \omega(X^{\vee} \otimes X) \to \omega(X) \otimes \omega(1) = \omega(X),$$

which is identity by the rigidity of $\mathcal{C}$, hence $\mu \circ (\mathcal{S} \otimes \mathrm{Id}) \circ \Delta = \iota \circ \epsilon$. The other equation is similiar.

(b) $\mathcal{C}$ **is rigid braided.** This means that there is a natural transformation:

$$r : \omega \boxtimes \omega \to \omega \boxtimes \omega,$$

which gives the braiding. This corresponds to a homomorphism of R-modules

$$\mathcal{R} : A \otimes A \to R,$$

let's define it to be the universal R-matrix. The fact that $r$ is a natural transformation is equivalent to the diagram below being commutative

$$
\begin{array}{ccccc}
\omega(U) \otimes \omega(V) & \xrightarrow{\rho \otimes \rho} & \omega(U) \otimes \omega(V) \otimes A \otimes A & \xrightarrow{\mathrm{Id} \otimes \mathrm{Id} \otimes \mu} & \omega(U) \otimes \omega(V) \otimes A \\
\downarrow{r} & & & & \downarrow{r \otimes \mathrm{Id}} \\
\omega(V) \otimes \omega(U) & \xrightarrow{\rho \otimes \rho} & \omega(V) \otimes \omega(U) \otimes A \otimes A & \xrightarrow{\mathrm{Id} \otimes \mathrm{Id} \otimes \mu} & \omega(V) \otimes \omega(U) \otimes A
\end{array}
,
$$

which in turn translates to the following equation of $\mathcal{R}$:

$$\mathcal{R}_{12} \circ \mu_{24} \circ (\Delta \otimes \Delta) = \mathcal{R}_{23} \circ \mu_{13} \circ \tau_{13} \circ (\Delta \otimes \Delta),$$

where $\tau : A \otimes A \to A \otimes A$ sends $x \otimes y$ to $y \otimes x$. The compactibility of $r$ with the identity

$$
\begin{array}{ccc}
\omega(X) & \longrightarrow & \omega(X) \otimes \omega(1) \\
\downarrow{\mathrm{Id}} & & \downarrow{r} \\
\omega(X) & \longleftarrow & \omega(1) \otimes \omega(X)
\end{array}
,
$$

translates to $\mathcal{R} \circ (\mathrm{Id}_A \otimes 1) = \epsilon$. And symmetrically $\mathcal{R} \circ (1 \otimes \mathrm{Id}_A) = \epsilon$.

Finally, the hexagon axiom of braiding:

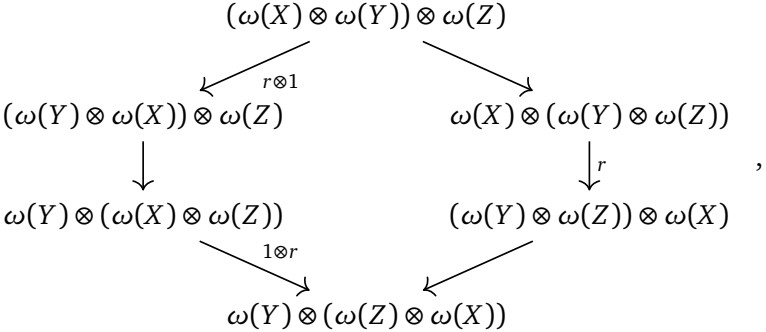

translates to the commutativity of the diagram

$$
\begin{array}{ccc}
A \otimes A \otimes A & \xrightarrow{\mathrm{Id} \otimes \mathrm{Id} \otimes \Delta} & A \otimes A \otimes A \otimes A \\
\downarrow{\mu \otimes \mathrm{Id}} & & \downarrow{\mathcal{R}_{13} \cdot \mathcal{R}_{24}} \\
A \otimes A & \xrightarrow{\mathcal{R}} & R
\end{array}
,
$$

and the same hexagon but with $r^{-1}$ instead of $r$ gives another one:

$$
\begin{array}{ccc}
A \otimes A \otimes A & \xrightarrow{\;\Delta \otimes \mathrm{Id} \otimes \mathrm{Id}\;} & A \otimes A \otimes A \otimes A \\
{\scriptstyle \mathrm{Id} \otimes \mu}\downarrow & & \downarrow{\scriptstyle \mathcal{R}_{14} \cdot \mathcal{R}_{23}} \\
A \otimes A & \xrightarrow{\qquad \mathcal{R} \qquad} & R
\end{array} \quad .
$$

So we end up confirming all the properties that universal R-matrix should satisfy, and we conclude that $A$ is indeed a coquasitriangular Hopf algebra.

$\square$

**Reconstruction from representation**    It is tempting to dualize everything above to formalize the Tannaka reconstruction for the category of representations. In other words, we can take the dual of $A$ instead of $A$ itself, and a corepresentation becomes the representaion, and when the category has extra structures, those structures will be dualized, for example, when $\mathcal{C}$ is a $k$-linear abelian rigid braided monoidal category, it should come from the representation category of a flat $R$-quasitriangular Hopf algebra, since the dual of those diagrams involved in the proof of Theorem 3 are exactly properties of universal R-matrix of a quasitriangular Hopf algebra.

This is naive because the statement:

$$
\mathrm{Hom}_R(U, V \otimes A) \cong \mathrm{Hom}_R(U \otimes A^*, V),
$$

is not true in general, since $A$ can be infinite dimensional, thus the naive dualizing procedure is not feasible. To resolve this subtlety, we observe that $A$ is constructed from a filtered colimit of finite projective $R$-modules, each is an $R$-coalgebra, and any finitely generated corepresentation of $A$ comes from a corepresentation of a finite coalgebra, so it is natural to define the action of $A^*$ on those modules by factoring through some finite quotient $A_X^*$ for some $X \in \mathrm{ob}(\mathcal{C})$. Similarly, the multiplication structure on $A^*$ can be defined by first projecting down to some finite quotient and taking multiplication

$$
A^* \otimes A^* = \varprojlim_{i \in I} A_{X_i} \otimes \varprojlim_{i \in I} A_{X_i} \longrightarrow A_{X_i} \otimes A_{X_i} \longrightarrow A_{X_i}
$$

which is compatible with transition map $A_{X_j} \to A_{X_i}$ then taking the inverse limit gives the multiplication of $A^*$. For antipode $\mathcal{S}$, its dual is a map $A^* \to A^*$.

On the other hand, the comultiplication on $A^*$, is still subtle. If we dualize the multiplication of $A$, cut-off at some finite submodule

$$
A_{X_i} \otimes A_{X_j} \longrightarrow A,
$$

we only get an inverse system of morphisms from $A^*$ to $A_{X_i}^* \otimes A_{X_j}^*$ and the latter's inverse limit is $A^* \widehat{\otimes} A^*$, instead of $A^* \otimes A^*$. So we actually get a *topological Hopf algebra* with topological basis

$$
N_i := \ker(A^* \to A_{X_i}^*),
$$

so that the comultiplication is continuous. Similiarly the counit, multiplication, and anipode are continuous as well. Finally when $\mathcal{C}$ is braided, there exists an invertible element $\mathcal{R} \in A^* \widehat{\otimes} A^*$, and the dual of the structure homomorphism in $A$ is exactly the condition that $\mathcal{R}$ is the universal R-matrix of a topological quasitriangular Hopf algebra.

So we can restate Theorem 3 in terms of representations of topological Hopf algebras:

**Theorem 4.** *Let $R$ be a commutative $k$-algebra, $\mathcal{C}$ a $k$-linear abelian rigid monoidal category (resp. abelian rigid braided monoidal) with a fiber functor $\omega$ to $Mod_f(R)$, then there exists a unique topological $R$-Hopf algebra $H$ (resp. $R$-quasitriangular Hopf algebra) which is an inverse limit of finite projective $R$-modules endowed with discrete topology, up to unique isomorphism, such that $H$ represents the endomorphism of $\omega$ in the sense that*

$$H \cong Nat(\omega, \omega).$$

*Moreover, there is a functor $\phi : \mathcal{C} \to Rep_R(H)$ which sends an object in $\mathcal{C}$ to a continuous representation of $H$ and makes the following diagram commutative:*

$$
\begin{array}{ccc}
\mathcal{C} & \xrightarrow{\phi} & Rep_R(H) \\
& \searrow{\scriptstyle \omega} & \downarrow{\scriptstyle forget} \\
& & Mod_f(R)
\end{array} ,
$$

*and $\phi$ is an equivalence if $R = k$.*

**Application to Quantization**    We now consider the case that we have a category $\mathcal{C}_\hbar$, which is a *quantization* of the category of representations of some Hopf algebra $H$ over $\mathbb{C}$. The quantization, namely $\mathcal{C}_\hbar$, of $Rep_{\mathbb{C}}(H)$ is a $\mathbb{C}$-linear abelian monoidal category which has the same set of generators as $Rep_{\mathbb{C}}(H)$, together with a fiber functor $\omega_\hbar : \mathcal{C}_\hbar \to Mod_f(\mathbb{C}[\![\hbar]\!])$ which acts on generators of $Rep_{\mathbb{C}}(H)$ by tensoring with $\mathbb{C}[\![\hbar]\!]$, and

$$\mathrm{Hom}_{\mathcal{C}_\hbar}(X, Y) \cong \mathrm{Hom}_{\mathcal{C}_\hbar}(X, Y)/\hbar = \mathrm{Hom}_{Rep_{\mathbb{C}}(H)}(X, Y)$$

for any pair of generators $X$ and $Y$. For example, the classical algebra of local observables in 4d Chern-Simons theory is $U(g[z])$, the universal enveloping algebra of Lie algebra $g[z]$, which has the category of representations generated by classical Wilson lines. Quantized Wilson lines naturally generated a $\mathbb{C}$-linear abelian monoidal category.

Applying Theorem 4, $(\mathcal{C}_\hbar, \omega_\hbar)$ gives us a (topological) $\mathbb{C}[\![\hbar]\!]$-Hopf algebra $H_\hbar$. Since $\mathcal{C}_\hbar$ and $\mathcal{C}$ shares the same set of generators, and the construction of those Hopf algebras as $\mathbb{C}[\![\hbar]\!]$-modules only involves generators of corresponding categories, so $H_\hbar$ is isomorphic to the completion of $H \otimes \mathbb{C}[\![\hbar]\!]$ in the $\hbar$-adic topology:

$$
\begin{aligned}
H_\hbar := \varprojlim_{i \in I} H_{X_i} \otimes \mathbb{C}[\![\hbar]\!] &\cong \varprojlim_{i \in I} \varprojlim_{n} H_{X_i} \otimes \mathbb{C}[\hbar]/(\hbar^n) \\
&\cong \varprojlim_{n} \varprojlim_{i \in I} H_{X_i} \otimes \mathbb{C}[\hbar]/(\hbar^n) \\
&\cong \varprojlim_{n} H \otimes \mathbb{C}[\hbar]/(\hbar^n).
\end{aligned}
$$

For the same reason, tensor product of two copies of $H_\hbar$ and completed in the inverse limit topology is isomorphic to the completion of $H_\hbar \otimes_{\mathbb{C}[\![\hbar]\!]} H_\hbar$ in the $\hbar$-adic topology:

$$H_\hbar \widehat{\otimes} H_\hbar \cong \varprojlim_{n} H_\hbar \otimes_{\mathbb{C}[\![\hbar]\!]} H_\hbar / (\hbar^n).$$

From the construction of those Hopf algebras and the condition that a morphism in $\mathcal{C}_\hbar$ modulo $\hbar$ is a morphism in $Rep_{\mathbb{C}}(H)$, it is easy to see that modulo $\hbar$ respects all structure homomorphisms, thus $H_\hbar$ modulo $\hbar$ and $H$ are isomorphic as Hopf algebras. Finally, structure homomorphisms of $H_\hbar$ are continuous in the $\hbar$-adic topology because they are $\hbar$-linear. Thus we conlude that:

**Theorem 5.** *$H_\hbar$ is a quantization of $H$ in the sense of Definition 6.1.1 of [45], i.e. it is a topological Hopf algebra over $\mathbb{C}[\![\hbar]\!]$ with $\hbar$-adic topology, such that*

(i) *$H_\hbar$ is isomorphic to $H[\![\hbar]\!]$ as a $\mathbb{C}[\![\hbar]\!]$-module;*

(ii) *$H_\hbar$ modulo $\hbar$ is isomorphic to $H$ as Hopf algebras.*

In our case, $H = U(g[z])$ for $g = \mathfrak{gl}_K[z]$, so $H_\hbar$ is a quantization of $U(\mathfrak{gl}_K[z])$, and according to Theorem 12.1.1 of [45], this is unique up to isomorphisms. This proves Proposition (2).

# C Technicalities of Witten Diagrams

## C.1 Vanishing lemmas

We introduce some lemmas to allow us to readily declare several Witten diagrams in the 4d Chern-Simons theory to be zero.

**Lemma 6.** *The product of two or three bulk-to-bulk propagators vanish when attached cyclically, diagrammatically this means:*

$$v_1 \bullet\!\!\bigcirc\!\!\bullet v_0 \;=\; \bigcirc\!\!\bullet v_0 \;=\; 0 \,. \tag{213}$$

*Proof.* Two propagators: We can choose one of the two bulk points, say $v_0$, to be at the origin and denote $v_1$ simply as $v$. This amounts to taking the projection (111), namely: $\mathbb{R}^4_{v_0} \times \mathbb{R}^4_{v_1} \ni (v_0, v_1) \mapsto v_1 - v_0 =: v \in \mathbb{R}^4$. Then the product of the two propagators become:

$$P(v_0, v_1) \wedge P(v_1, v_0) \mapsto \overline{P}(v) \wedge \overline{P}(-v) = -\overline{P}(v) \wedge \overline{P}(v) \,. \tag{214}$$

This is a four form at $v$, however, $P$ does not have any $\mathrm{d}z$ component, therefore the four form $P(v) \wedge P(v)$ necessarily contains repetition of a one form and thus vanishes.

Three propagators: By choosing $v_0$ to be the origin of our coordinate system we can turn the product to the following:

$$\overline{P}(v_1) \wedge \overline{P}(v_2) \wedge P(v_1, v_2) \,. \tag{215}$$

We now need to look closely at the propagators (see (111) and (114)):

$$\overline{P}(v_i) = \frac{\hbar}{2\pi} \frac{x_i \, \mathrm{d}y_i \wedge \mathrm{d}\overline{z}_i + y_i \, \mathrm{d}\overline{z}_i \wedge \mathrm{d}x_i + 2\overline{z}_i \, \mathrm{d}x_i \wedge \mathrm{d}y_i}{d(v_i, 0)^4} \,, \tag{216a}$$

$$P(v_1, v_2) = \frac{\hbar}{2\pi} \frac{x_{12} \, \mathrm{d}y_{12} \wedge \mathrm{d}\overline{z}_{12} + y_{12} \, \mathrm{d}\overline{z}_{12} \wedge \mathrm{d}x_{12} + 2\overline{z}_{12} \, \mathrm{d}x_{12} \wedge \mathrm{d}y_{12}}{d(v_1, v_2)^4} \,, \tag{216b}$$

where $v_i := (x_i, y_i, z_i, \overline{z}_i)$, $x_{ij} := x_i - x_j$, $y_{ij} := y_i - y_j, \cdots$, and $d(v_i, v_j)^2 := (x_{ij}^2 + y_{ij}^2 + z_{ij}\overline{z}_{ij})$. Since the propagators don't have any $\mathrm{d}z$ component the product (215) must be proportional to $\omega := \bigwedge_{i \in \{1,2\}} \mathrm{d}x_i \wedge \mathrm{d}y_i \wedge \mathrm{d}\overline{z}_i$. In the product there are six terms that are proportional to $\omega$. For example, we can pick $\mathrm{d}x_1 \wedge \mathrm{d}y_1$ from $\overline{P}(v_1)$, $\mathrm{d}\overline{z}_2 \wedge \mathrm{d}x_2$ from $\overline{P}(v_2)$ and $\mathrm{d}y_{12} \wedge \mathrm{d}\overline{z}_{12}$ from $P(v_1, v_2)$, this term is proportional to:

$$\mathrm{d}x_1 \wedge \mathrm{d}y_1 \wedge \mathrm{d}\overline{z}_2 \wedge \mathrm{d}x_2 \wedge \mathrm{d}y_{12} \wedge \mathrm{d}\overline{z}_{12} = -\mathrm{d}x_1 \wedge \mathrm{d}y_1 \wedge \mathrm{d}\overline{z}_2 \wedge \mathrm{d}x_2 \wedge \mathrm{d}y_2 \wedge \mathrm{d}\overline{z}_1 = +\omega \,. \tag{217}$$

The other five such terms are:

$$\mathrm{d}y_1 \wedge \mathrm{d}\bar{z}_1 \wedge \mathrm{d}\bar{z}_2 \wedge \mathrm{d}x_2 \wedge \mathrm{d}x_{12} \wedge \mathrm{d}y_{12} = -\omega,$$
$$\mathrm{d}y_1 \wedge \mathrm{d}\bar{z}_1 \wedge \mathrm{d}x_2 \wedge \mathrm{d}y_2 \wedge \mathrm{d}\bar{z}_{12} \wedge \mathrm{d}x_{12} = +\omega,$$
$$\mathrm{d}\bar{z}_1 \wedge \mathrm{d}x_1 \wedge \mathrm{d}y_2 \wedge \mathrm{d}\bar{z}_2 \wedge \mathrm{d}x_{12} \wedge \mathrm{d}y_{12} = +\omega, \tag{218}$$
$$\mathrm{d}\bar{z}_1 \wedge \mathrm{d}x_1 \wedge \mathrm{d}x_2 \wedge \mathrm{d}y_2 \wedge \mathrm{d}y_{12} \wedge \mathrm{d}\bar{z}_{12} = -\omega,$$
$$\mathrm{d}x_1 \wedge \mathrm{d}y_1 \wedge \mathrm{d}y_2 \wedge \mathrm{d}\bar{z}_2 \wedge \mathrm{d}\bar{z}_{12} \wedge \mathrm{d}x_{12} = -\omega.$$

These signs can be determined from a determinant, stated differently, we have the following equation:

$$\det \begin{pmatrix} \mathrm{d}y_1 \wedge \mathrm{d}\bar{z}_1 & \mathrm{d}\bar{z}_1 \wedge \mathrm{d}x_1 & \mathrm{d}x_1 \wedge \mathrm{d}y_1 \\ \mathrm{d}y_2 \wedge \mathrm{d}\bar{z}_2 & \mathrm{d}\bar{z}_2 \wedge \mathrm{d}x_2 & \mathrm{d}x_2 \wedge \mathrm{d}y_2 \\ \mathrm{d}y_{12} \wedge \mathrm{d}\bar{z}_{12} & \mathrm{d}\bar{z}_{12} \wedge \mathrm{d}x_{12} & \mathrm{d}x_{12} \wedge \mathrm{d}y_{12} \end{pmatrix} = -6\omega, \tag{219}$$

where the product used in taking determinant is the wedge product. The above equation implies that in the product (215) the coefficient of $-\omega$ is given by the same determinant if we replace the two forms with their respective coefficients as they appear in (216). Therefore, the coefficient is:

$$\frac{1}{8\pi^3 d(v_1,0)^4 d(v_2,0)^4 d(v_1,v_2)^4} \det \begin{pmatrix} x_1 & y_1 & \bar{z}_1 \\ x_2 & y_2 & \bar{z}_2 \\ x_{12} & y_{12} & \bar{z}_{12} \end{pmatrix} = 0. \tag{220}$$

The determinant vanishes because the three rows of the matrix are linearly dependent. Thus we conclude that the product (215) vanishes. $\qquad \square$

**Lemma 7.** *The product of two bulk-to-bulk propagators joined at a bulk vertex where the other two endpoints are restricted to the Wilson line, vanishes, i.e., in any Witten diagram:*

$$\overset{p_1 \quad p_2}{\underset{v \, \bullet}{\diagdown\!\!\diagup}} = 0. \tag{221}$$

*Proof.* This simply follows from the explicit form of the bulk-to-bulk propagator. Computation verifies that:

$$\iota_{\partial_{x_1} \wedge \partial_{x_2}} (P(v,p_1) \wedge P(v,p_2)) = 0, \tag{222}$$

where $x_1$ and $x_2$ are the $x$-coordinates of the points $p_1$ and $p_2$ respectively. $\qquad \square$

The world-volume on which the CS theory is defined is $\mathbb{R}^2_{x,y} \times \mathbb{C}_z$, which in the presence of the Wilson line at $y = z = 0$ we view as $\mathbb{R}_x \times \mathbb{R}_+ \times S^2$. When performing integration over this space we approximate the non-compact direction by a finite interval and then taking the length of the interval to infinity. In doing so we introduce boundaries of the world-volume, namely the two components $B_{\pm D} := \{\pm D\} \times \mathbb{R}_+ \times S^2$ at the two ends of the interval $[-D, D]$. Our next lemma concerns some integrals over these boundaries.

**Lemma 8.** *The integral over a bulk point vanishes when restricted to the spheres at infinity, in diagram:*

$$\lim_{D \to \infty} \int_{v_0 \in B_{\pm D}} v_0 \, \bullet \!\!\!\underset{v_n}{\overset{v_1}{<}}\vdots = 0. \tag{223}$$

*Proof.* Symbolically, the integration can be written as:

$$\lim_{D \to \infty} \int_{B_{\pm D}} \mathrm{dvol}_{B_{\pm D}} \, \iota_{\partial_y \wedge \partial_{\bar{z}}} \left( P(v_0, v_1) \wedge \cdots \wedge P(v_0, v_n) \right), \tag{224}$$

where $y$ and $\bar{z}$ are coordinates of $v_0$. Note that the $\mathrm{d}z$ required for the volume form on $B_{\pm D}$ comes from the structure constant at the interaction vertex, not from the propagators. In the above integration the $x$-component of $v_0$ is fixed at $\pm D$, which introduces $D$ dependence in the integrand. The bulk-to-bulk propagator has the following asymptotic scaling behavior:[65]

$$P((D, y, z, \bar{z}), v_j) \overset{D \to \infty}{\sim} D^{-2} + \mathcal{O}(D^{-3}). \tag{225}$$

The integration measure on $B_{\pm D}$ is independent of $D$, therefore the integral behaves as $D^{-2n}$ for large $D$, and consequently vanishes in the limit $D \to \infty$. $\qquad \square$

## C.2 Comments on integration by parts

Finally, let us make a few general remarks about the integrals involved in computing Witten diagrams. Since the boundary-to-bulk propagators are exact and the bulk-to-bulk propagators behave nicely when acted upon by differential (see (112)), we want to use Stoke's theorem to simplify any given Witten diagram. Suppose we have a Witten diagram with $m$ propagators connected to the boundary, $n$ propagators connected to the Wilson line, and $l$ bulk points. Let us denote the bulk points by $v_i$ for $i = 1, \cdots, l$, the points on the Wilson line by $p_j$ for $j = 1, \cdots, n$, and the points on the boundary as $x_k$ for $k = 1, \cdots, m$. The domain of integration for the diagram is then $M^l \times \Delta_n$, where $M = \mathbb{R} \times \mathbb{R}_+ \times S^2$ and $\Delta_n$ is an $n$-simplex defined as:

$$\Delta_n := \{(p_1, \cdots, p_n) \in \mathbb{R}^n \,|\, p_1 \leq p_2 \leq \cdots \leq p_n\}. \tag{226}$$

This domain may need to be modified in some Witten diagrams due to the integral over this domain having UV divergences. UV divergences can occur when some points along the Wilson line collide with each other. To avoid such divergences we shall use a *point splitting* regulator, i.e., we shall cut some corners from the simplex $\Delta_n$. Let us denote the regularized simplex as $\widetilde{\Delta}_n$. The exact description of $\widetilde{\Delta}_n$ will vary from diagram to diagram, and we shall describe them as we encounter them.

When we do integration by parts with respect to the differential in a boundary-to-bulk propagator, we get the following three types of terms:

1. A boundary term. Boundaries of our integration domain comes from boundaries of $M$ and $\widetilde{\Delta}_n$. For $M$ we get:
$$\partial M = B_{+\infty} \sqcup B_{-\infty}. \tag{227}$$

   Due to Lemma 8, integrations over $\partial M$ will vanish. Therefore, nonzero contribution to the boundary integration, when we do integration by parts, will only come from the boundary of the regularized simplex, namely $\partial \widetilde{\Delta}_n$. Schematically, the appearance of such a boundary integral will look like:

$$\int_{M^l \times \widetilde{\Delta}_n} \mathrm{d}\theta \wedge (\cdots) = \int_{M^l \times \partial \widetilde{\Delta}_n} \theta \wedge (\cdots) + \cdots. \tag{228}$$

2. The differential acts on a bulk-to-bulk propagator. Due to (112), this identifies the two end points of the propagator, schematically:

$$\mathsf{b} \in \{0, 1\}, \qquad \int_{M^l \times \partial^{\mathsf{b}} \widetilde{\Delta}_n} \mathrm{d}\theta \wedge P \wedge (\cdots) = \int_{M^{l-1} \times \partial^{\mathsf{b}} \widetilde{\Delta}_n} \theta \wedge (\cdots) + \cdots. \tag{229}$$

---

[65] Keep in mind that $\hbar$ has a (length) scaling dimension 1.

3. The differential acts on a step function left by a previous integration by parts. This does not change the domain of integration.

The third option does not to lead a simplification of the domain of integration. Therefore, at the present abstract level, our strategy to simplify an integration is: first go to the boundary of the simplex, and then keep collapsing bulk-to-bulk propagators until we have no more differential left or when no more bulk-to-bulk propagator can be collapsed without the diagram vanishing due the vanishing lemmas from §C.1.

# D  Proof of Lemma 2

All the diagrams that we draw in this section only exist to represent color factors, their numerical values are irrelevant. Which is why we also ignore the color coding we used in the diagrams in the main body of the paper.

We start with yet another lemma:

**Lemma 9.** *The color factor of any Witten diagram with two boundary-to-bulk propagators connected by a single bulk-to-bulk propagator, that is any Witten diagrams with the following configuration:*

$$\tag{230}$$

*upon anti-symmetrizing the color labels of the boundary-to-bulk propagators, involves the following factor:*

$$f_{\mu\nu}{}^{\xi}X_{\xi}, \tag{231}$$

*for some matrix $X_{\xi}$ that transforms under the adjoint representation of $\mathfrak{gl}_K$. In particular, this color factor is the image in $\mathrm{End}(V)$ of some element of $\mathfrak{gl}_K$ where $V$ is the representation of some distant Wilson line.*

*Proof.* The two bulk vertices in the diagram results in the following product of structure constants: $f_{\mu o}{}^{\pi} f_{\nu\rho}{}^{o}$ where the indices $\pi$ and $\rho$ are contracted with the rest of the diagram. Anti-symmetrizing the indices $\mu$ and $\nu$ we get $f_{\mu o}{}^{\pi} f_{\nu\rho}{}^{o} - f_{\nu o}{}^{\pi} f_{\mu\rho}{}^{o}$, which using the Jacobi identity becomes $-f_{\mu\nu}{}^{o} f_{\rho o}{}^{\pi}$. Once $\pi$ and $\rho$ are contracted with the rest of the diagram we get an expression of the general form (231). Furthermore, any expression of the form (231) is an image in $\mathrm{End}(V)$ of some element in $\mathfrak{gl}_K$, since the structure constant $f_{\mu\nu}{}^{\xi}$ can be viewed as a map:

$$f : \wedge^2 \mathfrak{gl}_K \to \mathfrak{gl}_K, \qquad f : t_\mu \wedge t_\nu \mapsto f_{\mu\nu}{}^{\xi} t_\xi. \tag{232}$$

Now composing the above map with a representation of $\mathfrak{gl}_K$ on $V$ gives the aforementioned image. $\qquad\square$

Let us now look at the color factor (161) of the diagram (160), both of which we repeat here:

$$, \qquad f_{\mu}{}^{\xi o} f_{\xi}{}^{\pi\rho} f_{\nu\pi}{}^{\sigma} \varrho(t_o)\varrho(t_\rho)\varrho(t_\sigma). \tag{233}$$

By commuting $\varrho(t_o)$ and $\varrho(t_\rho)$ in the color factor we create a difference which is the color factor of the following diagram:

$$\text{(234)}$$

The key feature of the above diagram is the loop with three propagators attached to it. Such a loop produces a color factor which is a $\mathfrak{gl}_K$-invariant inside $(\mathfrak{gl}_K)^{\otimes 3}$, explicitly we can write a loop and its associated color factor respectively as:

$$\text{and} \quad f_{\mu o}{}^\pi f_{\nu\rho}{}^o f_{\xi\pi}{}^\rho \, . \tag{235}$$

The color factor is $\mathfrak{gl}_K$-invariant since the structure constant itself is such an invariant. To find the invariants in $(\mathfrak{gl}_K)^{\otimes 3}$ we start by writing $\mathfrak{gl}_K$ as:

$$\mathfrak{gl}_K = \mathfrak{sl}_K \oplus \mathbb{C} \, , \tag{236}$$

where by $\mathfrak{sl}_K$ we mean the complexified algebra $\mathfrak{sl}(K, \mathbb{C})$. This gives us the decomposition

$$(\mathfrak{gl}_K)^{\otimes 3} = (\mathfrak{sl}_K)^{\otimes 3} \oplus \cdots \, , \tag{237}$$

where the "$\cdots$" contains summands that necessarily include at leas one factor of the center $\mathbb{C}$. However, none of the three indices that appear in the diagram in (235) can correspond to the center, because each of these indices belong to an instance of the structure constant, which vanishes whenever one of its indices correspond to the center.[66] This means that the $\mathfrak{gl}_K$ invariant we are looking for must lie in $(\mathfrak{sl}_K)^{\otimes 3}$. For $K > 2$, there are exactly two such invariants [57], one of them is the structure constant itself, which is totally anti-symmetric. The other invariant is totally symmetric. However the structure constant is even (invariant) under the $\mathbb{Z}_2$ outer automorphism of $\mathfrak{sl}_K$ whereas the symmetric invariant is odd. Since our theory has this $\mathbb{Z}_2$ as a symmetry, only the structure constant can appear as the invariant in a diagram.[67] This means, as far as the color factor is concerned, we can collapse a loop such as the one in (235) to an interaction vertex. As soon as we do this operation to the diagram (234), Lemma 9 tells us that the color factor of the diagram is an image in $\text{End}(V)$ of an element in $\mathfrak{gl}_K$. This shows that we can swap the positions of any of the two pairs of the adjacent matrices in the color factor in (233) and the difference we shall create is an image of a map $\mathfrak{gl}_K \to \text{End}(V)$. To achieve all permutations of the three matrices wee need to be able to keep swapping positions, let us therefore keep looking forward.

Suppose we commute $\varrho(t_o)$ and $\varrho(t_\rho)$ in (233), then we end up with the color factor of the diagram (159). Now if we commute $\varrho(t_o)$ and $\varrho(t_\sigma)$, we create a difference that corresponds the color factor of the following diagram:

$$\text{(238)}$$

---

[66]In other words, the central abelian photon in $\mathfrak{gl}_K$ interacts with neither itself nor the non-abelian gluons and therefore can not contribute to the diagrams we are considering.

[67]This is also apparent from the way this invariant is written in (235), since the structure constant is invariant under this $\mathbb{Z}_2$, certainly a product of them is invariant as well.

The key feature of this diagram is a loop with four propagators attached to it. The loop and its associated color factor can be written as:

$$f_{\mu\pi}{}^{\tau}f_{o\tau}{}^{\sigma}f_{\nu\sigma}{}^{\rho}f_{\xi\rho}{}^{\pi}. \tag{239}$$

As before, the color factor is a $\mathfrak{gl}_K$-invariant in $(\mathfrak{gl}_K)^{\otimes 4}$. This time, it will be more convenient to write the color factor as a trace. Noting that the structure constants are the adjoint representations of the generators of the algebra we can write the above color factor as:

$$\mathrm{tr}_{\mathrm{ad}}(t_\mu t_o t_\nu t_\xi). \tag{240}$$

The adjoint representation of $\mathfrak{gl}_K$ factors through $\mathfrak{sl}_K$, and the adjoint representation of $\mathfrak{sl}_K$ has a non-degenerate metric with which we can raise and lower adjoint indices. Suitably changing positions of some of the indices in the color factor we can conclude:

$$\mathrm{tr}_{\mathrm{ad}}(t_\mu t_o t_\nu t_\xi) = \mathrm{tr}_{\mathrm{ad}}(t_\mu t_\xi t_\nu t_o). \tag{241}$$

Using the cyclic symmetry of the trace we then find that the color factor is symmetric under the exchange of $\mu$ and $\nu$, therefore when we anti-symmetrize the diagram with respect to $\mu$ and $\nu$ it vanishes.

In summary, starting from the color factor in (233), we can keep swapping any two adjacent matrices and the difference can always be written as an image of some map $\mathfrak{gl}_K \to \mathrm{End}(V)$. The same argument applies to the color factors of all the diagrams in (159). This proves the lemma.

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
