# Peer review of "Topological Holography: The Example of The D2-D4 Brane System"

_SciPost Physics, doi:SciPost Phys. 9, 017 (2020)_

## Round 3 · Referee Report · Anonymous (Referee 1) · 2020-5-9

Strengths

1.) The system under consideration, though slightly exotic, can be argued (Section 6) to arise as a subsector of a physical string theory after a suitable twist + Omega deformation.
2.) Theorem 1, the appearance of the Yangian algebra at large N on both sides of the duality, is supported by an array of detailed and impressive computations and proofs.
3.) Several computations presented in the twisted framework parallel traditional computations in AdS/CFT, following Witten. In particular, the authors take care not just to prove Theorem 1 abstractly but to compute correlation functions and (loop-level) Witten-like diagrams on both sides and extract relations of the Yangian from the results.

Weaknesses

1.) The presentation of the material is a bit uneven: some parts (particularly Appendix B) are written for a mathematical audience and will be inaccessible to most physicists.
2.) There are numerous small grammatical typos throughout the text.
3.) Since the computations are essentially done in flat space, rather than in a fixed AdS-background, some aspects of the connection to physical holography remain puzzling.

Report

This paper is an important contribution to twisted/topological holography (in the sense of Costello-Li) by offering one of the first explicit examples of the twisted duality, and provides new interesting links between mathematics and physics. The technical developments and results are illuminating, and certainly worth publication, and should be of interest to both the physics and mathematics communities. The paper would benefit from a number of relatively minor corrections and clarifications.

Requested changes

1.) There are a number of minor typos and grammatical errors throughout the text, particularly with respect to punctuation, run-on sentences, etc. They are too numerous to list, but the text would strongly benefit from being read through and edited for grammar one more time. Some other typos include:
-'expectations' should be replaced with 'expectation' in the first paragraph
-'Kodaira' is misspelled above equation 9
-'loose' should be 'lose' in Footnote 12
-'describe' is misspelled at the beginning of section 6
-'omega' is uncapitalized in various places (it could simply be replaced with the symbol \Omega as is done elsewhere in the text, for uniformity)
-Rozansky is misspelled on page 38

2.) The authors should consider summarizing the logic of Appendix B at the beginning of the section for a physics audience. Alternatively, if the paper is intended for a dual audience of both mathematicians and physicists, the authors should consider (e.g. in the introduction) summarizing some of the key mathematical results and the key physical results separately and point out where in the text the results are presented to guide the readers.

3.) There is a typo in equation 5 (a minus sign where an equals sign should be).

4.) The text has numerous footnotes, some of which include basic information that would be better served incorporated into the main text (e.g. footnote 6) to avoid breaking up the flow of the text so much.

5.) The authors discuss the equality of bulk and boundary partition functions in the introduction to motivate the isomorphism between the boundary OPE and what they call the scattering algebra. A more refined statement expected in holography is that the Hilbert spaces of bulk and boundary theories are isomorphic, but the algebras of operators of the two theories are claimed to be Koszul dual (and on the CFT side, there is a state-operator correspondence). Could the authors comment on the relationship between the expected isomorphism of the Hilbert spaces in holography and the Koszul duality of the operator algebras they study?

6.) The authors discuss operators spanning the chiral ring (or more generally, operators that generate an associative algebra) as a motivating example when introducing the boundary and scattering algebras. Other interesting subsectors of theories can include vertex algebras with singular OPEs. Can the authors expand on how the scattering algebra should generalize to this case (e.g. do any subtleties involving regularization appear?)

7.) Are the authors neglecting D4-brane backreaction on the closed string fields because they are working in the probe brane approximation & assuming K is small? Or could K even be large and there is a stronger argument for neglecting backreaction, along the lines of the arguments on pages 6/7?

8.) It would be great if the authors could clarify the following somewhere in the text. In Section 6, the authors discuss the relation with physical string theory and motivate the duality they propose along the lines of 'rigid holography' in the sense of Aharony et al. In the latter case, one has fields living on a fixed AdS background, though in this paper the computations are essentially done in flat space (up to KK reduction on an S3 linking the defect). Similarly, the authors quotient out by the center of the gravitational algebra which should encode the closed string operators. The localization discussed in Section 6 therefore looks very much like localizing two field theories coupled together along a common defect, rather than localizing a gauge/gravity dual pair. Can the authors elaborate on the sense in which their 'gravitational' theory is really gravitational? More precisely, can they clarify which aspects of their computations are genuinely holographic, as opposed to computations that could be done in ordinary coupled (twisted) defect field theories? In particular, Koszul duality is known to arise when studying topological line defects in non-gravitational quantum field theories (see, e.g., papers by Costello, Gaiotto-Oh, and Costello-Paquette) and the gravitational backreaction that takes flat space to a (background) AdS space can deform ordinary Koszul duality. That deformation does not arise in this context, so it would be great if the authors could elaborate on these points in Section 6, and perhaps the introduction or discussion sections.

9.) Relatedly, recent twisted holography papers have also appeared by Gaiotto-Oh, Oh-Zhou, Costello-Paquette, and Gaiotto-Abajian that the authors may wish to cite.

  • validity: high
  • significance: high
  • originality: top
  • clarity: ok
  • formatting: reasonable
  • grammar: reasonable

Author:  Nafiz Ishtiaque  on 2020-05-21  [id 831]

(in reply to Report 1 on 2020-05-09)
Category:
remark
answer to question
pointer to related literature

We would like to thank the referee for the detailed review and the very relevant questions. We try to address them in the following.

5) In the particular model we studied the Hilbert space is the representation assigned to the line defects in both theories, which, in the large N limit is infinite dimensional. In the bulk side -- the 4d CS side -- the local operator algebra on the line and what we called the scattering algebra act on this vector space. The local operator algebra (generated by ghost no. 1 operators, we did not discuss them in our paper) is the Koszul dual of the Yangian and the scattering algebra is the Yangian, in the 4d CS side we see both of them acting on the same vector space. In our paper, we were only concerned with establishing the isomorphism between this scattering algebra and the boundary OPE algebra (algebra of operators on the line in the BF theory). It is possible to compactify the 4d CS theory on a 2d surface linking the line defect so that it reduces to a 2d TQFT with two boundaries, one corresponding to the line defect and the other corresponding to a line at infinity where the scattering algebra lives. In this setup the Hilbert space on which both the Yangian and its Koszu dual act is the Hilbert space associated to the "spatial" line joining these two boundaries, the appearance of Koszul duality in this setup is expected generally purely from a TQFT perspective -- as described in the appendix of https://arxiv.org/abs/2001.02177.

6) The scattering algebra is essentially an algebra of modifications of boundary conditions at infinity. If the CFT side of the duality is rich enough to have singular OPE structure then the dual bulk theory will also have rich enough boundary conditions with such pole structures that -- roughly speaking -- collisions of poles will mimic the OPEs. One example of such theories can be found in https://arxiv.org/abs/1812.09257v3 where the holographic dual to certain chiral CFTs were identified as B-model on some Calabi-Yau's.

7) We are considering K to be small, i.e. we think of the D4-branes as mere probes. If K is large there should be a back-reaction/change in the topology of the bulk. This should be analogous to the so called bubbling geometry as it appears in, for example, https://arxiv.org/abs/hep-th/0612190 in the context of holographic duality between 3d CS and topological strings on conifolds.

8) We are going to borrow some arguments from section 6.1 of https://arxiv.org/abs/2001.02177. To the best of our understanding, one way to arrive at holographically dual theories is to start from a coupled open closed theory, schematically effectively described as L_{brane} + L_{bulk} + L_{int}. Then in the decoupling limit, we lose the interaction term, the brane theory becomes the CFT side of the duality, and generally, the bulk term is deformed to describe a theory on a backreacted geometry which becomes the holographic dual to the CFT side. What we find in our particular model is that the L_{bulk} (the 4d CS side) is too simple/rigid in a sense to notice any backreaction. Therefore, it seems to be a valid remark that the setup still looks like the setup for two coupled theories in flat space. We think that our computation can indeed be phrased as demonstrating in a particular example the universal expectation that line defects that can be coupled to a TQFT are described by an algebra which is Koszul dual to the algebra of the TQFT. However, since we arrive at our setup following the procedure that one would follow to arrive at holographically dual theories, we claim that the two sides (the BF side and the CS side) can just as well be thought of as holographic duals to each other.

These are certainly some important points that have become clearer to us over the course of time, we thank the reviewer for mentioning these. We look forward to any further questions, corrections, or recommendations that may arise.

We will correct the grammatical errors and add the references that have come out since writing this paper.

---

## Round 4 · Referee Report · Anonymous · 2020-7-2

Report

The authors have clarified several issues that were raised in the previous referee report, and have further improved the readability of the paper. Therefore, we are happy to recommend the paper for publication in its current form.

---

## Round 4 · Author Response

This is a resubmission with minor revision as per editorial recommendation.

---

## Round 4 · List of Changes

1. Corrected several spellings, and a typo in eq. 5.
2. At the end of section 1 pointed out relevant new references that came out in the last couple of years.
3. Slightly expanded the introduction to appendix B to better clarify the motivation and logic behind the mathematical results to follow.
4. Some footnotes have been moved to the main text.
5. Commented on the special nature of the lack of backreaction in the 4d Chern-Simons theory after eq. 14 with references to literature with different examples with and without backreaction.

You are currently on this page

Resubmission 1809.00372v4 on 2 July 2020

---

## Editorial Decision

published